Journal of Data-centric Machine Learning Research (2024)          Submitted 10/23; Revised 01/24; Published 01/24

# Benchmarking Robustness of Multimodal Image-Text Models under Distribution Shift

**Jielin Qiu**[1*]                                    JIELINQ@CS.CMU.EDU

**Yi Zhu**[2*]                                        YI@BOSON.AI

**Xingjian Shi**[2*]                                  XINGJIAN@BOSON.AI

**Florian Wenzel**[3*]                                FLORIAN@MIRELO.AI

**Zhiqiang Tang**[4]                                  ZQTANG@AMAZON.COM

**Ding Zhao**[1]                                      DINGZHAO@CMU.EDU

**Bo Li**[4,5]                                        LBO@ILLINOIS.EDU

**Mu Li**[2*]                                         MU@BOSON.AI

[1] *Carnegie Mellon University*

[2] *Boson AI*

[3] *Mirelo AI*

[4] *Amazon Web Services*

[5] *University of Chicago*

**Reviewed on OpenReview:** *https://openreview.net/forum?id=Vc1fXQ8mJg*

**Editor:** Hongyang Zhang

## Abstract

Multimodal image-text models have shown remarkable performance in the past few years. However, evaluating robustness against distribution shifts is crucial before adopting them in real-world applications. In this work, we investigate the robustness of 12 popular open-sourced image-text models under common perturbations on five tasks (image-text retrieval, visual reasoning, visual entailment, image captioning, and text-to-image generation). In particular, we propose several new multimodal robustness benchmarks by applying 17 image perturbation and 16 text perturbation techniques on top of existing datasets. We observe that multimodal models are not robust to image and text perturbations, especially to image perturbations. Among the tested perturbation methods, character-level perturbations constitute the most severe distribution shift for text, and zoom blur is the most severe shift for image data. We also introduce two new robustness metrics (**MMI** for MultiModal Impact score and **MOR** for Missing Object Rate) for proper evaluations of multimodal models. We hope our extensive study sheds light on new directions for the development of robust multimodal models. More details can be found on the project webpage: https://MMRobustness.github.io.

**Keywords:** Multimodal, Robustness, Distribution Shift

## 1 Introduction

Multimodal learning has drawn increasing attention, and many datasets and models are collected and proposed to accelerate research in this field (Chen et al., 2020; Gan et al., 2020; Li et al., 2022b, 2020b; Zhang et al., 2021; Radford et al., 2021; Kim et al., 2021; Li et al., 2021a, 2022a;

---

∗. work done while at Amazon

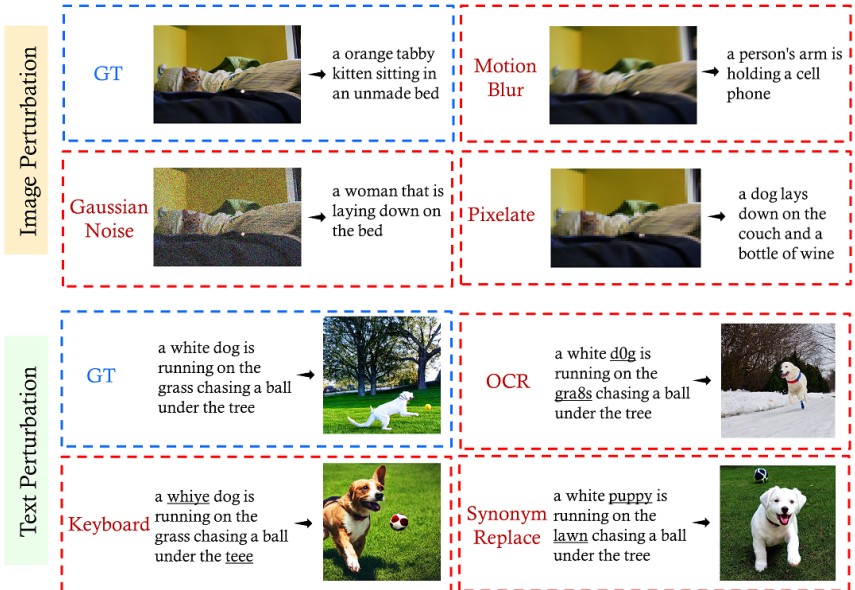

Figure 1: Multimodal models are sensitive to image/text perturbations (original image-text pairs are shown in blue boxes, perturbed ones are in red). Image captioning (Top): Adding image perturbations can result in incorrect captions, e.g., the tabby kitten is mistakenly described as a woman/dog. Text-to-image generation (bottom): Applying text perturbations can result in the generated images containing incomplete visual information, e.g., the tree is missing in the two examples above.

Yang et al., 2022; Dou et al., 2021; Ramesh et al., 2022; Wang et al., 2022b; Alayrac et al., 2022; Radford et al., 2021; Yu et al., 2022). Despite the extraordinary performance and exciting potential, we find that multimodal models are often vulnerable under distribution shifts. In Figure 1, we show interesting examples of image captioning under image perturbations using BLIP (Li et al., 2022a), and text-to-image generation under text perturbations using Stable Diffusion (Rombach et al., 2022). For image captioning, we observe that by simply adding noise, blur, or pixelation to the original image, the generated captions become incorrect. For text-to-image generation, applying keyboard typos, OCR errors, or synonym replacements to the original sentence, can lead to generated images containing incomplete visual information.

There is a sizable literature on robustness evaluation of unimodal vision models (Yin et al., 2019; Zheng et al., 2016; Drenkow et al., 2021; Djolonga et al., 2021; Goyal et al., 2022; Paul and Chen, 2022; Bhojanapalli et al., 2021; Mahmood et al., 2021; Mao et al., 2021; Aldahdooh et al., 2021; Zhou et al., 2022; Wenzel et al., 2022) or unimodal language models (Wang et al., 2022c; Chang et al., 2021; Wang et al., 2020; Rychalska et al., 2019; Goel et al., 2021; Singh et al., 2021; Dong et al., 2021; Gui et al., 2021; Malfa and Kwiatkowska, 2022; Wang et al., 2021). Several recent work (Galindo and Faria, 2021; Fort, 2021; Noever and Noever, 2021; Goh et al., 2021; Daras and Dimakis, 2022) have unsystematically tested or probed a few pre-trained multimodal models, including CLIP (Radford et al., 2021) and DALL-E 2 (Ramesh et al., 2022). However, the robustness evaluation of multimodal image-text models under distribution shift has rarely been studied. To our best knowledge, there is currently no benchmark dataset nor a comprehensive study of how the perturbed data can affect their performance. Hence in this work:

- We build multimodal robustness evaluation benchmarks by leveraging existing datasets and tasks, e.g., image-text retrieval (Flicker30K, COCO), visual reasoning (NLVR2), visual

entailment (SNLI-VE), image captioning (COCO), and text-to-image generation (COCO). We analyze the robustness of 12 multimodal models under distribution shifts, which include 17 image perturbation and 16 text perturbation methods.

- We introduce two new robustness metrics, one termed MMI (MultiModal Impact score), to account for the relative performance drop under distribution shift in 5 downstream applications. The other one is named MOR (Missing Object Rate), which is based on open-set language-guided object detection and the first object-centric metric proposed for text-to-image generation evaluation.

- We find that multimodal image-text models are more sensitive to image perturbations than text perturbations. In addition, *zoom blur* is the most effective attack for image perturbations, while character-level perturbations show a higher impact than word-level and sentence-level perturbations for text. In addition, we provided interpretations of performance drop by different perturbation methods using Optimal Transport alignment and attention.

## 2 Related Work

**Multimodal Learning** has advanced quickly in recent years with appealing applications in different fields, i.e., embodied learning (Bisk et al., 2020; Hu et al., 2019; Jain et al., 2022; Min et al., 2022), multimedia image/video and language understanding (Zolfaghari et al., 2021; Erickson et al., 2022; Rombach et al., 2022; Hu et al., 2022), and psychology (Liu et al., 2022b; Han et al., 2022). Thanks to the larger datasets (Radford et al., 2021; Yuan et al., 2021; Schuhmann et al., 2021, 2022; Patraucean et al., 2022) and larger transformer models (Zhai et al., 2022; Chen et al., 2022; Brown et al., 2020; Chowdhery et al., 2022; Liang et al., 2022), many powerful multimodal image-text models have been developed and shown great capability. However, unlike unimodal models, the robustness study of multimodal models under distribution shift has rarely been explored.

**Robustness of Multimodal Models** There is a sizable literature on robustness evaluation of unimodal vision models (Yin et al., 2019; Zheng et al., 2016; Drenkow et al., 2021; Djolonga et al., 2021; Goyal et al., 2022; Paul and Chen, 2022; Bhojanapalli et al., 2021; Mahmood et al., 2021; Mao et al., 2021; Aldahdooh et al., 2021; Zhou et al., 2022) or unimodal language models (Wang et al., 2022c; Chang et al., 2021; Wang et al., 2020; Rychalska et al., 2019; Goel et al., 2021; Singh et al., 2021; Dong et al., 2021; Gui et al., 2021; Malfa and Kwiatkowska, 2022; Wang et al., 2021). However, robustness evaluation of multimodal image-text models under distribution shift has rarely been studied (Goh et al., 2021; Daras and Dimakis, 2022). Previous works Galindo and Faria (2021); Fort (2021); Goh et al. (2021); Noever and Noever (2021) have unsystematically tested some pre-trained models, i.e., CLIP Radford et al. (2021), by attacking with text patches and adversarial pixel perturbations. Daras and Dimakis (2022) found that DALLE-2 (Ramesh et al., 2022) has a hidden vocabulary that can be used to generate images with absurd prompts. Fang et al. (2022) found that diverse training distribution is the main cause for robustness gains. Cho et al. (2022) studied the text-to-image generative models about visual reasoning skills and social bias. For benchmarks, Li et al. (2021b) collected an Adversarial VQA dataset to evaluate the robustness of VQA models. Schiappa et al. (2022) studied the robustness of video-text models under perturbations, but they only focused on one video-text retrieval task. In this work, we conduct a systematic robustness evaluation of recent multimodal image-text models on 5 different downstream tasks based on new datasets and metrics. (More related work can be found in Appendix J).

## 3 Multimodal Robustness Benchmark

Distribution shift is one of the significant problems of applying models in real-world scenarios (Taori et al., 2020; Liu et al., 2022c). Distribution shift happens when the training data distribution $p_{tr}(\boldsymbol{x} \mid \boldsymbol{y})$ is different from the data distribution to which the model has applied at test time $p_{te}(\boldsymbol{x} \mid \boldsymbol{y})$. A model is said to be robust on the out-of-distribution (OOD) data, if it still produces accurate predictions on the test data. To evaluate the robustness of large pretrained multimodal models under distribution shift, we start by building several evaluation benchmark datasets via perturbing the original image-text pairs on either the image side or text side. We use these perturbations to simulate distribution shifts of various intensities and use them to stress-test the robustness of the given models.

### 3.1 Image Perturbation

To simulate distribution shifts for the image data, we adopt the perturbation strategies from ImageNet-C (Hendrycks and Dietterich, 2019) and Stylize-ImageNet (Geirhos et al., 2019; Michaelis et al., 2019). We include Stylize-ImageNet for its effectiveness in perturbing the original image by breaking its shape and texture (Geirhos et al., 2019). Examples of the perturbed images can be seen in Figure 2. The perturbations are grouped into five categories: **noise, blur, weather, digital**, and **stylize**. Specifically, we use 17 image perturbation techniques, (1) Noise: *Gaussian noise, shot noise, impulse noise, speckle noise*; (2) Blur: *defocus blur, frosted glass blur, motion blur, zoom blur*; (3) Weather: *snow, frost, fog, brightness*; (4) Digital: *contrast, elastic, pixelate, JPEG compression*; and (5) *stylize*. Note that real-world corruptions can manifest themselves at varying intensities, we thus introduce variation for each corruption following (Hendrycks and Dietterich, 2019; Geirhos et al., 2019; Michaelis et al., 2019). In our evaluation setting, each category has five levels of severity, resulting in 85 perturbation methods in total. More details can be found in Appendix Sec. A. Note that these strategies are commonly considered synthetic distribution shifts and can serve as a good starting point since they are precisely defined and easy to apply.

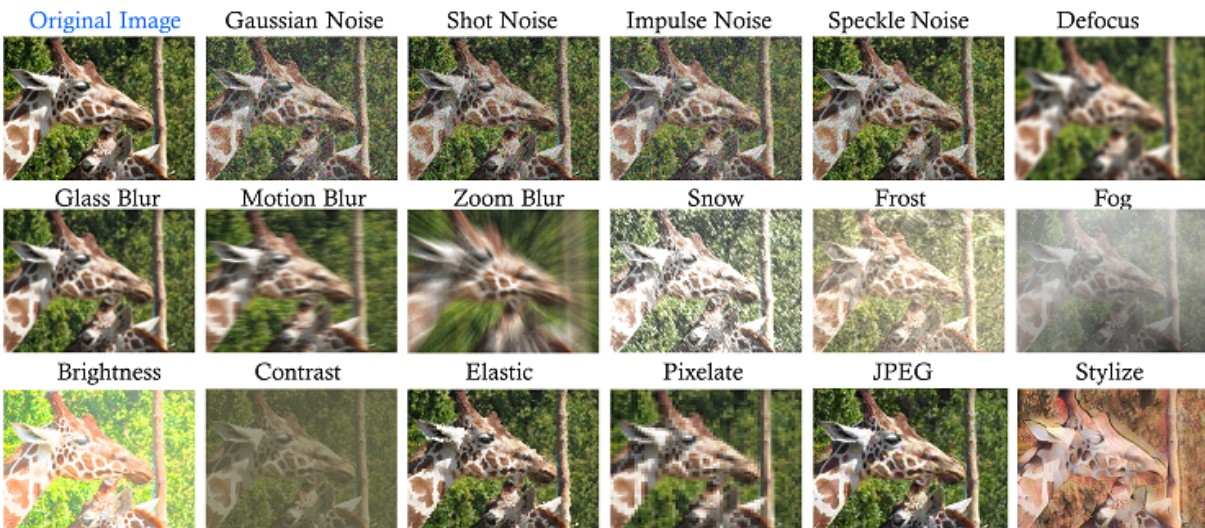

Figure 2: Examples of our 17 image perturbations. The original image is taken from the COCO dataset and shown on the top left.

## 3.2 Text Perturbation

To simulate the distribution shifts in language, we design 16 text perturbation techniques grouped into three categories: **character-level, word-level**, and **sentence-level**. Examples of the text perturbations are shown in Table 1. In detail, for character-level perturbation, we adopt 6 strategies from (Ma, 2019), including *keyboard, OCR, character insert (CI), character replace (CR), character swap (CS), character delete (CD).* These perturbations can be considered as simulating real-world typos or mistakes during typing. For word-level perturbation, we adopt 5 strategies from EDA and AEDA (Wei and Zou, 2019; Karimi et al., 2021), including *synonym replacement (SR), word insertion (WR), word swap (WS), word deletion (WD),* and *insert punctuation (IP).* These perturbations aim to simulate different writing habits that people may replace, delete, or add words to express the same meaning. For sentence-level perturbation, (1) we first adopt the style transformation strategies from (Li et al., 2018; Etinger and Black, 2019; Schmidt, 2020; Schiappa et al., 2022), i.e., transferring the style of text into *formal, casual, passive,* and *active*; (2) we also adopt the *back translation* method from (Ma, 2019). These perturbations will focus more on language semantics, due to the differences in speaking or writing styles, or translation errors. Similar to image perturbations, we introduce severity levels to each strategy. For strategies within the character-level and word-level perturbations, we apply 5 severity levels similar to image perturbations, while for strategies within the sentence-level perturbations, there is only one severity level. This leads to a total of 60 text perturbation methods. More details about each text perturbation strategy can be found in Appendix Sec. A. We emphasize that these perturbation techniques cover some of the actual text distribution shifts we encounter in real-world applications (e.g., typos, word swaps, style changes, etc.). Models for text data that are deployed in real-world settings need to be robust with respect to these perturbations.

Table 1: Example of our 16 text perturbations. The original text is taken from the COCO dataset and denoted as clean in the first row.

| Category | Perturbation | Example |
|---|---|---|
| Original | Clean | An orange metal bowl strainer filled with apples. |
| Character | Keyboard | An orange metal bowk strainer filled witj apples. |
| | OCR | An 0range metal bowl strainer filled with app1es. |
| | CI | And orange metal bowl strainer filled with atpples. |
| | CR | An orange metal towl strainer fillet with apples. |
| | CS | An orange meatl bowl stariner filled with apples. |
| | CD | An orang[X] metal bowl strainer fil[X]ed with apples. |
| Word | SR | An orange alloy bowl strainer filled with apples. |
| | WI | An old orange metal bowl strainer filled with apples. |
| | WS | An orange metal strainer bowl filled with apples. |
| | WD | An orange metal bowl strainer [X] with apples. |
| | IP | An orange metal bowl ? strainer filled with apples. |
| Sentence | Formal | An orange metal bowl strainer contains apples. |
| | Casual | An orange metal bowl is filled with apples. |
| | Passive | Some apples are in an orange metal bowl strainer. |
| | Active | There are apples in an orange metal bowl strainer. |
| | Back trans | Apples are placed in an orange metal bowl strainer. |

Table 2: Evaluation tasks, datasets, models and metrics used in our study.

| Task | Datasets | Models | Evaluation metrics |
|---|---|---|---|
| Image-text Retrieval | Flicker30K, COCO | CLIP, ViLT, TCL, ALBEF, BLIP | Recall R@K, K={1,5,10}, and RSUM |
| Visual Reasoning | NLVR2 | ALBEF, ViLT, BLIP, TCL, METER | Prediction accuracy |
| Visual Entailment | SNLI-VE | ALBEF, TCL, METER | Prediction accuracy |
| Image Captioning | COCO | BLIP, GRIT, LLaVA, Mini-GPT4, BLIP2 | BLEU, METEOR, ROUGE-L, CIDEr |
| Text-to-image Generation | COCO | Stable Diffusion, GLIDE | FID, CLIP-FID, MOR (ours) |

**Fidelity** To build a convincing benchmark, we need to ensure that the perturbed text has the same semantics as the original one. Otherwise, for image-text pairs in multimodal learning, the perturbed text will not match the original image and, hence, would no longer represent a meaningful image-text pair. In this work, we use paraphrases from pretrained sentence-transformers (Reimers and Gurevych, 2019) to evaluate the semantic similarity between the original and the perturbed sentences. Specifically, "paraphrase-mpnet-base-v2" (Reimers and Gurevych, 2019) is used to extract the original and perturbed sentence embeddings for computing similarity score $\alpha_s$. Given a predefined tolerance threshold $\alpha_0$, a higher score $\alpha_s > \alpha_0$ means the perturbed text still has similar semantics with the original text. However, if $\alpha_s < \alpha_0$ indicating their semantics are different, we will perturb the sentence again until the semantic similarity score meets the requirement, in a reasonable looping time $N_{max} = 100$. Beyond $N_{max}$, we will remove this text sample from our robustness benchmark. More details about the fidelity control process can be found in Appendix Sec. A. This procedure guarantees semantic closeness and ensures our perturbed data could serve as a valid evaluation benchmark for multimodal image-text models.

## 4 Experiments

Using our multimodal robustness benchmark, we are able to answer the following questions: **(1)** How robust are multimodal pretrained image-text models under distribution shift? **(2)** What is the sensitivity of each model under different perturbation methods? **(3)** Which model architecture or loss objectives might be more robust under image or text perturbations? **(4)** Are there any particular image/text perturbation methods that can consistently show significant influence?

### 4.1 Evaluation Tasks, Datasets and Models

As shown in Table 2, we select five widely adopted downstream tasks for a comprehensive robustness evaluation under distribution shift, including image-text retrieval, visual reasoning (VR), visual entailment (VE), image captioning, and text-to-image generation. For each task, we perturb the corresponding datasets, i.e., Flickr30K (Young et al., 2014), COCO (Lin et al., 2014), NLVR2 (Suhr et al., 2017), and SNLI-VE (Xie et al., 2018, 2019b), using the image perturbation (IP) and text perturbation (TP) methods introduced in Sec. 3. This leads to our 8 benchmark datasets: (1) Flickr30K-IP, Flickr30K-TP, COCO-IP, and COCO-TP for image-text retrieval evaluation; (2) NLVR2-IP and NLVR2-TP for visual reasoning evaluation; (3) SNLI-VE-IP and SNLI-VE-TP for visual entailment evaluation; (4) COCO-IP for image captioning evaluation; and (5) COCO-TP for text-to-image generation evaluation. We select 12 representative large multimodal models, which have publicly released their code and pretrained weights: CLIP (Radford et al., 2021), ViLT (Kim et al., 2021), ALBEF (Li et al., 2021a), BLIP (Li et al., 2022a), TCL (Yang et al., 2022), METER (Dou et al., 2021), GRIT (Nguyen et al., 2022), LLaVa (Liu et al., 2023), Mini-GPT4 (Zhu et al., 2023), BLIP2 (Li et al., 2023), GLIDE (Nichol et al., 2022) and Stable

Diffusion (Rombach et al., 2022). We appreciate the authors for making their models publicly available.

## 4.2 Evaluation Metrics

We adopt standard evaluation metrics for each task. To be specific, for image-text retrieval, we use recall and RSUM (i.e., the sum of recall R@K metric (Wu et al., 2019)). As for visual reasoning and visual entailment tasks, we use prediction accuracy. For image captioning, we use standard text evaluation metrics, i.e., BLEU (Papineni et al., 2002), METEOR (Denkowski and Lavie, 2014), ROUGE-L (Lin, 2004), and CIDEr (Vedantam et al., 2015). For text-to-image generation, we use FID (Heusel et al., 2017) and CLIP-FID (Kynkaanniemi et al., 2022; Parmar et al., 2022) scores, and our proposed MOR (details will be introduced later) to evaluate the quality of the generated images.

**MultiModal Impact score (MMI)**  To evaluate the robustness of a model, it is crucial to measure the relative performance drop between the in-distribution (ID) and out-of-distribution (OOD) performance. Recall the example given by Taori et al. (2020), let $d_1$ be the ID dataset (where the model is trained), and $d_2$ be an OOD dataset, then a model $m_1$ should be considered more robust than model $m_2$ if $m_1$'s performance drop is less significant than $m_2$ when evaluated from $d_1$ to $d_2$, even though $m_2$'s absolute accuracy/recall on $d_2$ may still be higher than $m_1$'s. To quantitatively measure the robustness of multimodal image-text models, we introduce a new robustness evaluation metric, termed **M**ulti**M**odal **I**mpact score (MMI). We compute MMI as the averaged performance drop compared with the non-perturbed performance ("clean"), i.e., $\text{MMI} = (s_c - s_p)/s_c$ where $s_p$ is the perturbed score and $s_c$ is the clean score. Here, the score can be any standard metric mentioned above, e.g., recall, RSUM, accuracy, FID, and CLIP-FID. In the following experiments, we report both the standard evaluation metrics on the perturbed (OOD) datasets as well as their corresponding MMI variants. More details about experimental settings can be found in Appendix Sec. B.

## 4.3 Robustness Evaluation under Distribution Shift

**Image-text retrieval**  We present the evaluation results under image perturbations in Table 3 [Top] and results under text perturbations in Table 3 [Bottom]. For simplicity, we only report the RSUM scores here, and the detailed results on each recall (i.e., R1, R5, and R10) and perturbation level can be found in Appendix Sec. C.

Inspecting Table 3 [Top], we observe that the performance of all models drops under image perturbation. Although different perturbation methods have various impacts on different models, we observe the following general trends. We find that most multimodal models are most sensitive to *zoom blur*. Additionally, we find that *glass blur* and *brightness* are the two "softest" perturbation methods, where the performance of all evaluated models deteriorates the least. Comparing the MMI score for both Flickr30K and COCO datasets, CLIP zero-shot (ZS) is more robust than other models, possibly due to it being trained on the large WIT400M dataset (Radford et al., 2021). As indicated in Taori et al. (2020), training models on large and diverse datasets often leads to increased robustness. For text perturbations in Table 3 [Bottom], we also find the performance of all models drop. In addition, we observe the following general trends. Character-level perturbations show more influence than word-level and sentence-level perturbations. In particular, *keyboard* and *character replace (CR)* consistently show a high impact on models' robustness, while *insert punctuation (IP)*, *formal*, and *active* are the least effective text perturbations.

Table 3: **Image-text retrieval.** [**Top**] Robustness evaluations on Flickr30k-IP and COCO-IP. [**Bottom**] Robustness evaluations on Flickr30k-TP and COCO-TP datasets. We report averaged RSUM where the most effective perturbation results are marked in bold, and the least effective perturbation results are underlined. The MMI impact score is marked in blue, the lower the better.

| Dataset | Method | Clean | Noise | | | | Blur | | | | Weather | | | | Digital | | | | Stylize | ave | MMI |
| --- | --- | --- | --- | --- | --- | --- | --- | --- | --- | --- | --- | --- | --- | --- | --- | --- | --- | --- | --- | --- | --- |
| | | | Gauss. | Shot | Impulse | Speckle | Defocus | Glass | Motion | Zoom | Snow | Frost | Fog | Bright | Contrast | Elastic | Pixel | JPEG | Stylize | | |
| Flickr30K | ViLT FT | 522.0 | 413.0 | 419.6 | 396.9 | 387.1 | 417.6 | 489.0 | 388.4 | 236.3 | 332.7 | 453.1 | 455.8 | _496.9_ | 372.2 | 461.7 | **277.4** | 487.6 | 387.1 | 408.7 | ↓ 21.7% |
| | CLIP ZS | 533.7 | 501.7 | 504.2 | 481.2 | 515.5 | 502.1 | _530.1_ | 509.7 | 457.8 | 470.7 | 495.6 | 519.7 | _530.1_ | 515.4 | 510.4 | 469.5 | 524.6 | **447.6** | 499.2 | ↓ 6.5% |
| | CLIP FT | 544.3 | 500.1 | 503.8 | 479.1 | 522.1 | 493.3 | _536.9_ | 513.3 | **444.4** | 464.4 | 503.2 | 529.7 | 543.5 | 521.5 | 513.9 | 453.9 | 528.6 | 436.9 | 499.3 | ↓ 8.3% |
| | TCL ZS | 563.8 | 464.9 | 467.0 | 458.4 | 498.0 | 429.8 | 506.6 | 388.5 | **251.3** | 407.3 | 449.5 | 434.2 | _509.1_ | 473.2 | 434.4 | 247.2 | 502.2 | 343.4 | 427.4 | ↓ 24.2% |
| | TCL FT | 573.4 | 529.9 | 532.6 | 527.7 | 551.6 | 504.5 | _566.0_ | 513.9 | 397.3 | 521.7 | 551.0 | 554.1 | 568.0 | 557.1 | 421.0 | **372.0** | 555.4 | 448.7 | 516.2 | ↓ 10.0% |
| | ALBEF FT | 577.7 | 533.8 | 538.3 | 532.0 | 557.8 | 528.8 | 569.2 | 516.0 | **416.1** | 532.0 | 558.1 | 560.4 | _572.0_ | 550.6 | 538.7 | 435.9 | 559.8 | 464.1 | 527.3 | ↓ 8.7% |
| | BLIP FT | 580.9 | 536.2 | 538.9 | 528.6 | 560.8 | 529.4 | 571.6 | 525.7 | **412.1** | 456.6 | 513.4 | 568.5 | _574.4_ | 555.1 | 545.6 | 490.8 | 563.8 | 482.1 | 527.2 | ↓ 9.2% |
| COCO | ViLT | 441.5 | 372.2 | 372.6 | 362.9 | 396.7 | 378.1 | _432.0_ | 365.4 | **193.7** | 281.1 | 366.1 | 398.1 | 422.4 | 327.1 | 402.2 | 229.8 | 425.8 | 333.9 | 356.5 | ↓ 19.3% |
| | CLIP ZS | 394.5 | 363.0 | 361.2 | 330.2 | 368.7 | 358.7 | 391.6 | 362.2 | 294.6 | **294.7** | 329.0 | 371.8 | _391.9_ | 356.4 | 369.7 | 308.2 | 388.0 | 314.9 | 350.3 | ↓ 11.2% |
| | CLIP FT | 420.5 | 367.2 | 365.3 | 331.7 | 381.5 | 371.0 | 412.2 | 374.4 | 291.0 | **289.3** | 337.3 | 389.9 | _413.9_ | 371.7 | 379.7 | 306.4 | 402.1 | 310.2 | 358.5 | ↓ 14,7% |
| | TCL ZS | 477.2 | 419.8 | 418.4 | 418.4 | 439.0 | 400.0 | 450.8 | 357.5 | **177.3** | 316.5 | 372.0 | 400.6 | _452.2_ | 416.1 | 369.0 | 190.3 | 442.7 | 280.1 | 371.8 | ↓ 22.1% |
| | TCL FT | 497.2 | 454.3 | 454.4 | 453.9 | 468.1 | 447.8 | _491.9_ | 433.8 | **259.9** | 408.9 | 443.2 | 470.1 | 489.1 | 467.8 | 438.2 | 309.1 | 474.9 | 360.9 | 430.9 | ↓ 13.3% |
| | ALBEF FT | 504.6 | 460.0 | 460.6 | 460.3 | 376.4 | 447.1 | 493.0 | 436.5 | **282.2** | 408.8 | 449.8 | 472.6 | _493.8_ | 452.1 | 455.0 | 347.0 | 480.9 | 475.8 | 438.3 | ↓ 13.1% |
| | BLIP FT | 516.6 | 471.9 | 472.1 | 467.7 | 489.5 | 466.1 | _507.2_ | 451.7 | **291.6** | 432.8 | 471.8 | 494.2 | 506.8 | 470.4 | 472.3 | 404.7 | 499.6 | 402.9 | 458.7 | ↓ 11.2% |

| Dataset | Method | Clean | Character-level | | | | | | Word-level | | | | | Sentence-level | | | | | ave | MMI |
| --- | --- | --- | --- | --- | --- | --- | --- | --- | --- | --- | --- | --- | --- | --- | --- | --- | --- | --- | --- | --- |
| | | | Keyboard | OCR | CI | CR | CS | CD | SR | WI | WS | WD | IP | Formal | Casual | Passive | Active | Back_trans | | |
| Flickr30K | ViLT FT | 522.0 | **385.3** | 461.9 | 388.0 | 386.2 | 395.6 | 398.6 | 471.9 | 492.2 | 480.1 | 489.8 | 507.7 | _510.1_ | 504.5 | 488.1 | 508.3 | 500.1 | 460.5 | ↓ 11.8% |
| | CLIP ZS | 533.7 | **431.8** | 478.2 | 450.5 | 435.2 | 444.6 | 451.3 | 497.1 | 509.6 | 503.3 | 514.1 | 519.4 | _531.7_ | 529.3 | 524.8 | 531.4 | 524.2 | 492.3 | ↓ 7.8% |
| | CLIP FT | 544.3 | **458.4** | 500.1 | 477.6 | 461.6 | 471.1 | 475.5 | 515.4 | 530.4 | 526.0 | 531.1 | 536.4 | _545.8_ | 542.1 | 537.9 | 545.1 | 537.3 | 512.0 | ↓ 5.9% |
| | TCL ZS | 563.8 | 433.3 | 499.9 | 443.3 | **428.4** | 444.4 | 448.9 | 511.9 | 523.8 | 519.1 | 528.8 | _548.6_ | 544.4 | 542.4 | 530.1 | 547.1 | 535.8 | 501.9 | ↓ 11.0% |
| | TCL FT | 573.4 | 494.3 | 545.0 | 504.9 | **492.8** | 501.9 | 502.4 | 554.7 | 566.4 | 560.0 | 564.2 | _573.4_ | 571.5 | 569.6 | 562.8 | 572.1 | 566.5 | 543.9 | ↓ 5.1% |
| | ALBEF FT | 577.7 | 506.2 | 552.0 | 516.2 | **505.0** | 511.7 | 513.0 | 561.9 | 571.6 | 568.6 | 570.0 | _577.7_ | 576.2 | 575.0 | 569.5 | 576.4 | 572.5 | 551.5 | ↓ 4.5% |
| | BLIP FT | 580.9 | **518.0** | 559.5 | 527.3 | **518.0** | 526.5 | 525.7 | 565.6 | 576.1 | 572.8 | 573.8 | _580.7_ | 579.0 | 578.6 | 574.5 | 579.6 | 574.7 | 558.1 | ↓ 3.9% |
| COCO | ViLT | 441.5 | **319.2** | 386.2 | 327.0 | 321.7 | 333.1 | 334.1 | 397.8 | 417.5 | 404.4 | 413.6 | 433.1 | _436.5_ | 433.6 | 423.2 | 437.1 | 426.0 | 390.3 | ↓ 11.6% |
| | CLIP ZS | 394.5 | 285.5 | 286.4 | 286.1 | **285.4** | 285.6 | 285.8 | 347.5 | 363.8 | 355.5 | 368.6 | 374.2 | 393.0 | 391.6 | 379.6 | _393.5_ | 381.2 | 341.5 | ↓ 13.4% |
| | CLIP FT | 420.5 | 316.1 | 316.7 | 316.5 | **316.4** | 316.7 | 315.6 | 376.2 | 394.6 | 389.9 | 395.3 | 406.6 | 417.3 | 415.2 | 408.7 | _419.4_ | 406.2 | 370.5 | ↓ 11.9% |
| | TCL ZS | 477.2 | **368.0** | 428.4 | 381.3 | 368.4 | 382.0 | 383.4 | 439.3 | 453.4 | 439.3 | 450.9 | _477.2_ | 474.4 | 471.8 | 464.7 | 475.7 | 462.0 | 432.9 | ↓ 9.3% |
| | TCL FT | 497.2 | **397.8** | 455.1 | 412.0 | 398.5 | 408.8 | 410.5 | 463.7 | 481.3 | 471.8 | 477.7 | _497.1_ | 494.6 | 493.0 | 487.3 | 496.0 | 483.5 | 458.0 | ↓ 7.9% |
| | ALBEF FT | 504.6 | **404.5** | 461.7 | 418.9 | 406.1 | 414.7 | 415.5 | 471.4 | 488.9 | 483.3 | 486.3 | _504.5_ | 503.1 | 502.0 | 496.4 | 503.7 | 491.3 | 465.8 | ↓ 7.7% |
| | BLIP FT | 516.6 | **429.1** | 479.1 | 442.4 | 430.8 | 441.3 | 441.4 | 484.3 | 502.1 | 494.6 | 499.7 | _515.8_ | 514.4 | 513.6 | 508.1 | 515.4 | 504.3 | 482.3 | ↓ 6.6% |

For both image and text perturbations, we see that BLIP shows the best robustness performance on two datasets, i.e., the lowest MMI score. We hypothesize that using an encoder-decoder architecture and generative language modeling objective in BLIP is helpful for image-text retrieval. Given the recent paradigm shift to using generative loss objectives in pre-training multimodal models, e.g., BLIP (Li et al., 2022a), CoCa (Yu et al., 2022), SimVLM (Wang et al., 2022d) PaLI (Chen et al., 2022), Unified-IO (Lu et al., 2022), OFA (Wang et al., 2022b), we believe this observation could be generalized to other multimodal tasks.

We provide qualitative evidence by visualizing the cross-modal alignment between the image patch and word query using optimal transport (Kim et al., 2021). As shown in Figure 3, when using GT image-text pair, the retrieval model can accurately locate the image patches given word query. After image perturbations, in particular the ones with high impact like *pixelate* and *zoom blur*, we can clearly see that the model has difficulties finding the correct alignment. However, for the "softest" perturbations like *brightness* and *glass blur*, the model is still able to generate a transport plan (OT coupling matrix) between word and image patch. Similarly, in Figure 4 where the text are perturbed, we can see the retrieval model cannot locate the correct word query under *keyboard* and *CR*, but still functions well under *IP* and *formal*. Overall, the visualization of word patch alignments in Figure 3 and 4 confirm the conclusion drawn from Table 3, showing that the alignments are worst for perturbations that lead to highest performance degradation.

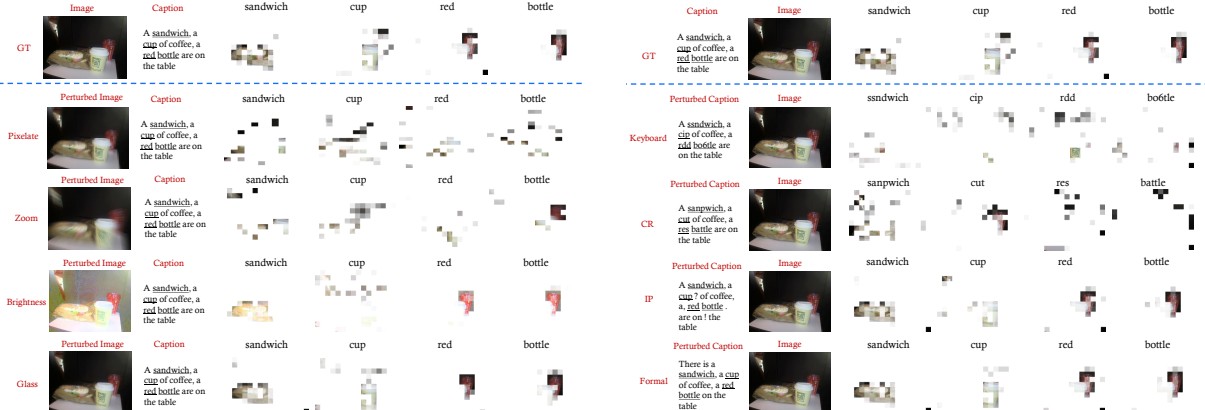

Figure 3: Optimal Transport (OT) alignment visualization between text and **perturbed images**, where *pixelate* and *zoom blur* are two high-effective image perturbation methods, *brightness* and *glass blur* are two low-effective ones.

Figure 4: Optimal Transport (OT) alignment visualization between **perturbed text** and images, where *keyboard* and *character replace* are two high-effective text perturbation methods, *insert punctuation* and *formal* are two soft ones.

**Visual reasoning and visual entailment**   These two tasks are commonly considered to be multimodal classification problems. We present the accuracy results in Tables 11 & 13, and Tables 12 & 14 (in Appendix Sec. D and Appendix Sec. E) under image and text perturbations, respectively.

For both the visual reasoning (VR) and visual entailment (VE) task, we observe that *zoom blur* consistently impacts the model performance the most. Character-level perturbations show a stronger influence than word-level and sentence-level perturbations, which conform to the observation for image-text retrieval. Note that for visual reasoning, the most influential text perturbations are different across the different models, but they all belong to the character-level perturbation category. *Glass blur* is the "softest" image perturbation for visual reasoning and *brightness* for visual entailment. Regarding text perturbations, *insert punctuation* and sentence-level perturbations like *formal* and *active* have the least impact on the model's performance for both tasks.

Interestingly, when comparing the robustness of the different models, we make the following observation. Despite TCL is closely related to ALBEF, its robustness performance in terms of MMI score is significantly better. The major difference between both models is that TCL incorporates an intra-modal contrastive loss objective on top of ALBEF, which enforces the learned representations to be semantic meaningful. Additionally to our findings, it has been previously shown that this strategy is also useful in mitigating the noise in training data (Yang et al., 2022). Building on these observations, we recommend that we should consider both intra-modal and cross-modal relations in multimodal representation learning to improve the robustness.

**Image captioning**   In this section, we present the image captioning results of BLIP (Li et al., 2022a) and GRIT (Nguyen et al., 2022) under image perturbations. We present the common evaluation metric Bleu_4 and CIDEr in Figure 5 and leave other metrics and more results with LLaVa (Liu et al., 2023), Mini-GPT4 (Zhu et al., 2023), BLIP2 (Li et al., 2023) to Appendix Sec. F. As shown in Figure 5, *zoom blur* consistently has the most considerable impact across all perturbations on both models. On the other hand, both models are least sensitive to *glass*

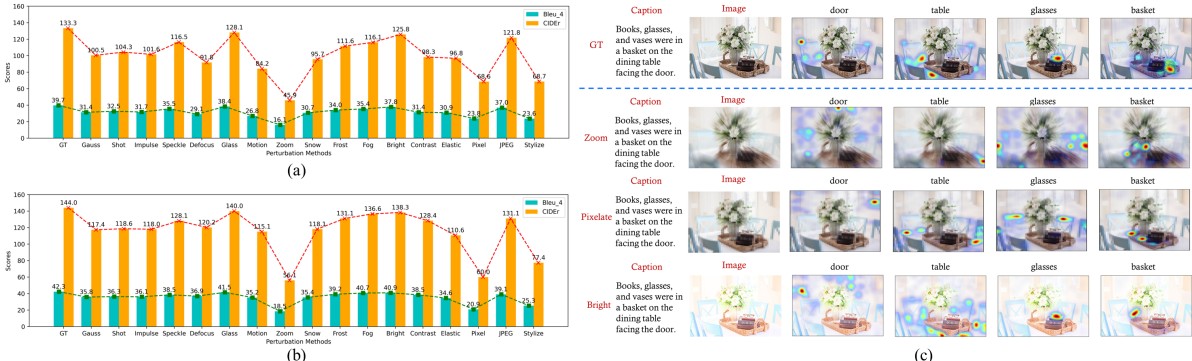

Figure 5: (a) Image captioning results of BLIP; (b) Image captioning results of GRIT; (c) Grad-CAM visualizations on the cross-attention maps corresponding to individual words under image perturbations, where *zoom blur* and *pixelate* perturbed images show worse word-image attention alignment than the *brightness* perturbed image. For example, in *zoom blur* and *pixelate*, the "door" and "glasses" words' attention maps are not matched with the correct image patches, while in *pixelate*, all words' attention maps match correctly.

*blur*, *brightness*, and *JPEG compression*. In addition, we find that across all considered six evaluation metrics, the CIDEr scores are most sensitive to the perturbations, which suggests it is an informative metric for robustness evaluation.

We provide further insights into the effect of the perturbations by inspecting the Grad-CAM (Selvaraju et al., 2017) visualization of BLIP in Figure 5 (c). Given an image, we expect that a robust model is able to attend to different objects according to the word query. Confirming the results shown in the bar plots of Figure 5, we find that "hardest" perturbations, including *zoom blur* and *pixelate* distract the attention of the model the most. For instance, BLIP cannot localize the table or the glasses in the perturbed images. However, for "soft" perturbations like brightness, BLIP is able to provide reasonable localization.

**Text-to-image generation** We present a robustness evaluation for text-to-image generation using two popular generative models, Stable Diffusion (Rombach et al., 2022) and GLIDE (Nichol et al., 2022), under text perturbations. Due to limited space, we only show results and the analysis for Stable Diffusion here and present the results for GLIDE in Appendix Sec. G. Since diversity is essential in text-to-image generation, we generate multiple images given one text for a proper analysis. To assess the diversity, we provide three evaluation settings, where each caption in the dataset is used to generate 4, 8, and 16 images. We adopt the common FID (Heusel et al., 2017) score and CLIP-FID (Kynkaanniemi et al., 2022; Parmar et al., 2022) score as evaluation metrics and report the mean and standard deviation.

As shown in Figure 6 (a) and (b), we surprisingly find that even for the generation task, character-level perturbations affect the robustness of the models the most compared to word-level and sentence-level perturbations. Furthermore, generating more images reduces the variance under each perturbation (e.g., comparing the green against the blue bars). Additionally, we perform a t-test on the generated images and find them to be not correlated after perturbation according to the p-value. This indicates that most text perturbations have an influence on text-to-image generation. Our finding is also corroborated by recent prompt engineering work, where well-designed prompt components can produce coherent outputs (Liu and Chilton, 2022).

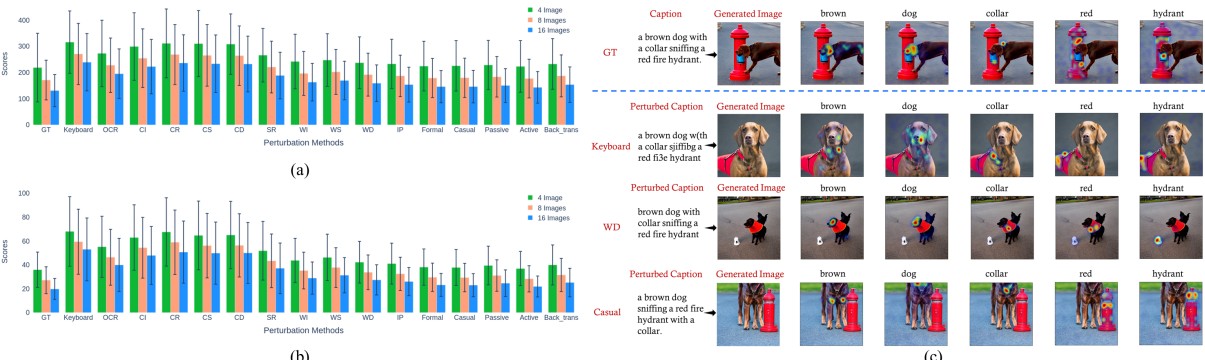

Figure 6: (a) Text-to-image generation results of Stable-diffusion in terms of (a) FID scores; (b) CLIP-FID scores. Since both scores are the lower the better, a higher bar indicates the model is less robust to a particular perturbation. (c) Grad-CAM visualizations on the cross-attention maps corresponding to perturbed captions and images generated by perturbed captions. We use the original unperturbed word query to visualize the attention map. In *keyboard*, the hydrant is missing; in *word deletion*, the color of the hydrant is incorrect, but no object is missing; in *casual*, the attention map perfectly matches the generated images, which shows character-level perturbations could be more effective than word level and sentence-level perturbations.

Lastly, we also provide a further inspection of Stable Diffusion by Grad-CAM visualization in Figure 6 (c). We use the original unperturbed word query to visualize the attention map. *Keyboard*, *word deletion*, and *casual* are shown as character-level, word-level, and sentence-level perturbation examples, respectively. In *keyboard*, the hydrant is missing; in *word deletion*, the color of the hydrant is incorrect, but no object is missing; in *casual*, the attention map perfectly matches the generated images, which shows character-level perturbations could be more effective than word level and sentence-level perturbations. As the *word deletion* in Figure 6 (c), we found Stable Diffusion does not explicitly bind attributes to objects and the reconstructions from the model often mix up attributes and objects, similar to (Ramesh et al., 2022).

**Missing Object Rate (MOR)** To further provide a quantitative evaluation of the quality of the generated images, we propose a new detection-based metric to capture if the model can faithfully generate images with all the objects mentioned in the text. To achieve this goal, we leverage an open-set zero-shot language-guided object detection model, i.e., GLIP (Li et al., 2021c), to detect salient objects in the generated images. As shown in Figure 7 left, the inputs to the GLIP model are text prompt and the generated images from text-to-image generation models. Given COCO is an object detection dataset, and it has ground truth labels for the objects, we can simply use the combination of object names from the ground truth labels as the text prompt, i.e., "dog, cake, broccoli", If the ground truth object can be detected (with a detection threshold $\alpha$), we assume the object is successfully generated by the text-to-image generation model, otherwise, the object is classified as missing.

In Figure 7 right, we show a visual comparison of how perturbed captions can affect the generation quality with respect to missing objects. We first use GT captions and perturbed captions to generate some images, and then perform object detection using GLIP on these images. Note that for all generated images, we always use the same ground truth COCO object names as text prompts. On the top row, we can find that the prompt "cat, pillow, desk" can be detected successfully, which means they are faithfully generated by the Stable Diffusion model. However,

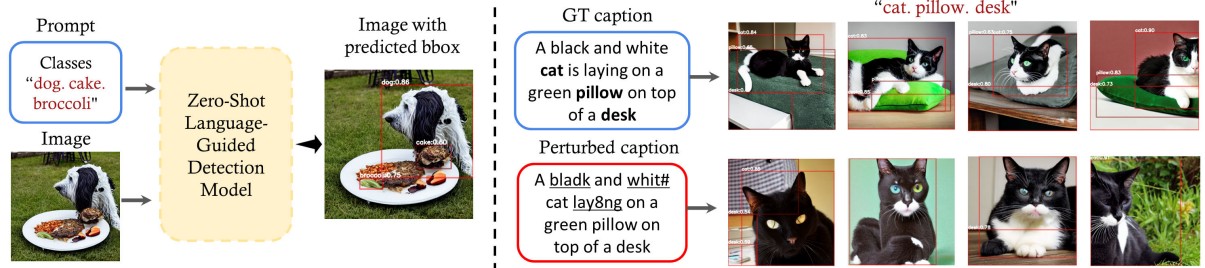

Figure 7: Left: Missing Object Rate (MOR) metric calculation. Right: Comparison of detection results between GT-caption-generated images (top) and perturbed-caption-generated images (bottom).

for the bottom row, the perturbed prompt (*CR* in this example), some objects can not be detected and are considered as missing, i.e., pillow and desk.

Hence, similar to mean corruption error (mCE) in Xie et al. (2019a), we define our detection-based score, termed Missing Object Rate (MOR), as $\mathtt{MOR} = (N_P - N_{GT})/N_{GT}$. Here $N_P$ is the number of detected objects from images generated by perturbed captions, and $N_{GT}$ is the number of detected objects from images generated by GT captions. A lower score indicates more objects are missing, which suggests the perturbed text has a high impact on the underlying text-to-image generation model. As shown in Table 4, we can clearly see that MOR drops significantly for images generated by character-level perturbed captions compared to word-level and sentence-level methods.

Table 4: Quantitative results of Missing Object Rate (MOR) of Stable Diffusion. The most effective perturbation results are marked in bold, and the least effective ones are underlined. The results show that more objects are missing from the images generated by character-level perturbed captions.

| Threshold | Setting | GT | Keyboard | Ocr | CI | CR | CS | CD | SR | RI | RS | RD | IP | Formal | Casual | Passive | Active | Back_trans |
|---|---|---|---|---|---|---|---|---|---|---|---|---|---|---|---|---|---|---|
| | 4-images | 0.00 | -12.47 | -5.22 | -8.41 | **-13.25** | -12.15 | -12.63 | -8.23 | -3.14 | -7.33 | -6.05 | -2.81 | -2.10 | -1.42 | -1.36 | 0.27 | -0.86 |
| 0.7 | 8-images | 0.00 | -11.00 | -4.27 | -6.62 | **-11.79** | -11.09 | -10.76 | -6.77 | -1.62 | -6.59 | -4.31 | -2.83 | 0.01 | 0.69 | -0.17 | 1.34 | 0.44 |
| | 16-images | 0.00 | -11.53 | -4.29 | -6.96 | **-11.72** | -11.59 | -10.86 | -6.88 | -1.65 | -6.66 | -4.48 | -2.90 | -0.16 | 0.17 | -0.75 | 0.76 | 0.48 |
| | 4-images | 0.00 | -5.33 | -2.97 | -2.96 | **-6.60** | -3.97 | -2.45 | -1.00 | 0.72 | -1.51 | -4.63 | -1.88 | -0.31 | -2.18 | 2.17 | -0.30 | 0.65 |
| 0.5 | 8-images | 0.00 | -4.94 | -2.28 | -1.18 | **-5.83** | -2.48 | -1.55 | -0.34 | 1.70 | -1.26 | -2.72 | -1.06 | 0.17 | -1.00 | 3.41 | 0.42 | 1.02 |
| | 16-images | 0.00 | -4.95 | -1.76 | -1.65 | **-5.02** | -2.01 | -2.03 | -0.62 | 1.41 | -0.90 | -2.50 | -0.69 | 0.50 | 0.08 | 3.36 | 0.26 | 1.41 |

## 5 Discussion

Reflecting on the results, we are now equipped to address the questions we initially posed:
*(1) How robust are multimodal pretrained image-text models under distribution shift?*
Multimodal image-text models are sensitive to distribution shifts caused by image and text perturbations, especially shifts in the image space.
*(2) What is the sensitivity of each model under different perturbation methods?*
The sensitivity of different models under different perturbation methods is different. For example, for the image-text retrieval task, under both image and text perturbations, we can see that BLIP shows the best robustness performance, i.e., the lowest MMI score.
*(3) Which model architecture or loss objectives might be more robust under image or text perturbations?*

We hypothesize that using an encoder-decoder architecture and generative language modeling objective is helpful . Given the recent paradigm shift to using generative loss objectives in pre-training multimodal models, e.g., BLIP (Li et al., 2022a), CoCa (Yu et al., 2022), SimVLM (Wang et al., 2022d), PaLI (Chen et al., 2022), Unified-IO (Lu et al., 2022), OFA (Wang et al., 2022a), we believe this observation could be generalized to other multimodal tasks.

*(4) Are there any particular image/text perturbation methods that can consistently show significant influence?*
For image perturbations, zoom blur consistently shows the highest impact on the model's robustness across 5 tasks, while glass blur and brightness are the least harmful ones. For text, character-level perturbations have a higher impact than word-level and sentence-level perturbations. In particular, keyboard and character replace consistently show high impact, while insert punctuation, formal, and active are the three least effective ones across different settings.

**Are our findings applicable to unimodal models?** Given our findings are consistent on five multimodal vision-language downstream tasks, we further investigate whether our findings still hold for unimodal models under distribution shift. The detailed results can be found in Appendix Sec. I. For image perturbations, we evaluate multiple vision models on ImageNet using the same image perturbation techniques in our multimodal setting. Interestingly, similar as in multimodal models, for unimodal vision models, *zoom blur* also has the highest impact on the model performance. For text perturbations, we evaluate several language models on IMDB (Maas et al., 2011) and MultiNLI (Williams et al., 2018) datasets, which leads to the same conclusions as for multimodal models: character-level perturbations also have more significant impacts than word-level and sentence-level perturbations. These observations can be corroborated by previous robustness studies on language models (Belinkov and Bisk, 2018; Ebrahimi et al., 2018; Liu et al., 2022a). In summary, we find that multimodal models show similar vulnerabilities to image and text perturbations as unimodal models in the corresponding modality.

> Takeaway: Our main findings are as follows.
> (1) Multimodal image-text models are sensitive to distribution shifts caused by image and text perturbations, especially to shifts in the image space.
> (2) For image perturbations, *zoom blur* consistently shows the highest impact on the model's robustness across 5 tasks, while *glass blur* and *brightness* are the least harmful ones.
> (3) For text, character-level perturbations have a higher impact than word-level and sentence-level perturbations. In particular, *keyboard* and *character replace* consistently show high impact, while *insert punctuation*, *formal*, and *active* are the three least effective ones across different settings.

**Limitations and future work** Given that our work is one of the early efforts in this direction, there are several promising future work directions and limitations that can be improved. First, we only adopt synthetic image and text perturbation strategies in our benchmark. Although the proposed text perturbations mimic realistic shifts, an exciting extension of our work will be to analyze real-world distribution shifts (Taori et al., 2020; Wenzel et al., 2022). Second, we select 5 important downstream tasks, but there are more tasks, such as visual question answering and visual grounding, that could be analyzed. In addition, we have introduced the MOR metric to evaluate image generation models, but new evaluation metrics beyond existing ones might be needed for proper robustness evaluation under distribution shifts. Third, our study focuses on evaluating image-text models and highlighting failure points. Building on these insights, it is

important to investigate methods that improve robustness. The next natural research direction is to study data augmentation techniques for multimodal models (Hao et al., 2022), which they have shown to be effective in improving the robustness of unimodal models (Hendrycks et al., 2020b, 2021a; Wenzel et al., 2022). Given the fact that both unimodal and multimodal models are sensitive to image *zoom blur* and character-level text perturbations, it might be a good practice to involve these data augmentations during model pre-training. Fourth, all considered multimodal models are learned from web-collected data, which likely contains multiple biases and stereotypes, e.g., w.r.t. gender, race, occupation, etc. This is particularly harmful when using large language models like GPT-3 (Brown et al., 2020), GPT-4 (OpenAI, 2023), or state-of-the-art text-to-image generation models (Saharia et al., 2022). An important research direction is to study the robustness and fairness of those models in a unified setting.

## 6 Conclusion

In this work, we investigate the robustness of large multimodal image-text models under distribution shifts. We introduce several evaluation benchmarks based on 17 image perturbation and 16 text perturbation strategies. We study 5 important downstream tasks, including image-text retrieval, visual reasoning, visual entailment, image captioning, and text-to-image generation, and evaluate 9 popular image-text models. We hope that our proposed benchmark is valuable for analyzing the robustness of image-text models and that our findings provide inspiration to develop and deploy more robust models for real-world applications.

## 7 Broader Impact Statement

**Positive Societal Consequences:** Our research provides a nuanced understanding of the robustness challenges faced by multimodal image-text models. By identifying weaknesses, we pave the way for the development of more robust AI systems, ensuring their reliability and effectiveness in real-world applications.

**Negative Societal Consequences:** Vulnerabilities identified in multimodal models could be exploited by malicious entities for harmful purposes, including deepfakes and misinformation campaigns. This underscores the urgency of addressing these vulnerabilities to safeguard individuals and communities from potential malicious activities.

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

## Appendix A. Perturbation Strategies

**Image Perturbation Strategies**   The details of all the image perturbation strategies are introduced in Table 5.

**Text Perturbation Strategies**   The details of all the text perturbation strategies are introduced in Table 6.

**Magnitude of Perturbation**   We used the same parameters to control the magnitude of perturbation as Hendrycks and Dietterich (2019), which has been taken as the standard parameters for robustness evaluation for the community. To make a fair comparison and to be consistent with previous robustness investigation works, we used the same set of parameters as in Hendrycks and Dietterich (2019).

**Image Quality Drop after Perturbation**   The quality drop of perturbed images has also been analyzed to make sure the model's performance drop is due to the nature of the perturbation methods, instead of the magnitude of different perturbation methods. To provide quantitative comparison results, we evaluated the perturbed images under 5 severity levels, using SSIM (structural similarity index measure (Wang et al., 2004)) and LPIPS (Perceptual Similarity Metric (Zhang et al., 2018)). The results are shown in Table 8. We can find that using SSIM and LPIPS as evaluation metrics, the image quality drop of all the imager perturbation methods are within the same level across different severities. This proved that the pre-trained models being more sensitive to some image corruptions is due to the nature of the perturbation methods themselves, instead of the quality drop not being at the same level.

**Human Verification**   Though we designed an automatic fidelity checking mechanism to ensure the quality of the perturbed data, it would also be good to have humans verify some images/texts. In our experiments, we recruited 10 volunteers to be involved in this verification study. Each person is given one image-text pair at a time (within the pair, either the image is perturbed or the text is perturbed). The person is asked to decide whether this image and text can be considered as a pair. Each person is asked to verify 5,000 image-text pairs, which are randomly sampled from the perturbed COCO dataset (COCO-IP and COCO-TP in the paper). The results are shown in Table 10. The average correction rate is 99.00%, which shows the perturbed image-text pair still preserved the alignment relationship.

## Appendix B. Experimental Settings

**Evaluation Tasks**   We select five widely adopted downstream tasks for a comprehensive evaluation on the robustness of multimodal image-text models, including image-text retrieval, visual reasoning (VR), visual entailment (VE), image captioning, and text-to-image generation. Image-text retrieval includes two subtasks: (1) retrieve images with given text (Image Retrieval) and (2) retrieve text with given images (Text Retrieval) (Cao et al., 2022; Hao et al., 2022). Visual Reasoning (VR) requires the model to determine whether a textual statement describes a pair of images (Suhr et al., 2017). Visual Entailment (VE) is a visual reasoning task to predict whether the relationship between an image and text is entailment, neutral, or contradictory (Xie et al., 2018, 2019b). Image captioning aims at describing the content of an image in words, resulting in textual captions (Lin et al., 2014). Text-to-image generation task is defined as taking input a natural language description and producing an image matching that description (Mansimov et al., 2016).

**Perturbation Datasets** For each task, we perturb the corresponding datasets i.e., Flickr30K (Young et al., 2014), COCO (Lin et al., 2014) , NLVR2 (Suhr et al., 2017), and SNLI-VE (Xie et al., 2018, 2019b), using the image perturbation (IP) and text perturbation (TP) methods introduced in Section 3 in the paper. This leads to our 8 benchmark datasets: (1) Flickr30K-IP, Flickr30K-TP, COCO-IP, and COCO-TP for image-text retrieval robustness evaluation; (2) NLVR2-IP and NLVR2-TP for visual reasoning robustness evaluation; (3) SNLI-VE-IP and SNLI-VE-TP for visual entailment robustness evaluation; (4) COCO-IP for image captioning evaluation; and (5) COCO-TP for text-to-image generation evaluation.

**Evaluation Models** We select 12 representative large pretrained multimodal models, which have publicly released their pretrained models (we appreciate all the authors for making the models publicly available), including CLIP (Radford et al., 2021), ViLT (Kim et al., 2021), ALBEF (Li et al., 2021a), BLIP (Li et al., 2022a), TCL (Yang et al., 2022), METER (Dou et al., 2021), GRIT (Nguyen et al., 2022), LLaVa (Liu et al., 2023), Mini-GPT4 (Zhu et al., 2023), BLIP2 (Li et al., 2023), GLIDE (Nichol et al., 2022), and Stable Diffusion (Rombach et al., 2022). In order to provide a fair comparison, we adopt the model weights provided by their official repositories for either zero-shot prediction or fine-tuned results. We only perform the tasks of each model that have been studied in its original work, where their reported scores are marked as "clean" or "GT" in our Tables.

**Task-Specific Experimental Settings**

- For image-text retrieval, the Flickr30K dataset contains 1,000 images, and each of them has 5 corresponding captions, while the COCO dataset contains 5,000 images, and each of them also has 5 corresponding captions. We report the RSUM score averaged on five perturbation levels under each perturbation method to reveal the overall performance. More detailed results, including the recall at K (R@K) metric, K = $\{1, 5, 10\}$, can be found in Section C in this supplementary material. For CLIP and TCL, we provide the evaluation results for both zero-shot (ZS) and fine-tuned (FT) settings, while for ALBEF and BLIP, we follow their original settings and report the fine-tuned (FT) results.

- For visual reasoning, the NLVR2 dev set contains 2,018 unique sentences and 6,982 samples, while the test-P set contains 1,995 unique sentences and 6,967 samples. We report the accuracy of both the dev set and test-P set of the NLVR2 dataset under image and text perturbations. We evaluate the robustness of ALBEF, ViLT, TCL, BLIP, and METER.

- For visual entailment, the SNLI-VE val set contains 1,000 images and 6,576 sentences, while the test set contains 1,000 images and 6,592 sentences. We evaluate the accuracy of both the dev set and test set of the SNLI-VE dataset under image and text perturbations. We report the results of ALBEF, TCL, and METER.

- For image captioning, we use the COCO-IP test set as an evaluation set. We adopted standard text evaluation metrics, i.e., BLEU (Papineni et al., 2002), METEOR (Denkowski and Lavie, 2014), ROUGE-L (Lin, 2004), and CIDEr (Vedantam et al., 2015).

- For text-to-image generation, we use the captions from the COCO-TP test set as inputs. The COCO-TP test set contains the captions for 5,000 test images, 5 captions for each image, and we select the first caption of each image as inputs, resulting in 5,000 text inputs. We take the FID (Heusel et al., 2017) and CLIP-FID (Kynkaanniemi et al., 2022; Parmar et al., 2022) scores to evaluate the quality of the generated images. We provide 3 settings, where each caption in the test set is used to generate 4,8,16 images, respectively.

## Appendix C. More Results on Image-Text Retrieval

**Results under Image Perturbations**  Detailed image-text retrieval results under image perturbations of ViLT (FT), CLIP (ZS), CLIP (FT), BLIP, ALBEF (FT), TCL (ZS), and TCL (FT), are shown in Tables 25, 26, 27, 28, 29, 30, 31, respectively.

**Results under Text Perturbations**  Detailed image-text retrieval results under text perturbations of CLIP (ZS), CLIP (FT), BLIP, ALBEF (FT), TCL (ZS), and TCL (FT), are shown in Tables 32, 33, 34, 35, 36, 37, 38, respectively.

**Visualization**  We show the image-text retrieval results: (1) image perturbations: in Figures 22, 23; (2) text perturbations: in Figures 28, 29. In Figures 13, 14, we show more Optimal Transport (OT) alignment visualization between images and text under image perturbations. In Figures 15, 16, we show more Optimal Transport (OT) alignment visualization between text and images under text perturbations.

## Appendix D. More Results on Visual Reasoning

**Results**  In Tables 11, 12, we show the results of the visual reasoning task under image perturbation and text perturbation, respectively.

**Visualization**  We show image perturbation results in Figures 24, 25 and text perturbation results in Figures 30, 31.

## Appendix E. More Results on Visual Entailment

**Results**  In Tables 13, 14, we show the results of the visual entailment task under image perturbation and text perturbation, respectively.

**Visualization**  We show image perturbation results in Figures 26, 27 and text perturbation results in Figures 32, 33.

## Appendix F. More Results on Image Captioning

**Results**  In Table 16, we show the value of image captioning results of BLIP, GRIT, LLaVa, Mini-GPT4, and BLIP2 under image perturbations, which are the results as in Figures 9, 10 in this supplementary material, and Figure 5 in the paper. In Figures 9, 10, we show the full metrics results in the image captioning task by BLIP and GRIT.

**Visualization**  In Figures 11, 12, we show examples of image captioning results under image perturbations by BLIP and GRIT, respectively. In Figures 17, 18, we show more Grad-CAM visualizations on the cross-attention maps under image perturbations.

## Appendix G. More Results on Text-to-Image Generation

**Results**  In Table 18, we show the value of text-to-image generation results of Stable Diffusion under text perturbations, which are the results as in Figure 6 in the paper. In Table 19, we show the value of text-to-image generation results of GLIDE under text perturbations.

**Visualization**   In Figures 19, 20, we show more Grad-CAM visualizations on the cross-attention maps corresponding to individual words under text perturbations. In Figure 21, we show the text-to-image generation comparison on all 16 generated images. We find that though the generated images do not guarantee to perfectly show all the notions described in the captions, the probability of generating matched images by the unperturbed captions is higher than the perturbed captions, especially character-level.

## Appendix H.  Learning-based Distribution Shift

In addition to the synthetic perturbation methods in the paper, we also conducted some learning-based distribution shifts (e.g. adversarial robustness) into evaluation. We followed Zhang et al. (2022) and adopted several adversarial perturbation methods, which are shown in Table 21.

We conducted experiments using the adversarial perturbation methods in Table 21 on the image-text retrieval task, and the results are shown in the tables below. We provide the results of ALBEF and CLIP on the Flickr30K and COCO datasets in Tables 22,23,24. In Table 22, we show the image-text retrieval results by adding adversarial perturbations on image modality only by FGSM (Goodfellow et al., 2014). In Table 23, we show the image-text retrieval results by adding adversarial perturbations on text modality only by BERT-Attack (Li et al., 2020a). In Table 24, we show the image-text retrieval results by adding adversarial perturbations on multi-modality by Fooling VQA (Xu et al., 2017), SSAP (Yang et al., 2021), SSAP-MIM (Dong et al., 2017), SSAP-SI (Lin et al., 2019), and Co-Attack (Zhang et al., 2022).

From the results in Tables 22,23,24, we can find that adversarial perturbations can also have a significant impact on the robustness performance. In particular, image adversarial perturbations show a larger influence on the model's performance than text adversarial perturbations. In addition, combining image and text adversarial perturbations can even lead to a larger performance impact than unimodal adversarial perturbations. As for the multimodal adversarial perturbations, Fooling VQA shows the least performance influence, while Co-Attack shows the highest ability in attacking models.

## Appendix I.  Discussion

**Unimodal Vision Model Robustness**   To evaluate whether the findings in our image perturbations of multimodal models are consistent with unimodal vision models, we conducted experiments on multiple unimodal vision models. The top1 classification accuracy is shown in Tables 15. In the results, we find that *zoom blur* is still very effective in most models, and brightness is the most "soft" image perturbation method, which is consistent with the findings in the multimodal setting.

**Conclusion**   To better present the findings, we show plots on the last page. As shown in Tables 26, 27, 28, 29, 30, 31 and Tables 33, 34, 35, 36, 37, 38, we found that: (1) For image perturbations, performance drop by *zoom blur* is larger than other perturbation methods across 5 tasks, while *glass blur* and *brightness* are the least harmful ones. (2) For text, character-level perturbations are more effective than word-level and sentence-level perturbations. In particular, *keyboard* and *character replace* are the most effective ones, while *insert punctuation*, *formal*, and *active* are the three least effective ones across different settings.

## Appendix J. More Related Work

**Robustness of unimodal vision models** is a longstanding and challenging goal of computer vision (Yin et al., 2019). Stable training, adversarial robustness, out-of-distribution, transfer learning, and many other aspects have been studied by previous works in deep learning era (Zheng et al., 2016; Drenkow et al., 2021; Djolonga et al., 2021; Goyal et al., 2022). Recently, several studies have shown that Vision Transformer (ViT) (Dosovitskiy et al., 2021) tend to be more robust than previous models, e.g., work that studied the robustness against common corruptions and perturbations (Bhojanapalli et al., 2021), robustness for distribution shifts and natural adversarial examples (Paul and Chen, 2022), robustness against different Lp-based adversarial attacks (Mahmood et al., 2021), adversarial examples (Mao et al., 2021), and adaptive attacks (Aldahdooh et al., 2021). Several robustness benchmarks have been proposed, e.g., ImageNet-C and ImageNet-P (Hendrycks and Dietterich, 2019), Stylized-ImageNet (Geirhos et al., 2019), ImageNet-A and ImageNet-O (Hendrycks et al., 2021b), ImageNet-V2 (Recht et al., 2019). Recently, (Wenzel et al., 2022) conducted a large-scale robustness study based on natural distribution shifts. (Gupta et al., 2022) built the GRIT benchmark to evaluate the performance, robustness, and calibration of a vision system across different image tasks.

**Robustness of unimodal language models** under distribution shift or adversarial attack has been explored by many previous works, i.e., Chang et al. (2021); Wang et al. (2022c) provided reviews of how to define, measure and improve robustness of NLP systems, Wang et al. (2020) proposed controlled adversarial text generation to improve robustness, Goel et al. (2021) unified four standard evaluation paradigms, Singh et al. (2021) proposed a search and semantically replace strategy, Dong et al. (2021) studied robustness against word substitutions, Malfa and Kwiatkowska (2022) formalised the concept of semantic robustness, etc. In terms of benchmark, Hendrycks et al. (2020a) systematically examined and measured the out-of-distribution (OOD) generalization for seven NLP datasets. Croce et al. (2020) built a large benchmark and analyzed the impact of robustness on the performance of distribution shifts, calibration, OOD detection, fairness, privacy leakage, smoothness, and transferability. Recently, Moradi and Samwald (2021) presented empirical results achieved with a comprehensive set of non-adversarial perturbation methods for testing the robustness of NLP systems on non-synthetic text. Gui et al. (2021) proposed a multilingual evaluation platform to provide comprehensive robustness analysis. Wang et al. (2021) proposed a benchmark to evaluate the vulnerabilities of modern large-scale language models under adversarial attacks.

Table 5: Image perturbations.

| Category | Perturbation | Description | Severities |
|---|---|---|---|
| Noise | Gaussian Noise | Gaussian noise can appear in low-lighting conditions. | 5 |
| | Shot Noise | Shot noise, also called Poisson noise, is electronic noise caused by the discrete nature of light itself. | 5 |
| | Impulse Noise | Impulse noise is a color analogue of salt-and-pepper noise and can be caused by bit errors. | 5 |
| | Speckle Noise | Speckle noise is the noise added to a pixel that tends to be larger if the original pixel intensity is larger. | 5 |
| Blur | Defocus Blur | Defocus blur occurs when an image is out of focus. | 5 |
| | Frosted Glass Blur | Frosted Glass Blur appears with "frosted glass" windows or panels. | 5 |
| | Motion Blur | Motion blur appears when a camera is moving quickly. | 5 |
| | Zoom Blur | Zoom blur occurs when a camera moves toward an object rapidly. | 5 |
| Weather | Snow | Snow is a visually obstructive form of precipitation. | 5 |
| | Frost | Frost forms when lenses or windows are coated with ice crystals. | 5 |
| | Fog | Fog shrouds objects and is rendered with the diamond-square algorithm. | 5 |
| | Brightness | Brightness varies with daylight intensity. | 5 |
| Digital | Contrast | Contrast can be high or low depending on lighting conditions and the photographed object's color. | 5 |
| | Elastic | Elastic transformations stretch or contract small image regions. | 5 |
| | Pixelate | Pixelation occurs when upsampling a low-resolution image. | 5 |
| | JPEG Compression | JPEG is a lossy image compression format that introduces compression artifacts. | 5 |
| Stylize | Stylize | Stylized data is generated by transferring the style information to the content images by AdaIN style transfer (Huang and Belongie, 2017). | 5 |
| Sum | **17** | — | **85** |

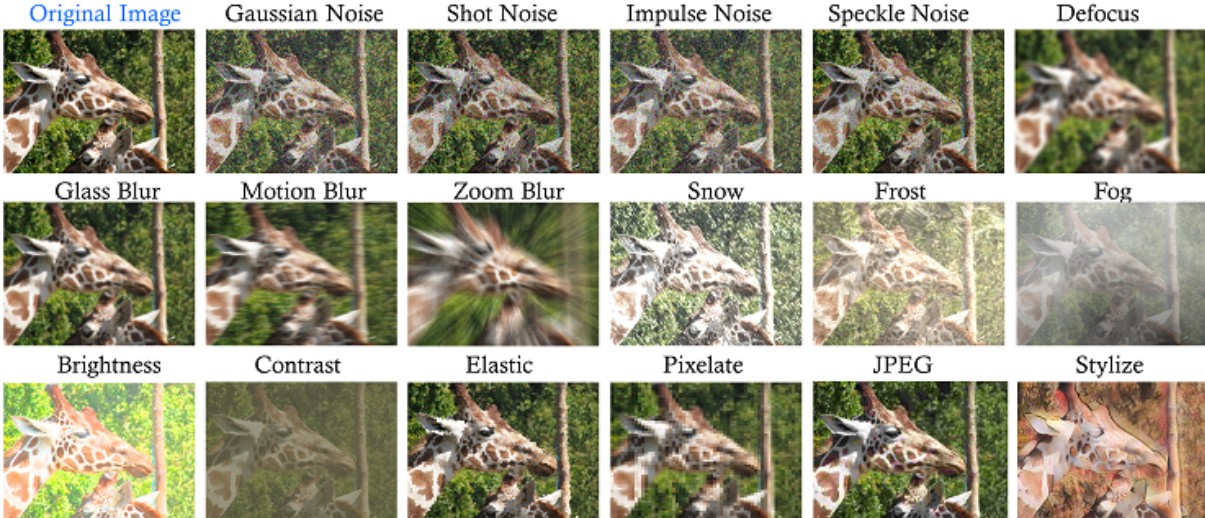

Figure 8: Examples of our 17 image perturbations. The original image is taken from the COCO dataset and shown on the top left.

Table 6: Text perturbations.

| Category | Perturbation | Description | Severities |
|---|---|---|---|
| Character-level | Keyboard | Substitute character by keyboard distance with probability $p$. | 5 |
| | OCR | Substitute character by pre-defined OCR error with probability $p$. | 5 |
| | Character Insert (CI) | Insert character randomly with probability $p$. | 5 |
| | Character Replace (CR) | Substitute character randomly with probability $p$. | 5 |
| | Character Swap (CS) | Swap character randomly with probability $p$. | 5 |
| | Character Delete (CD) | Delete character randomly with probability $p$. | 5 |
| Word-level | Synonym Replacement (SR) | Randomly choose $n$ words from the sentence that are not stop words. Replace each of these words with one of its synonyms chosen at random. | 5 |
| | Word Insertion (WI) | Find a random synonym of a random word in the sentence that is not a stop word. Insert that synonym into a random position in the sentence. Do this $n$ times. | 5 |
| | Word Swap (WS) | Randomly choose two words in the sentence and swap their positions. Do this $n$ times. | 5 |
| | Word Deletion (WD) | Each word in the sentence can be randomly removed with probability $p$. | 5 |
| | Insert Punctuation (IP) | Random insert punctuation in the sentence with probability $p$. | 5 |
| Sentence-level | Formal | Transfer the text style to Formal. | 1 |
| | Casual | Transfer the text style to Casual. | 1 |
| | Passive | Transfer the text style to Passive. | 1 |
| | Active | Transfer the text style to Active. | 1 |
| | Back Translation | Translate source to German and translate it back to English via (Ng et al., 2020). | 1 |
| Sum | **16** | — | **60** |

Table 7: Example of our 16 text perturbations. The original text is taken from the COCO dataset and denoted as clean in the first row.

| Category | Perturbation | Example |
|---|---|---|
| Original | Clean | An orange metal bowl strainer filled with apples. |
| Character | Keyboard | An orange metal bowk strainer filled witj apples. |
| | OCR | An 0range metal bowl strainer filled with app1es. |
| | CI | And orange metal bowl strainer filled with atpples. |
| | CR | An orange metal towl strainer fillet with apples. |
| | CS | An orange meatl bowl stariner filled with apples. |
| | CD | An orang[X] metal bowl strainer fil[X]ed with apples. |
| Word | SR | An orange alloy bowl strainer filled with apples. |
| | WI | An old orange metal bowl strainer filled with apples. |
| | WS | An orange metal strainer bowl filled with apples. |
| | WD | An orange metal bowl strainer [X] with apples. |
| | IP | An orange metal bowl ? strainer filled with apples. |
| Sentence | Formal | An orange metal bowl strainer contains apples. |
| | Casual | An orange metal bowl is filled with apples. |
| | Passive | Some apples are in an orange metal bowl strainer. |
| | Active | There are apples in an orange metal bowl strainer. |
| | Back trans | Apples are placed in an orange metal bowl strainer. |

Table 8: Magnitude of perturbations.

| Method | Parameters |
|---|---|
| Gaussian noise | First normalize the pixel values, then add a random normal noise scaled at values 0.08, 0.12, 0.18, 0.26, 0.38 based on severity |
| Shot noise | Simulate electronic noise caused by the discrete nature of light by applying a combination of salt and pepper noise with amounts ranging from 0.03, 0.06, 0.09, 0.17, 0.27 |
| Impulse noise | Simulate corruptions caused by bit errors by applying a combination of salt and pepper noise with amounts ranging from 0.03, 0.06, 0.09, 0.17, 0.27 |
| Speckle noise | Simulate additive noise and is similar to Gaussian but where the random value is then multiplied by the normalized pixel value |
| Defocus blur | Imitate a defocused lens over the entire frame, ranging from (3, 0.1), (4, 0.5), (6, 0.5), (8, 0.5), (10, 0.5) |
| Motion blur | Increase the radius and sigma of the kernel, ranging from (10, 3), (15, 5), (15, 8), (15, 12), and (20, 15) |
| Zoom blur | Increase the zoom factor based on severity, ranging from (1, 1.11), (1, 1.16), (1, 1.21), (1, 1.26), (1, 1.33) |
| Glass Blur | Appear with "frosted glass" windows or panels, ranging from (0.7, 1, 2), (0.9, 2, 1), (1, 2, 3), (1.1, 3, 2), (1.5, 4, 2) |
| Snow | Adding a visually obstructive form of precipitation, ranging from (0.1, 0.3, 3, 0.5, 10, 4, 0.8),(0.2, 0.3, 2, 0.5, 12, 4, 0.7), (0.55, 0.3, 4, 0.9, 12, 8, 0.7), (0.55, 0.3, 4.5, 0.85, 12, 8, 0.65), (0.55, 0.3, 2.5, 0.85, 12, 12, 0.55) |
| Frost | Simulate lenses or windows are coated with ice crystals, ranging from (1, 0.4), (0.8, 0.6), (0.7, 0.7), (0.65, 0.7),(0.6, 0.75) |
| Fog | Shroud objects and rendered with the diamond-square algorithm, ranging from (1.5, 2), (2, 2), (2.5, 1.7), (2.5, 1.5), (3, 1.4) |
| Brightness | Simulate daylight intensity, ranging from 0.1, 0.2, 0.3, 0.4, 0.5 |
| Contrast | Simulate lighting conditions, ranging from 0.4, 0.3, 0.2, 0.1, 0.05 |
| Elastic | Stretch or contract small image regions, ranging from (244 * 2, 244 * 0.7, 244 * 0.1), (244 * 2, 244 * 0.08, 244 * 0.2), (244 * 0.05, 244 * 0.01, 244 * 0.02), (244 * 0.07, 244 * 0.01, 244 * 0.02), (244 * 0.12, 244 * 0.01, 244 * 0.02) |
| Pixelate | Upsample a low-resolution image, ranging from 0.6, 0.5, 0.4, 0.3, 0.25 |
| JPEG Compression | Convert each frame to a JPEG with quality ranging from 25, 18, 15, 10, 7 |

Table 9: Image Quality Drop after Perturbation.

| SSIM / LPIPS | Clean | Gauss. | Shot | Impulse | Speckle | Defocus | Glass | Motion | Zoom |
|---|---|---|---|---|---|---|---|---|---|
| 1 | 1.00/0.00 | 0.61/0.26 | 0.65/0.25 | 0.58/0.37 | 0.72/0.20 | 0.65/0.30 | 0.78/0.25 | 0.70/0.24 | 0.68/0.20 |
| 2 | 1.00/0.00 | 0.49/0.42 | 0.52/0.42 | 0.45/0.55 | 0.66/0.28 | 0.59/0.49 | 0.73/0.34 | 0.59/0.33 | 0.57/0.36 |
| 3 | 1.00/0.00 | 0.37/0.63 | 0.41/0.60 | 0.37/0.68 | 0.51/0.51 | 0.50/0.61 | 0.58/0.47 | 0.51/0.41 | 0.50/0.44 |
| 4 | 1.00/0.00 | 0.27/0.86 | 0.29/0.85 | 0.27/0.89 | 0.34/0.63 | 0.35/0.68 | 0.48/0.60 | 0.46/0.58 | 0.47/0.58 |
| 5 | 1.00/0.00 | 0.19/0.99 | 0.23/0.99 | 0.19/0.99 | 0.27/0.76 | 0.32/0.72 | 0.32/0.70 | 0.33/0.61 | 0.36/0.69 |

| SSIM / LPIPS | Snow | Frost | Fog | Bright | Contrast | Elastic | Pixel | JPEG | Stylize |
|---|---|---|---|---|---|---|---|---|---|
| 1 | 0.66/0.24 | 0.68/0.20 | 0.63/0.17 | 0.79/0.12 | 0.67/0.17 | 0.66/0.26 | 0.72/0.22 | 0.78/0.17 | 0.72/0.21 |
| 2 | 0.48/0.37 | 0.58/0.28 | 0.57/0.32 | 0.70/0.21 | 0.58/0.25 | 0.53/0.29 | 0.57/0.35 | 0.67/0.23 | 0.58/0.34 |
| 3 | 0.52/0.45 | 0.53/0.34 | 0.53/0.37 | 0.65/0.31 | 0.49/0.38 | 0.42/0.39 | 0.53/0.43 | 0.65/0.35 | 0.47/0.43 |
| 4 | 0.46/0.51 | 0.53/0.44 | 0.52/0.52 | 0.53/0.48 | 0.39/0.58 | 0.41/0.52 | 0.47/0.52 | 0.60/0.46 | 0.37/0.52 |
| 5 | 0.32/0.63 | 0.40/0.52 | 0.38/0.64 | 0.45/0.64 | 0.33/0.73 | 0.39/0.78 | 0.37/0.63 | 0.45/0.68 | 0.31/0.67 |

Table 10: Human verification of perturbed image-text pairs, where the correction rate means the percentage of given image and text can still be considered as a pair.

| Correction Rate | Judge-1 | Judge-2 | Judge-3 | Judge-4 | Judge-5 | Judge-6 | Judge-7 | Judge-8 | Judge-9 | Judge-10 | Average |
|---|---|---|---|---|---|---|---|---|---|---|---|
| Results | 98.90% | 99.42% | 98.80% | 98.54% | 99.14% | 98.50% | 99.02% | 99.26% | 99.16% | 99.30% | 99.00% |

Table 11: **Visual reasoning:** *image robustness* evaluations for the *NLVR2-IP* dataset (averaged accuracy), where the most effective perturbation results are marked in bold and the least effective ones are underlined. Impact score is marked in blue, the lower the better.

| | | | Noise | | | | Blur | | | | Weather | | | | Digital | | | | Stylize | | |
|---|---|---|---|---|---|---|---|---|---|---|---|---|---|---|---|---|---|---|---|---|---|
| Dataset | Method | Clean | Gauss. | Shot | Impulse | Speckle | Defocus | Glass | Motion | Zoom | Snow | Frost | Fog | Bright | Contrast | Elastic | Pixel | JPEG | Stylize | ave | MMI |
| dev | ALBEF | 82.55 | 52.80 | 52.46 | 52.61 | 52.63 | 52.22 | 52.44 | 51.78 | 50.79 | 50.69 | 52.05 | 52.58 | 52.09 | 51.98 | 52.45 | 50.99 | 52.37 | 51.80 | 52.04 | ↓ 37.0% |
| | ViLT | 75.70 | 71.64 | 71.45 | 71.58 | 72.42 | 72.90 | 74.71 | 68.79 | 63.97 | 69.40 | 73.02 | 73.59 | 74.32 | 66.72 | 74.15 | 69.17 | 74.71 | 72.35 | 71.46 | ↓ 5.6% |
| | TCL | 80.54 | 78.20 | 77.63 | 78.21 | 78.60 | 77.04 | 81.20 | 77.37 | 66.67 | 75.96 | 79.47 | 79.65 | 80.76 | 74.04 | 78.92 | 73.92 | 81.01 | 75.05 | 77.28 | ↓ 4.0% |
| | BLIP | 82.48 | 85.37 | 78.54 | 72.68 | 76.59 | 80.00 | 73.66 | 78.54 | 60.98 | 73.66 | 76.59 | 83.90 | 76.10 | 77.07 | 81.46 | 74.63 | 82.93 | 71.71 | 77.42 | ↓ 6.1% |
| | METER | 82.33 | 77.39 | 76.25 | 77.25 | 77.76 | 78.76 | 82.01 | 78.26 | 69.31 | 76.17 | 79.40 | 81.02 | 80.76 | 77.50 | 79.36 | 72.91 | 80.67 | 76.10 | 77.70 | ↓ 5.6% |
| test-P | ALBEF | 83.14 | 53.17 | 52.85 | 53.22 | 53.50 | 52.68 | 53.09 | 52.39 | 51.19 | 51.60 | 52.98 | 53.49 | 52.78 | 53.13 | 53.12 | 51.72 | 53.10 | 52.95 | 52.76 | ↓ 36.5% |
| | ViLT | 76.13 | 74.24 | 73.80 | 74.43 | 74.20 | 72.32 | 76.70 | 72.55 | 62.34 | 69.24 | 73.36 | 75.05 | 74.73 | 68.68 | 74.07 | 69.06 | 76.52 | 71.50 | 72.54 | ↓ 4.7% |
| | TCL | 81.33 | 78.10 | 77.87 | 78.25 | 78.91 | 78.00 | 81.59 | 78.17 | 67.81 | 75.74 | 79.62 | 80.64 | 81.52 | 74.35 | 79.76 | 74.61 | 81.28 | 75.85 | 77.77 | ↓ 4.4% |
| | BLIP | 83.08 | 75.39 | 75.39 | 85.10 | 72.31 | 85.64 | 79.49 | 76.92 | 58.97 | 80.51 | 75.90 | 81.54 | 76.92 | 81.03 | 77.95 | 73.333 | 78.97 | 73.85 | 77.01 | ↓ 7.3% |
| | METER | 83.05 | 78.87 | 77.94 | 77.78 | 79.23 | 78.97 | 82.10 | 79.14 | 68.89 | 76.69 | 80.10 | 82.25 | 81.21 | 78.20 | 79.91 | 72.65 | 80.74 | 76.93 | 78.34 | ↓ 5.7% |

Table 12: **Visual reasoning:** *text robustness* evaluations for the *NLVR2-TP* dataset (averaged accuracy), where the most effective perturbation results are marked in bold and the least effective ones are underlined. Impact score is marked in blue, the lower the better.

| | | | Character-level | | | | | | Word-level | | | | | Sentence-level | | | | | |
|---|---|---|---|---|---|---|---|---|---|---|---|---|---|---|---|---|---|---|---|---|
| Dataset | Method | Clean | Keyboard | OCR | CI | CR | CS | CD | SR | WI | WS | WD | IP | Formal | Casual | Passive | Active | Back_trans | ave | MMI |
| dev | ALBEF | 82.55 | 50.64 | 51.02 | 50.81 | 50.66 | 50.53 | 50.58 | 51.96 | 51.48 | 51.58 | 51.39 | 51.56 | 50.99 | 51.93 | 51.52 | 51.75 | 51.90 | 51.22 | ↓ 38.0% |
| | ViLT | 75.70 | 66.23 | 69.16 | 65.47 | 64.36 | 64.76 | 64.96 | 67.11 | 72.71 | 70.77 | 71.75 | 73.42 | 73.22 | 73.40 | 71.83 | 74.47 | 74.51 | 69.88 | ↓ 7.7% |
| | TCL | 80.54 | 71.15 | 75.89 | 71.84 | 70.99 | 72.01 | 71.58 | 74.96 | 78.89 | 77.84 | 78.05 | 82.37 | 81.56 | 80.33 | 79.47 | 81.46 | 80.67 | 71.77 | ↓ 10.9% |
| | BLIP | 82.48 | 70.73 | 70.24 | 76.59 | 74.63 | 72.68 | 72.20 | 73.17 | 77.56 | 80.00 | 79.51 | 87.81 | 85.37 | 82.93 | 82.93 | 87.81 | 75.61 | 78.11 | ↓ 5.3% |
| | METER | 82.33 | 72.35 | 75.83 | 74.10 | 72.71 | 73.89 | 73.30 | 75.16 | 79.36 | 75.41 | 77.64 | 81.68 | 81.92 | 81.55 | 78.69 | 81.01 | 82.25 | 77.30 | ↓ 6.1% |
| test-P | ALBEF | 83.14 | 51.39 | 51.99 | 51.04 | 51.26 | 51.05 | 51.24 | 52.69 | 52.95 | 52.95 | 52.88 | 53.30 | 53.39 | 53.06 | 52.68 | 53.26 | 53.23 | 52.40 | ↓ 37.0% |
| | ViLT | 76.13 | 64.85 | 69.66 | 66.76 | 65.64 | 65.56 | 65.14 | 68.96 | 73.36 | 71.35 | 72.53 | 75.14 | 75.86 | 74.27 | 72.58 | 77.00 | 75.70 | 70.90 | ↓ 6.9% |
| | TCL | 81.33 | 71.16 | 76.31 | 72.35 | 71.56 | 71.90 | 72.07 | 75.49 | 80.03 | 78.80 | 78.78 | 82.88 | 82.46 | 81.52 | 80.25 | 82.28 | 81.53 | 72.37 | ↓ 11.0% |
| | BLIP | 83.08 | 67.69 | 85.64 | 67.18 | 67.69 | 75.90 | 74.87 | 69.23 | 72.82 | 78.46 | 83.59 | 83.59 | 79.49 | 87.18 | 82.05 | 82.05 | 74.36 | 76.99 | ↓ 7.3% |
| | METER | 83.05 | 73.10 | 77.63 | 74.05 | 72.49 | 70.64 | 74.27 | 76.10 | 79.62 | 75.96 | 78.55 | 82.58 | 81.87 | 80.42 | 79.52 | 82.34 | 81.45 | 77.54 | ↓ 6.6% |

Table 13: **Visual entailment:** *image robustness* evaluations for the *SNLI-VE-IP* dataset (averaged accuracy), where the most effective perturbation results are marked in bold and the least effective ones are underlined. Impact score is marked in blue, the lower the better.

| | | | Noise | | | | Blur | | | | Weather | | | | Digital | | | | Stylize | | |
|---|---|---|---|---|---|---|---|---|---|---|---|---|---|---|---|---|---|---|---|---|---|
| Dataset | Method | Clean | Gauss. | Shot | Impulse | Speckle | Defocus | Glass | Motion | Zoom | Snow | Frost | Fog | Bright | Contrast | Elastic | Pixel | JPEG | Stylize | ave | MMI |
| val | ALBEF | 80.80 | 77.52 | 77.56 | 77.34 | 78.76 | 76.59 | 79.26 | 76.67 | **71.70** | 75.61 | 78.71 | 78.76 | 79.83 | 78.19 | 78.49 | 74.29 | 78.91 | 74.58 | 77.22 | ↓4.4% |
| | TCL | 80.51 | 77.33 | 77.56 | 77.22 | 78.23 | 76.70 | 79.21 | 75.25 | **70.98** | 75.71 | 77.95 | 78.43 | 79.31 | 78.76 | 77.78 | 71.47 | 78.43 | 74.64 | 76.76 | ↓4.7% |
| | METER | 80.86 | 77.05 | 77.19 | 76.76 | 78.37 | 77.14 | 79.72 | 77.04 | **74.35** | 77.18 | 79.38 | 80.10 | 80.49 | 79.12 | 78.78 | 73.08 | 78.93 | 75.88 | 77.68 | ↓3.9% |
| test | ALBEF | 80.91 | 77.65 | 77.70 | 77.40 | 78.50 | 76.62 | 79.25 | 76.59 | **71.70** | 76.31 | 78.60 | 78.47 | 79.77 | 78.07 | 78.34 | 74.42 | 78.81 | 74.89 | 77.24 | ↓8.3% |
| | TCL | 80.29 | 77.46 | 77.38 | 77.30 | 78.17 | 76.80 | 79.27 | 75.56 | **71.07** | 76.13 | 78.24 | 78.38 | 79.19 | 78.68 | 77.74 | 71.76 | 78.59 | 74.70 | 76.85 | ↓4.3% |
| | METER | 81.19 | 77.16 | 77.09 | 76.90 | 78.58 | 77.14 | 80.13 | 77.39 | **74.35** | 77.79 | 79.84 | 80.18 | 80.46 | 79.18 | 78.91 | 72.67 | 79.32 | 76.08 | 77.79 | ↓4.2% |

Table 14: **Visual entailment:** *text robustness* evaluations for the *SNLI-VE-TP* dataset (averaged accuracy), where the most effective perturbation results are marked in bold and the least effective ones are underlined. Impact score is marked in blue, the lower the better.

| | | | Character-level | | | | | | Word-level | | | | | Sentence-level | | | | | | |
|---|---|---|---|---|---|---|---|---|---|---|---|---|---|---|---|---|---|---|---|---|---|
| Dataset | Method | Clean | Keyboard | OCR | CI | CR | CS | CD | SR | WI | WS | WD | IP | Formal | Casual | Passive | Active | Back_trans | ave | MMI |
| val | ALBEF | 80.80 | 65.35 | 71.97 | 66.54 | **65.17** | 67.22 | 67.46 | 74.63 | 74.15 | 74.88 | 78.62 | 80.56 | 80.56 | 80.56 | 80.56 | 80.56 | 76.94 | 74.11 | ↓8.3% |
| | TCL | 80.51 | 65.24 | 71.63 | 65.58 | **64.72** | 67.67 | 67.16 | 74.32 | 74.04 | 74.52 | 77.84 | 79.84 | 79.84 | 79.84 | 79.84 | 79.84 | 75.79 | 73.61 | ↓8.6% |
| | METER | 80.86 | **66.70** | 74.17 | 67.99 | 66.41 | 68.64 | 69.53 | 74.65 | 73.19 | 72.55 | 78.28 | 76.24 | 80.72 | 80.49 | 80.76 | 80.72 | 77.43 | 74.28 | ↓8.1% |
| test | ALBEF | 80.91 | **64.87** | 71.90 | 65.99 | 65.03 | 66.91 | 67.27 | 74.77 | 74.93 | 74.90 | 78.44 | 80.20 | 80.20 | 80.20 | 80.20 | 80.20 | 77.31 | 73.96 | ↓8.6% |
| | TCL | 80.29 | **65.27** | 71.83 | 65.81 | 64.66 | 67.69 | 67.25 | 74.59 | 73.70 | 74.49 | 78.01 | 79.77 | 79.77 | 79.77 | 79.84 | 79.84 | 76.62 | 73.67 | ↓8.2% |
| | METER | 81.19 | **66.09** | 74.26 | 67.39 | 66.30 | 68.92 | 69.71 | 74.88 | 73.89 | 72.95 | 78.38 | 76.65 | 80.96 | 80.83 | 81.21 | 81.05 | 77.14 | 74.41 | ↓8.4% |

Table 15: Top1 classification accuracy of unimodal vision models. The most effective perturbation results are marked in bold and the least effective ones are underlined.

| Model/corruption | bright | contrast | defocus | elastic | fog | glass | gauss | impulse | jpeg | motion | pixelate | saturate | shot | snow | spatter | speckle | zoom |
|---|---|---|---|---|---|---|---|---|---|---|---|---|---|---|---|---|---|
| deit_base_distilled | 0.81 | 0.81 | 0.57 | 0.64 | 0.79 | 0.59 | 0.67 | 0.66 | 0.69 | 0.64 | 0.66 | 0.80 | 0.66 | 0.67 | 0.74 | 0.72 | **0.54** |
| densenet169 | 0.72 | 0.61 | 0.41 | 0.45 | 0.60 | 0.41 | 0.42 | **0.37** | 0.57 | 0.41 | 0.52 | 0.69 | 0.41 | 0.42 | 0.52 | 0.47 | 0.38 |
| eca_nfnet_l0 | 0.79 | 0.77 | 0.47 | 0.52 | 0.69 | 0.50 | 0.40 | 0.44 | 0.65 | 0.59 | 0.48 | 0.78 | **0.39** | 0.62 | 0.72 | 0.55 | 0.51 |
| efficientnetv2 | 0.79 | 0.72 | 0.51 | 0.56 | 0.62 | 0.53 | **0.46** | 0.49 | 0.68 | 0.60 | 0.60 | 0.78 | **0.46** | 0.62 | 0.72 | 0.60 | 0.54 |
| gmlp_s16_224 | 0.71 | 0.72 | 0.42 | 0.57 | 0.66 | 0.46 | 0.55 | 0.53 | 0.59 | 0.52 | 0.58 | 0.70 | 0.54 | 0.34 | 0.60 | 0.61 | **0.39** |
| mixer_b16_224 | 0.71 | 0.72 | 0.31 | 0.44 | 0.62 | 0.35 | 0.31 | **0.26** | 0.44 | 0.41 | 0.48 | 0.63 | 0.29 | 0.35 | 0.50 | 0.38 | **0.28** |
| mobilenetv3_large | 0.71 | 0.46 | 0.35 | 0.47 | 0.51 | 0.38 | **0.33** | 0.35 | 0.56 | 0.47 | 0.44 | 0.68 | 0.33 | 0.38 | 0.55 | 0.45 | 0.38 |
| pit_s_224 | 0.79 | 0.77 | 0.50 | 0.56 | 0.72 | 0.51 | 0.64 | 0.62 | 0.67 | 0.58 | 0.57 | 0.77 | 0.62 | 0.62 | 0.70 | 0.68 | **0.46** |
| regnety_064 | 0.75 | 0.54 | 0.45 | 0.51 | 0.61 | 0.46 | 0.43 | **0.41** | 0.59 | 0.48 | 0.46 | 0.71 | 0.40 | 0.46 | 0.62 | 0.48 | 0.44 |
| resmlp_24_224 | 0.76 | 0.73 | 0.48 | 0.57 | 0.61 | 0.51 | 0.56 | 0.54 | 0.60 | 0.56 | 0.54 | 0.75 | 0.54 | 0.56 | 0.64 | 0.62 | **0.46** |
| resnet50d | 0.78 | 0.77 | 0.44 | 0.46 | 0.71 | 0.47 | 0.41 | **0.39** | 0.63 | 0.50 | 0.40 | 0.76 | 0.41 | 0.51 | 0.63 | 0.53 | 0.46 |
| resnext101_32x8d | 0.74 | 0.53 | 0.48 | 0.52 | 0.60 | 0.47 | 0.42 | **0.37** | 0.61 | 0.52 | 0.56 | 0.69 | 0.40 | 0.42 | 0.57 | 0.49 | 0.49 |
| swin_small_patch4 | 0.81 | 0.81 | 0.52 | 0.57 | 0.75 | **0.52** | 0.63 | 0.63 | 0.57 | 0.61 | 0.40 | 0.80 | 0.62 | 0.65 | 0.77 | 0.70 | **0.52** |
| vit_small_patch16 | 0.62 | 0.73 | 0.45 | 0.50 | 0.67 | 0.47 | 0.34 | 0.31 | 0.55 | 0.52 | 0.58 | 0.56 | 0.31 | **0.26** | 0.53 | 0.41 | 0.38 |

Table 16: Detailed image captioning results of BLIP and GRIT.

| | | | | Noise | | | | Blur | | | | Weather | | | | Digital | | | | Stylize | | |
|---|---|---|---|---|---|---|---|---|---|---|---|---|---|---|---|---|---|---|---|---|---|---|
| | | GT | Gauss | Shot | Impulse | Speckle | Defocus | Glass | Motion | Zoom | Snow | Frost | Fog | Bright | Contrast | Elastic | Pixel | JPEG | Stylize | Ave | MMI |
| BLIP | Bleu_1 | 78.9 | 70.9 | 71.9 | 71.1 | 74.8 | 68.5 | 77.5 | 66.5 | **55.9** | 70.6 | 73.4 | 75.0 | 77.3 | 71.3 | 69.9 | 62.7 | 76.2 | 63.7 | 66.5 | ↓15.7% |
| | Bleu_2 | 63.8 | 54.4 | 55.7 | 54.8 | 59.0 | 52.3 | 62.3 | 50.0 | **37.5** | 53.8 | 57.5 | 59.3 | 61.8 | 54.9 | 53.4 | 45.4 | 60.6 | 46.2 | 51.0 | ↓20.0% |
| | Bleu_3 | 50.5 | 41.3 | 42.6 | 41.7 | 45.8 | 39.1 | 49.1 | 36.7 | **24.4** | 40.5 | 44.3 | 45.9 | 48.4 | 41.6 | 40.6 | 32.6 | 47.4 | 32.9 | 38.6 | ↓23.6% |
| | Bleu_4 | 39.7 | 31.4 | 32.5 | 31.7 | 35.5 | 29.1 | 38.4 | 26.8 | **16.1** | 30.7 | 34.0 | 35.4 | 37.8 | 31.4 | 30.9 | 23.8 | 37.0 | 23.6 | 29.2 | ↓26.4% |
| | Meteor | 31.0 | 26.1 | 26.8 | 26.4 | 28.5 | 24.7 | 30.1 | 23.6 | **17.0** | 25.7 | 27.7 | 28.8 | 29.8 | 25.7 | 25.7 | 21.3 | 29.3 | 21.1 | 24.4 | ↓21.5% |
| | Rouge_L | 60.0 | 53.3 | 54.3 | 53.7 | 56.7 | 50.9 | 58.8 | 49.3 | **40.5** | 52.4 | 55.6 | 57.0 | 58.6 | 53.2 | 52.7 | 46.8 | 57.8 | 46.8 | 49.9 | ↓16.8% |
| | CIDEr | 133.3 | 100.5 | 104.3 | 101.6 | 116.5 | 91.8 | 128.1 | 84.2 | **45.9** | 95.7 | 111.6 | 116.1 | 125.8 | 98.3 | 96.8 | 68.6 | 121.8 | 68.7 | 93.1 | ↓30.1% |
| GRIT | Bleu_1 | 84.2 | 78.6 | 79.1 | 78.8 | 81.1 | 79.4 | 83.6 | 77.9 | **60.0** | 78.6 | 81.8 | 83.1 | 83.1 | 81.4 | 77.4 | 64.0 | 81.6 | 68.9 | 73.2 | ↓13.0% |
| | Bleu_2 | 69.1 | 62.2 | 62.6 | 62.4 | 65.0 | 63.1 | 68.4 | 61.3 | **40.5** | 61.8 | 65.9 | 67.5 | 67.7 | 65.3 | 60.5 | 44.3 | 65.8 | 50.0 | 57.4 | ↓16.8% |
| | Bleu_3 | 54.7 | 47.6 | 48.1 | 47.9 | 50.5 | 48.7 | 53.9 | 46.8 | **27.1** | 47.2 | 51.4 | 53.0 | 53.2 | 50.7 | 46.1 | 30.3 | 51.2 | 35.5 | 43.8 | ↓19.8% |
| | Bleu_4 | 42.3 | 35.8 | 36.3 | 36.1 | 38.5 | 36.9 | 41.5 | 35.2 | **18.5** | 35.4 | 39.2 | 40.7 | 40.9 | 38.5 | 34.6 | 20.9 | 39.1 | 25.3 | 33.0 | ↓22.0% |
| | Meteor | 30.6 | 27.0 | 27.2 | 27.1 | 28.4 | 27.5 | 30.1 | 26.7 | **17.7** | 27.0 | 28.8 | 29.6 | 29.9 | 28.5 | 26.2 | 18.7 | 28.9 | 21.2 | 25.0 | ↓18.3% |
| | Rouge_L | 60.7 | 55.8 | 56.2 | 56.0 | 57.8 | 56.6 | 60.1 | 55.4 | **42.6** | 55.6 | 58.3 | 59.4 | 59.8 | 58.0 | 55.0 | 44.5 | 58.4 | 48.2 | 52.1 | ↓14.2% |
| | CIDEr | 144.0 | 117.4 | 118.6 | 118.0 | 128.1 | 120.2 | 140.0 | 115.1 | **56.1** | 118.1 | 131.1 | 136.6 | 138.3 | 128.4 | 110.6 | 60.0 | 131.1 | 77.4 | 108.1 | ↓25.0% |

Table 17: Image captioning results of BLIP, GRIT, LLaVa, Mini-GPT4, and BLIP2.

| | GT | Noise | | | | Blur | | | | Weather | | | | Digital | | | | Stylize | | |
| | | Gauss | Shot | Impulse | Speckle | Defocus | Glass | Motion | Zoom | Snow | Frost | Fog | Bright | Contrast | Elastic | Pixel | JPEG | Stylize | Ave | MMI |
|---|---|---|---|---|---|---|---|---|---|---|---|---|---|---|---|---|---|---|---|---|
| BLIP | 60.0 | 53.3 | 54.3 | 53.7 | 56.7 | 50.9 | 58.8 | 49.3 | **40.5** | 52.4 | 55.6 | 57.0 | 58.6 | 53.2 | 52.7 | 46.8 | 57.8 | 46.8 | 49.9 | ↓16.8% |
| GRIT | 60.7 | 55.8 | 56.2 | 56.0 | 57.8 | 56.6 | 60.1 | 55.4 | **42.6** | 55.6 | 58.3 | 59.4 | 58.0 | 58.0 | 62.3 | 44.5 | 58.4 | 48.2 | 52.1 | ↓14.2% |
| LLaVA | 68.6 | 62.5 | 62.3 | 59.8 | 63.2 | 62.7 | 64.9 | 63.6 | 55.3 | 56.1 | 57.7 | 62.5 | 66.2 | 60.1 | 62.3 | 56.9 | 64.2 | 55.8 | 60.9 | ↓11.2% |
| Mini-GPT4 | 71.1 | 66.8 | 66.3 | 62.7 | 67.1 | 66.9 | 65.2 | 66.9 | 60.5 | 60.9 | 61.3 | 67.2 | 68.7 | 65.6 | 67.2 | 61.8 | 68.9 | 62.2 | 65.1 | ↓8.5% |
| BLIP2 | 64.2 | 61.3 | 59.3 | 55.2 | 60.2 | 59.7 | 60.9 | 60.1 | 52.1 | 53.7 | 55.8 | 60.1 | 64.2 | 57.4 | 59.2 | 53.8 | 61.9 | 51.5 | 58 | ↓9.6% |

Table 18: Text-to-image generation results of Stable Diffusion (FID and CLIP-FID), where "GT" means images generated by GT captions.

| | | FID | | | | | | | | | CLIP_FID | | | | | | | | |
| | | 4 image | | | 8 image | | | 16 image | | | 4 image | | | 8 image | | | 16 image | | |
| | | mean | std | MMI | mean | std | MMI | mean | std | MMI | mean | std | MMI | mean | std | MMI | mean | std | MMI |
|---|---|---|---|---|---|---|---|---|---|---|---|---|---|---|---|---|---|---|---|
| Character | Keyboard | 315.9 | 131.10 | ↓44.39% | 270.77 | 116.78 | ↓58.34% | 239.27 | 109.41 | ↓82.80% | 68.05 | 29.13 | ↓89.40% | 59.42 | 27.31 | ↓118.46% | 53.04 | 26.35 | ↓167.74% |
| | Ocr | 272.9 | 119.01 | ↓24.76% | 227.72 | 103.83 | ↓33.16% | 195.25 | 94.93 | ↓49.17% | 55.20 | 25.61 | ↓53.63% | 46.40 | 23.47 | ↓70.59% | 39.95 | 22.28 | ↓101.67% |
| | CI | 299.1 | 126.80 | ↓36.70% | 254.40 | 111.87 | ↓48.76% | 222.62 | 103.82 | ↓70.08% | 62.94 | 27.41 | ↓75.17% | 54.41 | 25.45 | ↓100.04% | 47.93 | 24.29 | ↓141.95% |
| | CR | 311.2 | 129.03 | ↓42.23% | 268.64 | 114.50 | ↓57.09% | 236.27 | 107.73 | ↓80.51% | 67.65 | 28.67 | ↓88.28% | 58.98 | 26.98 | ↓116.84% | 50.74 | 26.06 | ↓156.13% |
| | CS | 310.3 | 131.50 | ↓41.85% | 265.29 | 117.53 | ↓55.13% | 233.46 | 109.85 | ↓78.36% | 64.70 | 28.79 | ↓80.07% | 56.17 | 27.03 | ↓106.51% | 49.88 | 26.08 | ↓151.79% |
| | CD | 308.5 | 125.99 | ↓40.99% | 264.14 | 113.76 | ↓54.46% | 232.46 | 106.40 | ↓77.60% | 65.03 | 28.21 | ↓80.99% | 56.38 | 26.48 | ↓107.28% | 50.04 | 25.60 | ↓152.60% |
| Word | SR | 266.1 | 115.45 | ↓21.62% | 220.86 | 98.86 | ↓29.15% | 188.45 | 88.97 | ↓43.98% | 51.84 | 24.62 | ↓44.28% | 43.43 | 22.49 | ↓59.67% | 37.14 | 21.21 | ↓87.48% |
| | RI | 242.0 | 102.38 | ↓10.60% | 196.42 | 83.79 | ↓14.86% | 163.14 | 71.85 | ↓24.64% | 43.76 | 18.47 | ↓21.79% | 35.28 | 15.42 | ↓29.71% | 28.90 | 13.50 | ↓45.89% |
| | RS | 247.5 | 104.33 | ↓13.15% | 202.32 | 85.65 | ↓18.31% | 169.39 | 73.77 | ↓29.41% | 46.31 | 19.47 | ↓28.89% | 37.78 | 16.65 | ↓38.90% | 31.33 | 14.77 | ↓58.15% |
| | RD | 237.3 | 100.41 | ↓8.44% | 191.81 | 81.89 | ↓12.16% | 158.95 | 69.89 | ↓21.44% | 42.26 | 17.50 | ↓17.62% | 33.80 | 14.56 | ↓24.26% | 27.44 | 12.67 | ↓38.52% |
| | IP | 233.0 | 98.81 | ↓6.49% | 187.02 | 79.24 | ↓9.36% | 153.63 | 66.62 | ↓17.37% | 41.07 | 17.17 | ↓14.31% | 32.50 | 13.93 | ↓19.49% | 26.02 | 11.78 | ↓31.35% |
| Sentence | Formal | 224.4 | 93.92 | ↓2.58% | 178.94 | 74.71 | ↓4.64% | 145.92 | 61.51 | ↓11.48% | 38.15 | 15.20 | ↓6.18% | 29.60 | 11.88 | ↓8.82% | 23.21 | 9.63 | ↓17.16% |
| | Casual | 225.6 | 94.97 | ↓3.14% | 179.66 | 75.11 | ↓5.06% | 146.16 | 61.92 | ↓11.67% | 37.84 | 15.10 | ↓5.32% | 29.40 | 11.90 | ↓8.09% | 23.03 | 9.68 | ↓16.25% |
| | Passive | 228.7 | 96.34 | ↓4.54% | 183.60 | 77.26 | ↓7.36% | 150.21 | 64.49 | ↓14.76% | 39.46 | 16.21 | ↓9.82% | 31.08 | 13.18 | ↓14.26% | 24.65 | 11.14 | ↓24.43% |
| | Active | 223.1 | 93.94 | ↓1.96% | 176.82 | 73.85 | ↓3.40% | 143.15 | 60.26 | ↓9.37% | 36.94 | 14.36 | ↓2.81% | 28.35 | 10.99 | ↓4.23% | 21.91 | 8.77 | ↓10.60% |
| | Back_trans | 232.6 | 98.67 | ↓6.33% | 187.14 | 80.10 | ↓9.43% | 153.64 | 67.80 | ↓17.38% | 39.99 | 16.78 | ↓11.30% | 31.56 | 13.91 | ↓16.03% | 25.22 | 11.94 | ↓27.31% |
| GT | GT | 218.8 | 96.66 | — | 171.01 | 171.01 | — | 130.89 | 61.46 | — | 35.93 | 14.85 | — | 27.20 | 11.27 | — | 19.81 | 8.79 | — |

Table 19: Text-to-image generation results of GLIDE (FID and CLIP-FID), where "GT" means images generated by GT captions.

| | | FID | | | | | | | | | CLIP_FID | | | | | | | | |
| | | 4 image | | | 8 image | | | 16 image | | | 4 image | | | 8 image | | | 16 image | | |
| | | mean | std | MMI | mean | std | MMI | mean | std | MMI | mean | std | MMI | mean | std | MMI | mean | std | MMI |
|---|---|---|---|---|---|---|---|---|---|---|---|---|---|---|---|---|---|---|---|
| Character | Keyboard | 341.39 | 110.88 | ↓26.57% | 291.92 | 96.54 | ↓34.50% | 256.93 | 89.53 | ↓44.42% | 69.61 | 25.79 | ↓55.10% | 59.45 | 23.75 | ↓72.52% | 51.97 | 22.83 | ↓95.52% |
| | OCR | 305.16 | 108.71 | ↓13.14% | 255.81 | 93.50 | ↓17.86% | 219.74 | 85.15 | ↓23.51% | 58.83 | 24.76 | ↓31.08% | 48.81 | 22.44 | ↓41.64% | 41.18 | 21.08 | ↓54.93% |
| | CI | 333.82 | 110.89 | ↓23.76% | 284.49 | 97.22 | ↓31.08% | 248.06 | 89.42 | ↓39.43% | 67.45 | 25.38 | ↓50.29% | 57.16 | 23.35 | ↓65.87% | 49.53 | 22.30 | ↓86.34% |
| | CR | 339.82 | 108.85 | ↓25.99% | 290.13 | 94.90 | ↓33.68% | 254.63 | 88.26 | ↓43.12% | 69.44 | 25.47 | ↓54.72% | 59.17 | 23.57 | ↓71.71% | 51.53 | 22.68 | ↓93.87% |
| | CS | 339.20 | 110.27 | ↓25.76% | 288.79 | 95.76 | ↓33.06% | 253.56 | 88.69 | ↓42.52% | 67.75 | 25.00 | ↓50.96% | 57.58 | 22.91 | ↓67.09% | 50.17 | 22.06 | ↓88.75% |
| | CD | 340.87 | 111.23 | ↓26.37% | 291.76 | 98.32 | ↓34.43% | 252.68 | 87.79 | ↓42.03% | 67.23 | 24.82 | ↓49.80% | 57.07 | 22.75 | ↓65.61% | 49.52 | 21.80 | ↓86.31% |
| Word | SR | 306.08 | 110.11 | ↓13.48% | 255.45 | 94.17 | ↓17.70% | 254.17 | 88.23 | ↓42.86% | 56.7 | 22.89 | ↓26.34% | 46.65 | 20.27 | ↓35.37% | 39.19 | 18.88 | ↓47.44% |
| | RI | 286.23 | 106.35 | ↓6.12% | 234.68 | 88.62 | ↓8.13% | 196.88 | 77.27 | ↓10.66% | 50.86 | 20.32 | ↓13.32% | 40.64 | 17.14 | ↓17.93% | 32.91 | 15.14 | ↓23.81% |
| | RS | 283.53 | 103.71 | ↓5.12% | 230.54 | 85.17 | ↓6.22% | 195.23 | 77.45 | ↓9.74% | 48.96 | 18.82 | ↓9.09% | 38.61 | 15.33 | ↓12.04% | 30.84 | 13.06 | ↓16.03% |
| | RD | 286.36 | 106.72 | ↓6.17% | 234.16 | 88.39 | ↓7.89% | 196.08 | 76.79 | ↓10.21% | 50.01 | 19.44 | ↓11.43% | 39.86 | 16.32 | ↓15.67% | 32.20 | 14.34 | ↓21.14% |
| | IP | 278.34 | 105.05 | ↓3.19% | 225.52 | 85.21 | ↓3.91% | 189.22 | 74.35 | ↓6.36% | 47.64 | 18.07 | ↓6.15% | 37.21 | 14.32 | ↓7.98% | 29.39 | 11.80 | ↓10.57% |
| Sentence | Formal | 274.77 | 103.99 | ↓1.87% | 222.19 | 84.39 | ↓2.37% | 183.83 | 71.87 | ↓3.33% | 46.5 | 17.87 | ↓3.61% | 36.29 | 14.17 | ↓5.31% | 28.54 | 11.84 | ↓7.37% |
| | Casual | 275.48 | 103.52 | ↓2.13% | 222.96 | 84.60 | ↓2.73% | 184.38 | 72.27 | ↓3.64% | 46.82 | 18.3 | ↓4.32% | 36.57 | 14.63 | ↓6.12% | 28.76 | 12.29 | ↓8.20% |
| | Passive | 278.77 | 104.93 | ↓3.35% | 226.95 | 86.19 | ↓4.57% | 188.60 | 74.40 | ↓6.01% | 48.15 | 19.11 | ↓7.29% | 37.89 | 15.74 | ↓9.95% | 27.21 | 10.56 | ↓2.37% |
| | Active | 271.09 | 101.61 | ↓0.50% | 218.40 | 82.03 | ↓0.63% | 179.91 | 69.34 | ↓1.12% | 45.42 | 17.01 | ↓1.20% | 35.05 | 13.06 | ↓1.71% | 30.19 | 13.64 | ↓13.58% |
| | Back_trans | 283.70 | 107.07 | ↓5.18% | 231.85 | 88.16 | ↓6.82% | 190.23 | 73.53 | ↓6.92% | 49.21 | 19.86 | ↓9.65% | 39.13 | 16.59 | ↓13.55% | 31.46 | 14.55 | ↓18.36% |
| GT | GT | 269.73 | 269.73 | — | 217.04 | 81.72 | — | 177.91 | 68.55 | — | 44.88 | 16.57 | — | 34.46 | 12.47 | — | 26.58 | 9.79 | — |

Table 20: Quantitative results of Missing Object Rate (MOR) of Stable Diffusion. The most effective perturbation results are marked in bold, and the least effective ones are underlined. The results show that more objects are missing from the images generated by character-level perturbed captions.

| Threshold | Setting | GT | Keyboard | Ocr | CI | CR | CS | CD | SR | RI | RS | RD | IP | Formal | Casual | Passive | Active | Back_trans |
|---|---|---|---|---|---|---|---|---|---|---|---|---|---|---|---|---|---|---|
| | 4-images | 0.00 | -12.47 | -5.22 | -8.41 | **-13.25** | -12.15 | -12.63 | -8.23 | -3.14 | -7.33 | -6.05 | -2.81 | -2.10 | -1.42 | -1.36 | 0.27 | -0.86 |
| 0.7 | 8-images | 0.00 | -11.00 | -4.27 | -6.62 | **-11.79** | -11.09 | -10.76 | -6.77 | -1.62 | -6.59 | -4.31 | -2.83 | 0.01 | 0.69 | -0.17 | 1.34 | 0.44 |
| | 16-images | 0.00 | -11.53 | -4.29 | -6.96 | **-11.72** | -11.59 | -10.86 | -6.88 | -1.65 | -6.66 | -4.48 | -2.90 | -0.16 | 0.17 | -0.75 | 0.76 | 0.48 |
| | 4-images | 0.00 | -5.33 | -2.97 | -2.96 | **-6.60** | -3.97 | -2.45 | -1.00 | 0.72 | -1.51 | -4.63 | -1.88 | -0.31 | -2.18 | 2.17 | -0.30 | 0.65 |
| 0.5 | 8-images | 0.00 | -4.94 | -2.28 | -1.18 | **-5.83** | -2.48 | -1.55 | -0.34 | 1.70 | -1.26 | -2.72 | -1.06 | 0.17 | -1.00 | 3.41 | 0.42 | 1.02 |
| | 16-images | 0.00 | -4.95 | -1.76 | -1.65 | **-5.02** | -2.01 | -2.03 | -0.62 | 1.41 | -0.90 | -2.50 | -0.69 | 0.50 | 0.08 | 3.36 | 0.26 | 1.41 |

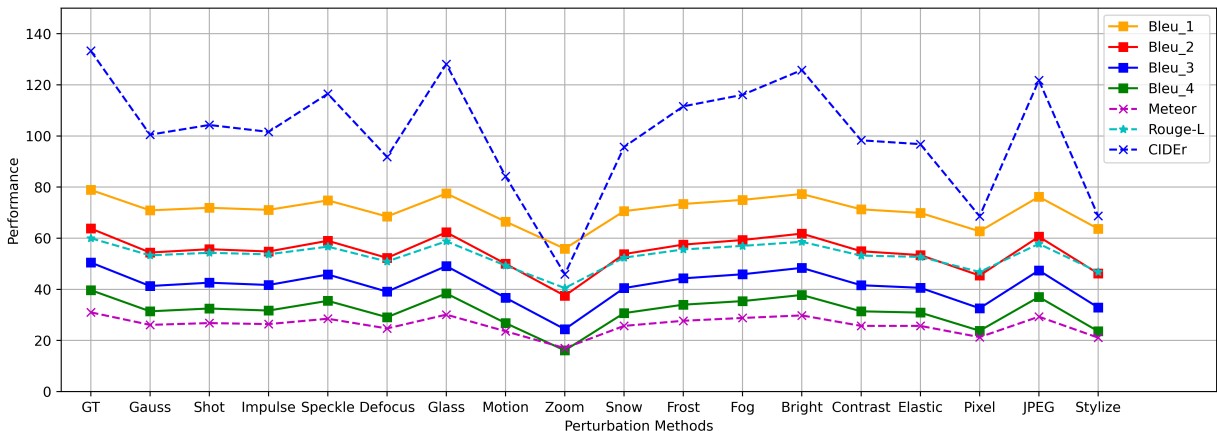

Figure 9: Image Captioning results of BLIP.

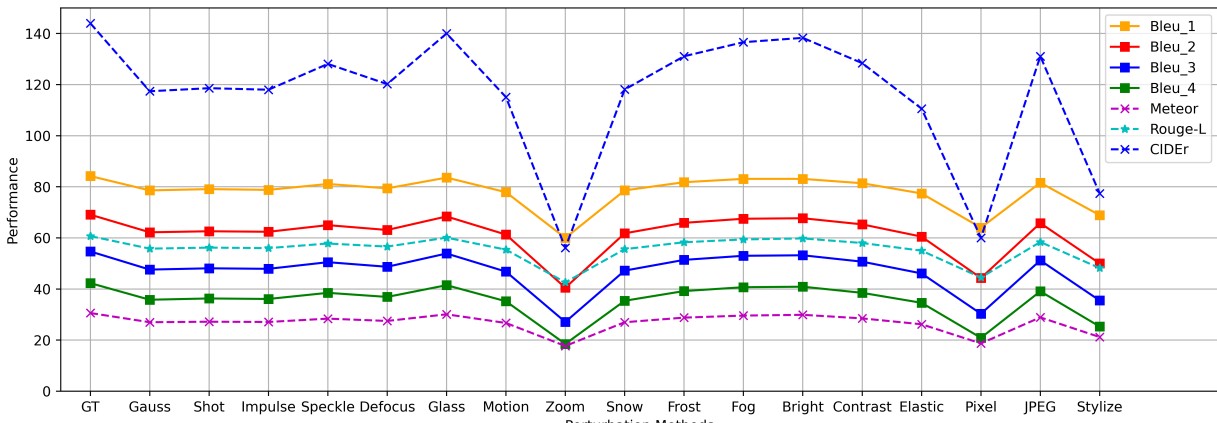

Figure 10: Image Captioning results of GRIT.

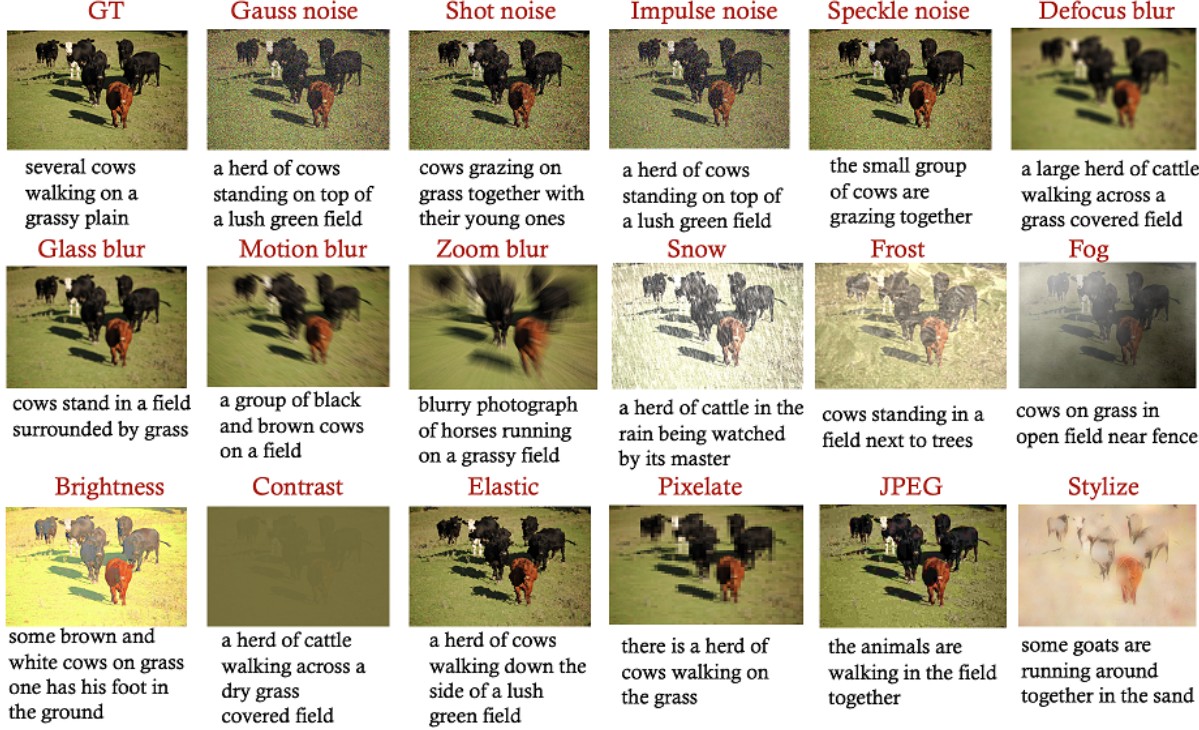

Figure 11: Examples of image captioning results under image perturbations of BLIP.

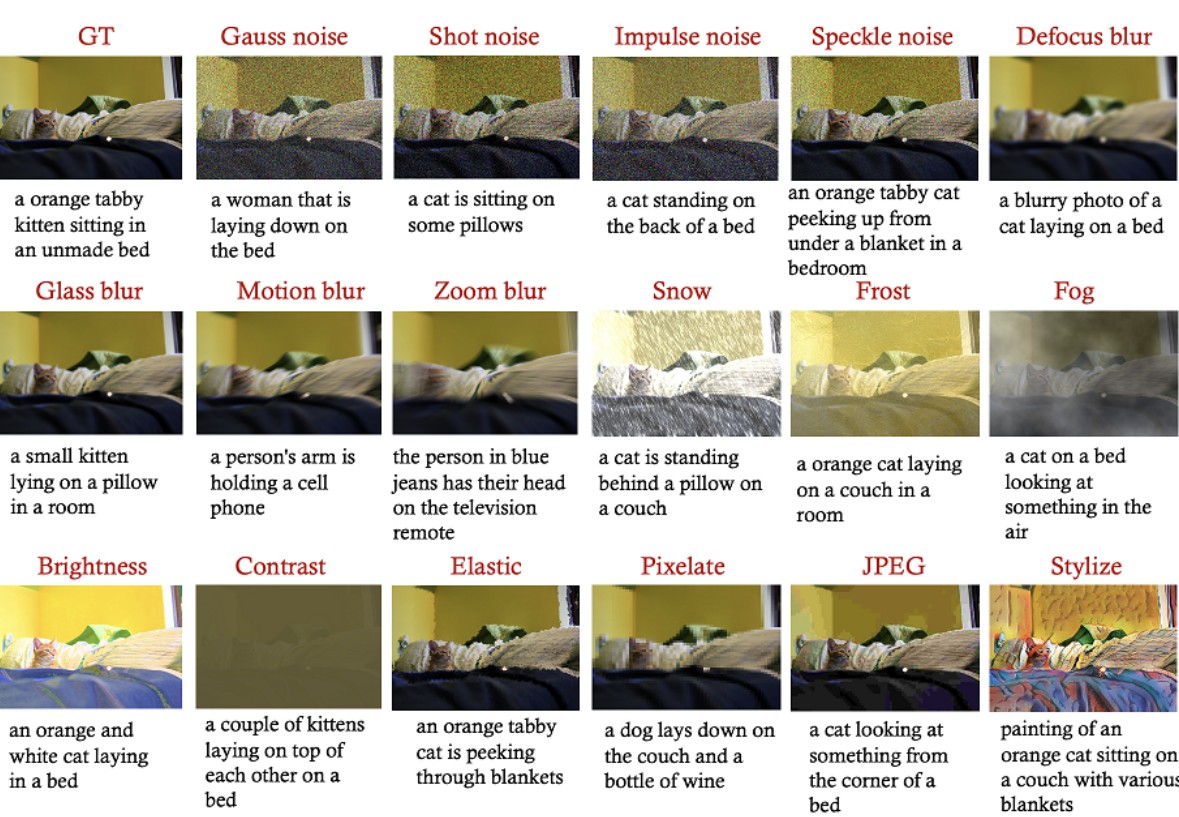

Figure 12: Examples of image captioning results under image perturbations of GRIT.

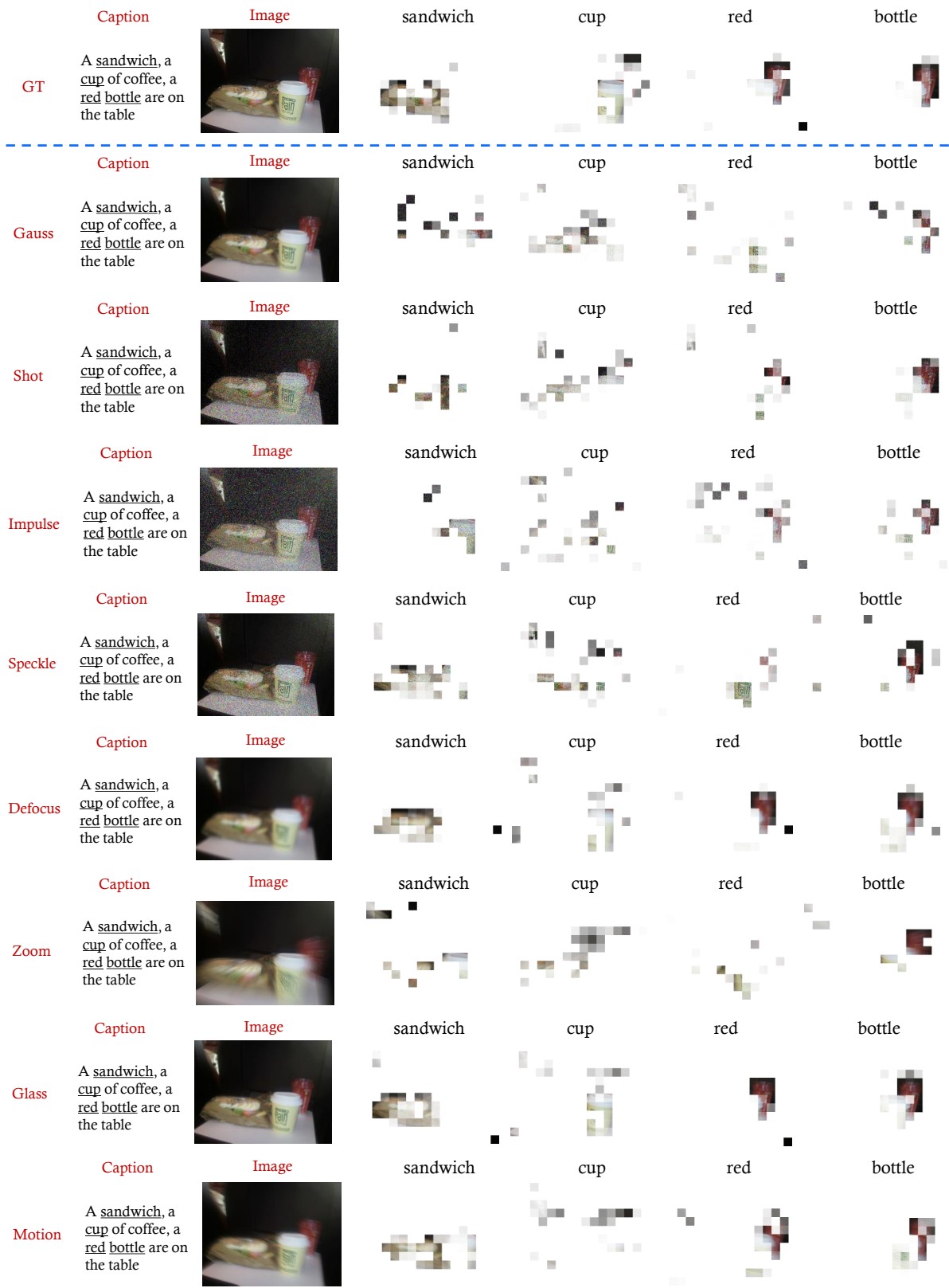

Figure 13: Optimal Transport (OT) alignment visualization between text and images under image perturbations (1/2).

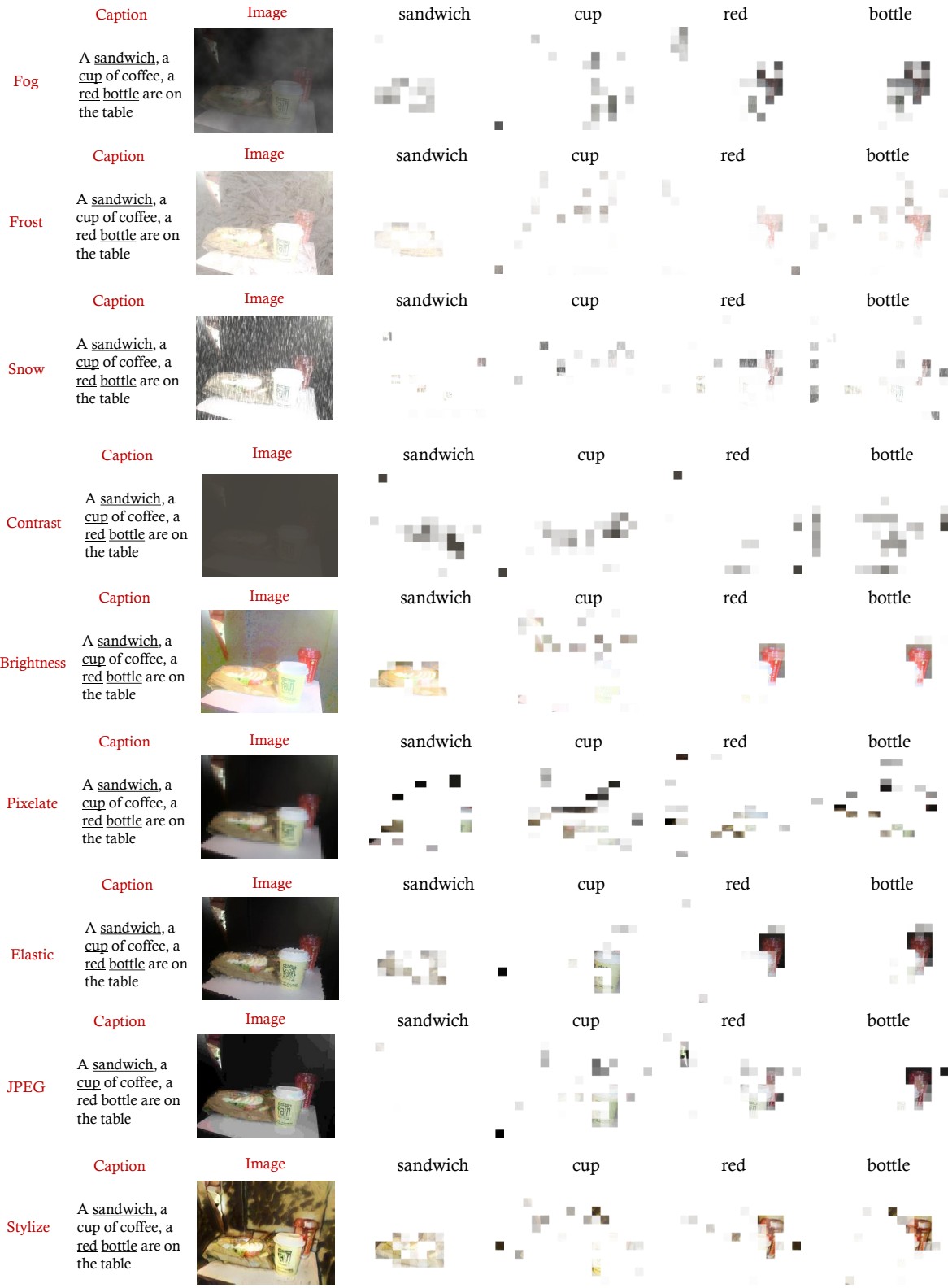

Figure 14: Optimal Transport (OT) alignment visualization between text and images under image perturbations (2/2).

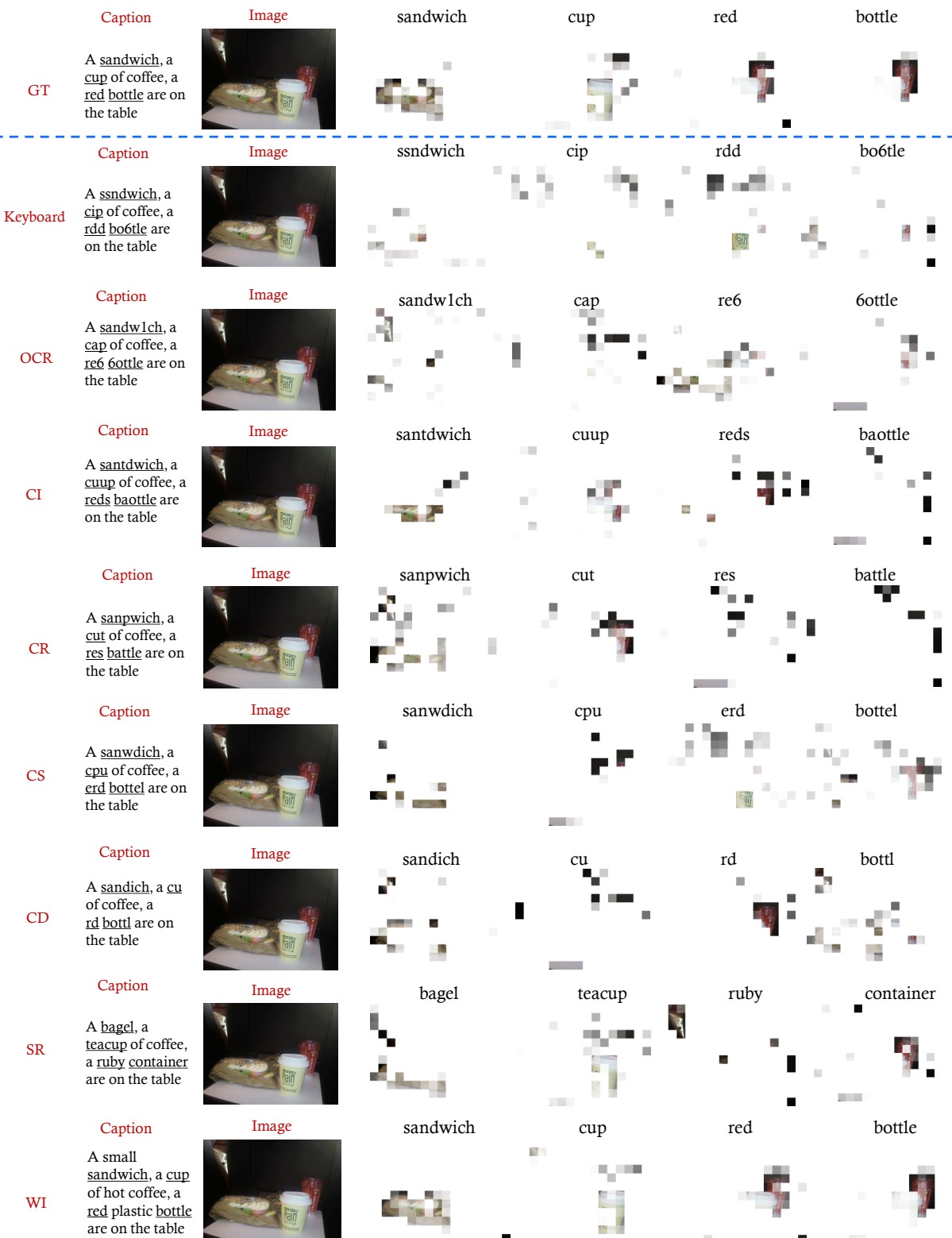

Figure 15: Optimal Transport (OT) alignment visualization between text and images under text perturbations (1/2).

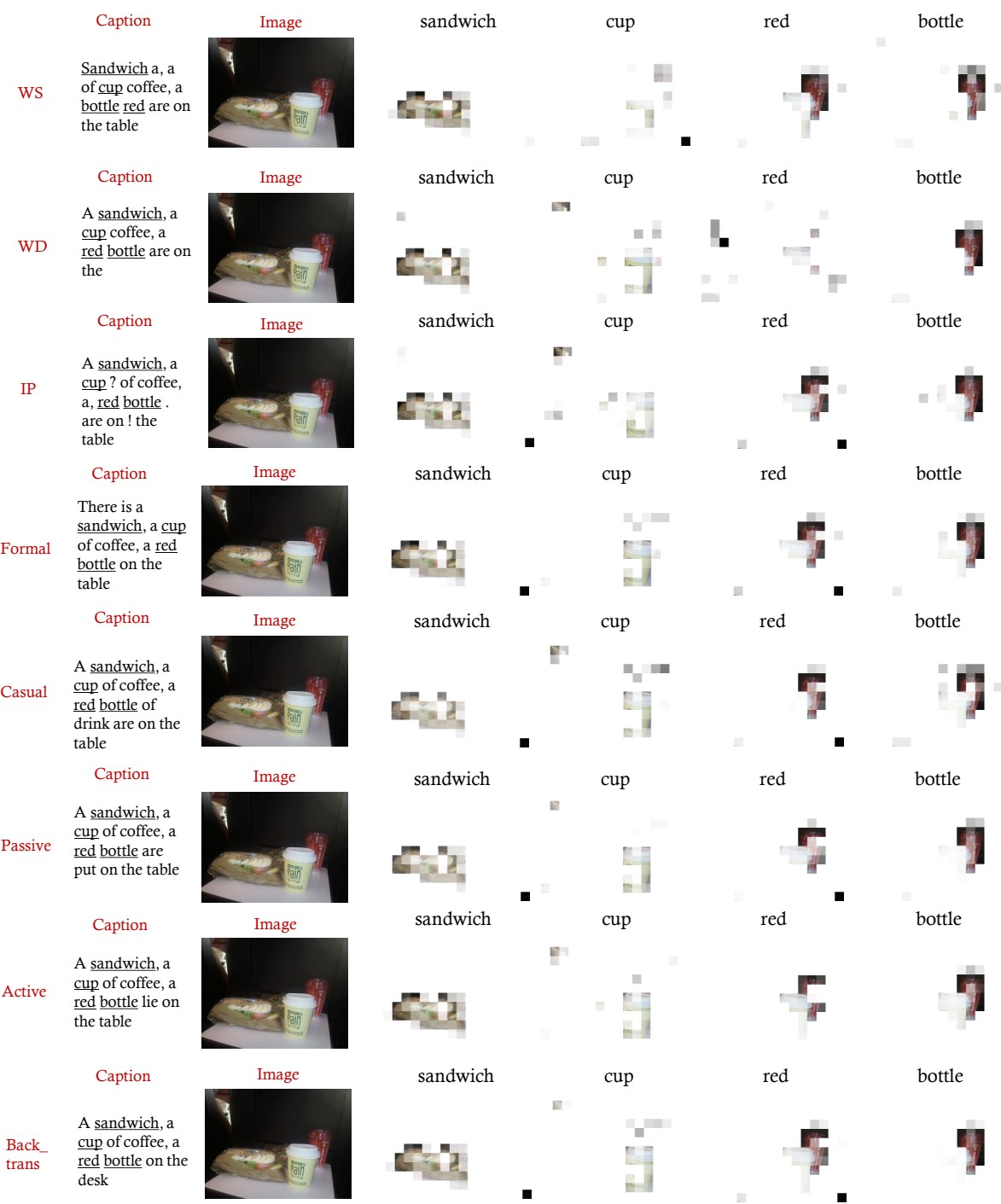

Figure 16: Optimal Transport (OT) alignment visualization between text and images under text perturbations (2/2).

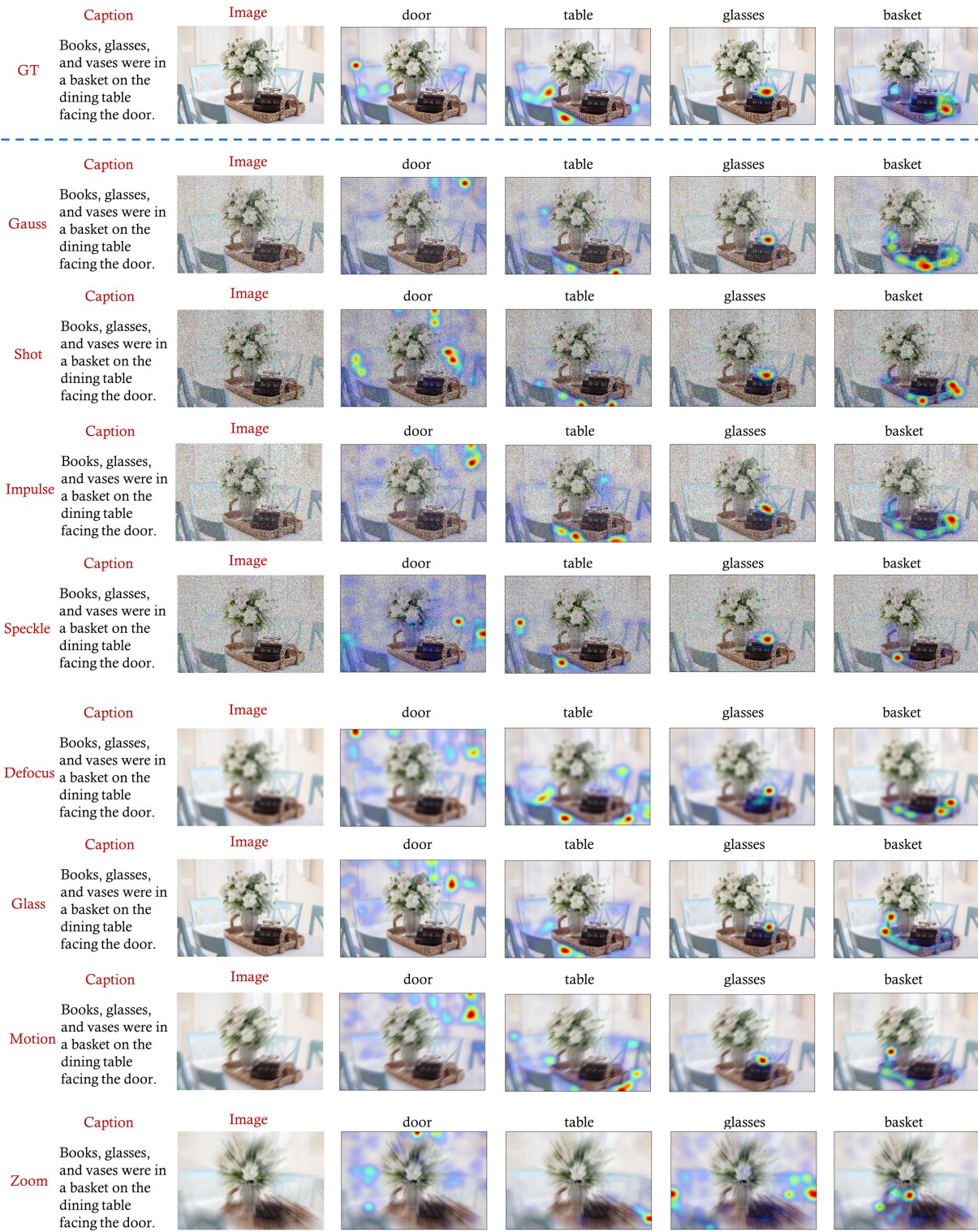

Figure 17: Grad-CAM visualizations on the cross-attention maps corresponding to individual words with image perturbations (1/2).

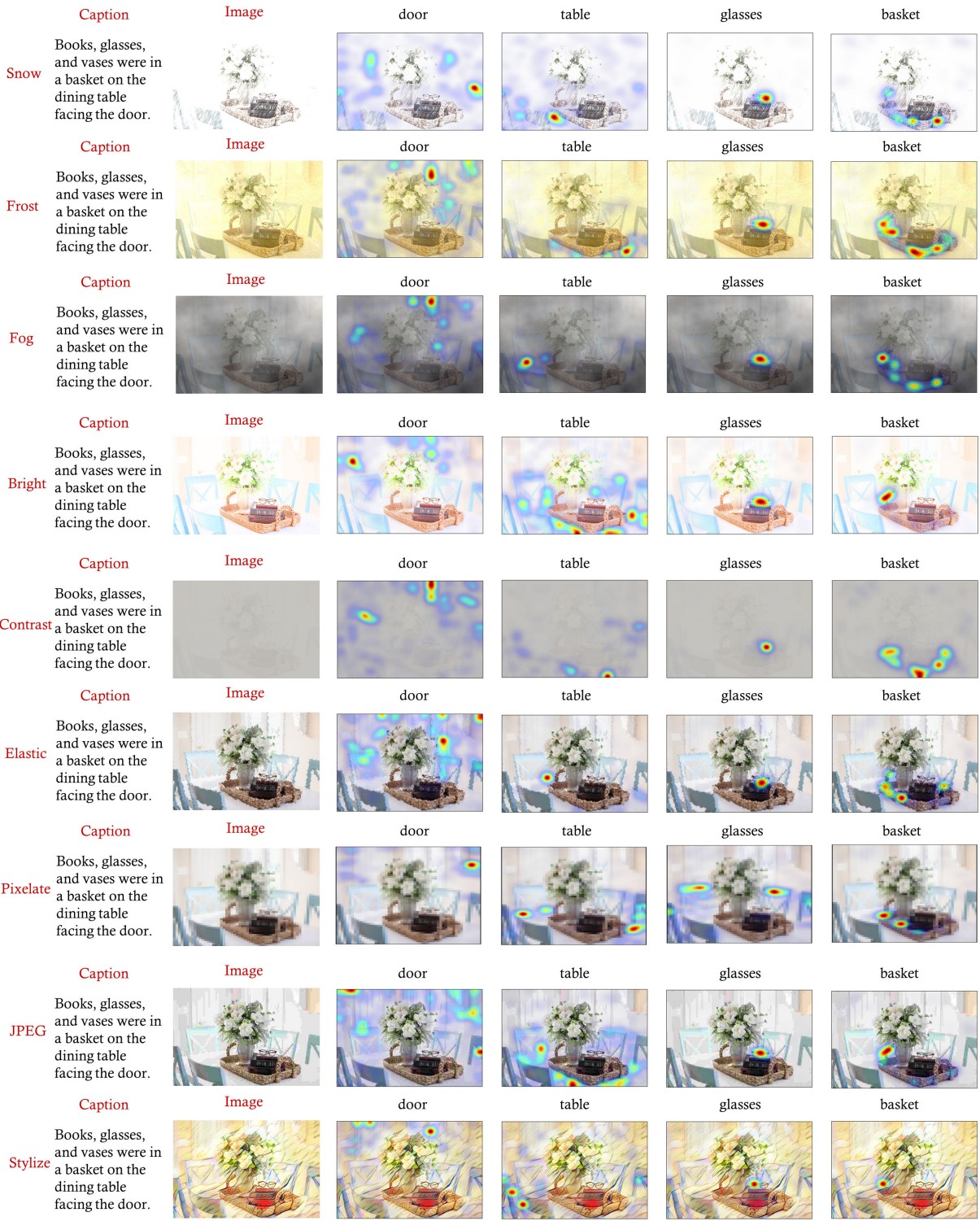

Figure 18: Grad-CAM visualizations on the cross-attention maps corresponding to individual words with image perturbations (2/2).

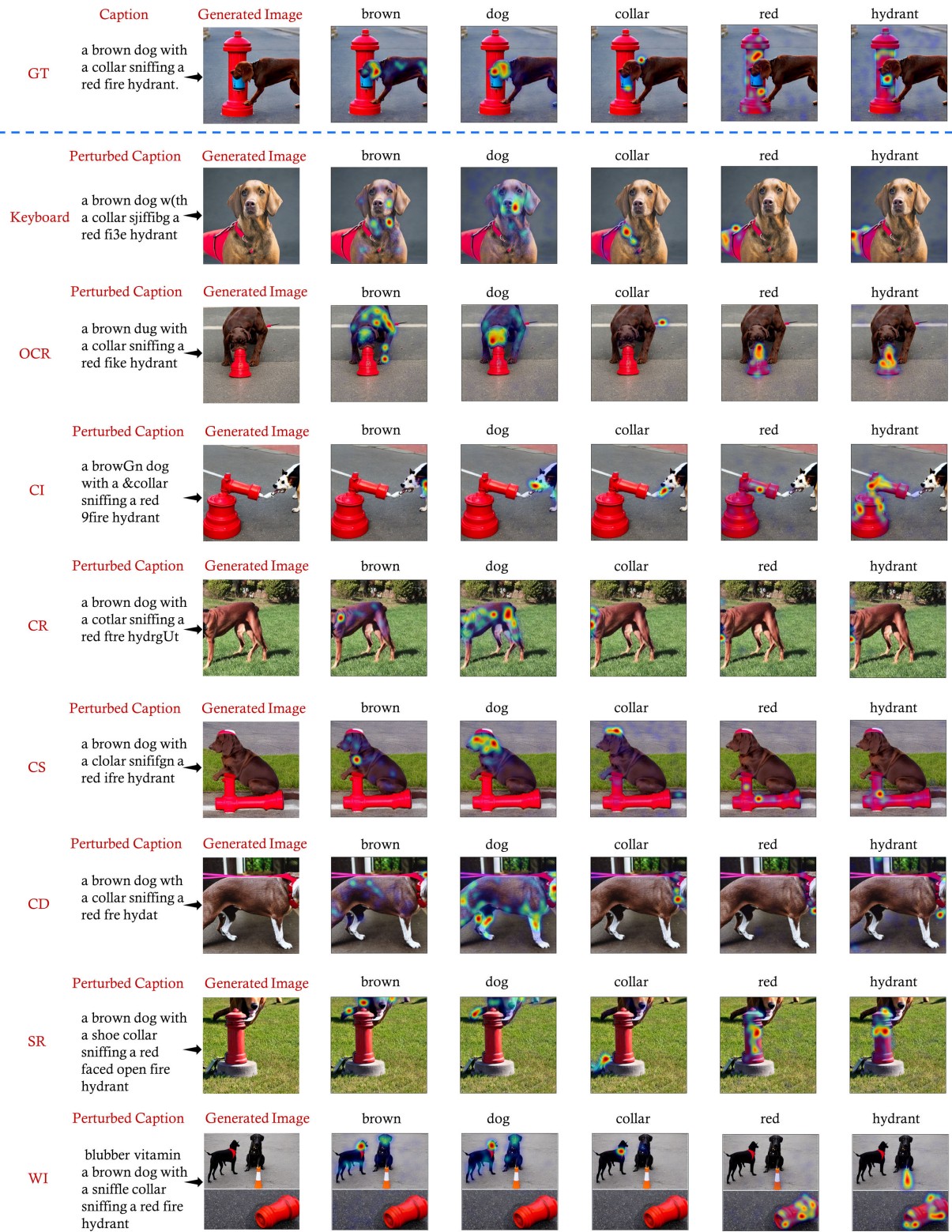

Figure 19: Text-to-image generation Grad-CAM visualizations on the cross-attention maps corresponding to individual words with text perturbations (1/2).

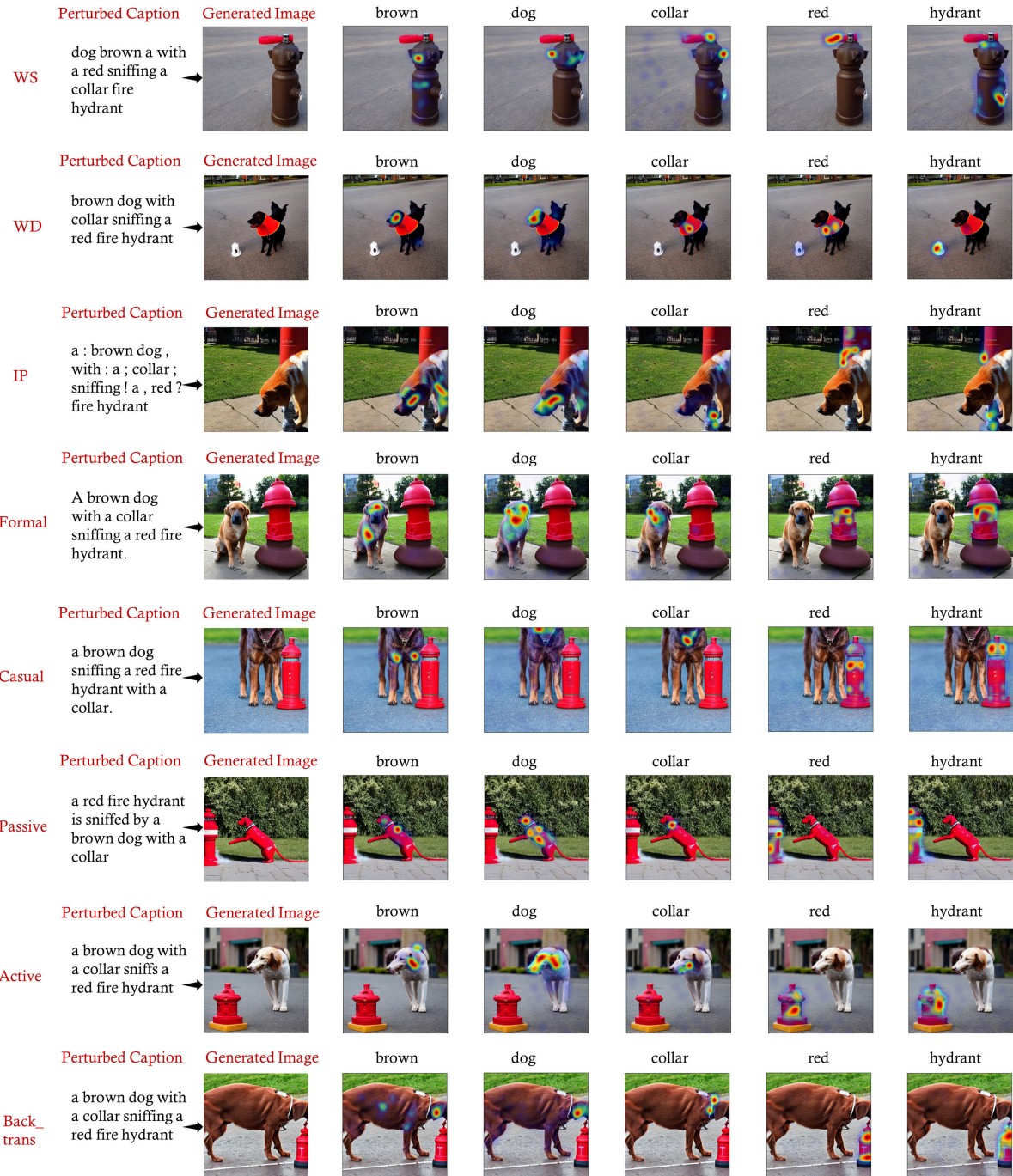

Figure 20: Text-to-image generation Grad-CAM visualizations on the cross-attention maps corresponding to individual words with text perturbations (2/2).

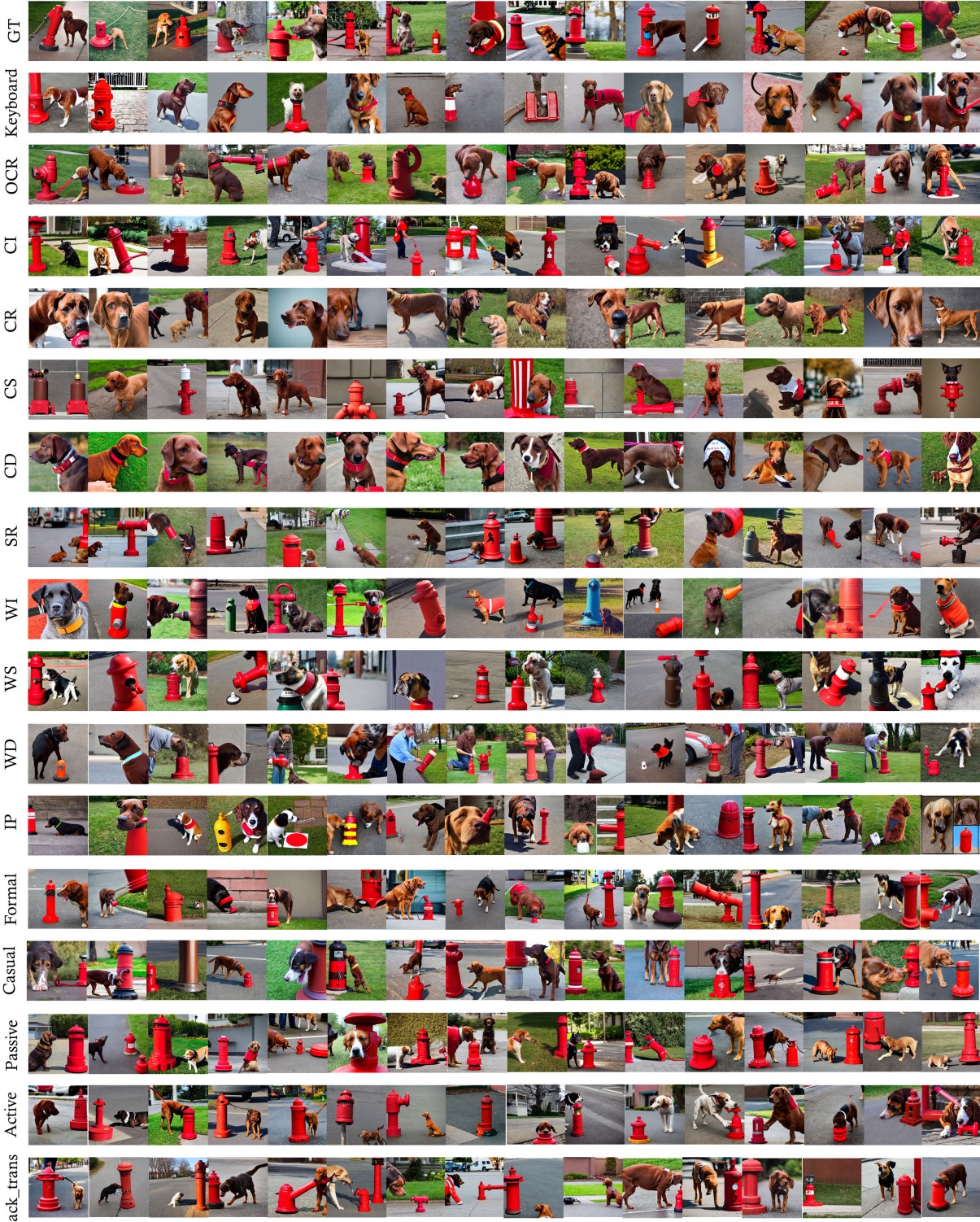

Figure 21: Text-to-image generation comparison on all 16 generated images. We find that though the generated images do not guarantee to perfectly show all the notions described in the captions, the probability of generating matched images by the unperturbed captions is higher than the perturbed captions, especially character-level perturbations.

Table 21: Adversarial perturbation methods.

| Modality | Adversarial Perturbation Methods |
|---|---|
| Image-only | FGSM |
| Text-only | BERT-Attack |
| Multimodal | Fooling VQA, SSAP, SSAP-MIM, SSAP-SI, Co-Attack |

Table 22: Image-text retrieval results by adding adversarial perturbations on image modality only by FGSM.

| Dataset | Method | Clean | GFSM |
|---|---|---|---|
| Flickr30K | ALBEF | 577.7 | 331.2 |
| | CLIP | 544.3 | 358.2 |
| COCO | ALBEF | 504.5 | 215.8 |
| | CLIP | 420.5 | 198.2 |

Table 23: Image-text retrieval results by adding adversarial perturbations on text modality only by BERT-Attack.

| Dataset | Method | Clean | BERT-Attack |
|---|---|---|---|
| Flickr30K | ALBEF | 577.7 | 534.9 |
| | CLIP | 544.3 | 512.3 |
| COCO | ALBEF | 504.5 | 431.8 |
| | CLIP | 420.5 | 374.6 |

Table 24: Image-text retrieval results by adding adversarial perturbations on multi-modality by Fooling VQA, SSAP, SSAP-MIM, SSAP-SI, and Co-Attack.

| Dataset | Method | Clean | Fooling VQA | SSAP | SSAP-MIM | SSAP-SI | Co-Attack |
|---|---|---|---|---|---|---|---|
| Flickr30K | ALBEF | 577.7 | 535.0 | 231.4 | 252.9 | 206.7 | 210.2 |
| | CLIP | 544.3 | 510.8 | 288.8 | 327.5 | 262.2 | 145.8 |
| COCO | ALBEF | 504.5 | 340.8 | 221.8 | 252.2 | 205.9 | 193.8 |
| | CLIP | 420.5 | 376.0 | 237.4 | 254.5 | 225.7 | 172.3 |

Table 25: ViLT image perturbation performance comparison of Fine-tuned (FT) image-text retrieval on Flickr30K and COCO datasets (results are averaged on five perturbation levels).

| | | Flickr30K (1K) | | | | | | | | | MSCOCO (5K) | | | | | | | | |
| | Method | Text Retrieval | | | | Image Retrieval | | | | | Text Retrieval | | | | Image Retrieval | | | | |
| | | R@1 | R@5 | R@10 | Mean | R@1 | R@5 | R@10 | Mean | RSUM | R@1 | R@5 | R@10 | Mean | R@1 | R@5 | R@10 | Mean | RSUM |
|---|---|---|---|---|---|---|---|---|---|---|---|---|---|---|---|---|---|---|---|
| Noise | Gaussian | 57.7 | 79.0 | 84.5 | 73.7 | 43.3 | 70.0 | 78.6 | 64.0 | 413.0 | 47.7 | 73.8 | 82.5 | 68.0 | 33.5 | 61.7 | 73.1 | 56.1 | 372.2 |
| | Shot | 58.9 | 80.4 | 85.7 | 75.0 | 43.9 | 70.9 | 79.7 | 64.8 | 419.6 | 47.9 | 73.9 | 82.6 | 68.1 | 33.3 | 61.7 | 73.2 | 56.1 | 372.6 |
| | Impluse | 54.3 | 76.0 | 82.3 | 70.9 | 40.6 | 67.4 | 76.3 | 61.4 | 396.9 | 45.9 | 71.8 | 80.8 | 66.2 | 32.1 | 60.3 | 71.9 | 54.8 | 362.9 |
| | Speckle | 67.9 | 89.0 | 93.5 | 83.4 | 49.4 | 77.8 | 85.6 | 70.9 | 463.2 | 52.2 | 78.7 | 87.0 | 72.6 | 36.2 | 65.7 | 77.0 | 59.6 | 396.7 |
| Blue | Defocus | 58.0 | 80.3 | 86.9 | 75.1 | 43.0 | 70.3 | 79.2 | 64.1 | 417.6 | 48.8 | 75.2 | 83.6 | 69.2 | 33.9 | 62.6 | 74.2 | 56.9 | 378.1 |
| | Glass | 74.5 | 92.7 | 95.9 | 87.7 | 55.2 | 81.9 | 88.9 | 75.3 | 489.0 | 58.6 | 84.4 | 91.1 | 79.7 | 40.8 | 70.6 | 81.3 | 63.2 | 432.0 |
| | Motion | 51.1 | 72.2 | 79.5 | 67.6 | 41.3 | 67.7 | 76.6 | 61.8 | 388.4 | 46.5 | 72.0 | 81.3 | 66.6 | 32.7 | 60.8 | 72.1 | 55.2 | 365.4 |
| | Zoom | 24.6 | 42.2 | 50.4 | 39.0 | 22.7 | 43.5 | 53.0 | 39.7 | 236.3 | 17.6 | 35.2 | 44.0 | 32.3 | 16.4 | 35.3 | 45.2 | 32.3 | 193.8 |
| Weather | Snow | 39.8 | 61.3 | 70.2 | 57.1 | 33.8 | 59.0 | 68.6 | 53.8 | 332.7 | 31.0 | 54.2 | 64.2 | 49.8 | 24.1 | 48.1 | 59.5 | 43.9 | 281.1 |
| | Frost | 65.1 | 87.2 | 92.1 | 81.5 | 47.9 | 76.1 | 84.6 | 69.6 | 453.1 | 46.2 | 72.7 | 81.5 | 66.8 | 32.7 | 60.8 | 72.3 | 55.3 | 366.1 |
| | Fog | 66.2 | 87.4 | 92.2 | 82.0 | 48.8 | 76.5 | 84.7 | 70.0 | 455.8 | 52.2 | 78.8 | 87.1 | 72.7 | 36.4 | 66.2 | 77.4 | 60.0 | 398.1 |
| | Brightness | 76.3 | 94.1 | 97.1 | 89.1 | 56.0 | 83.3 | 90.2 | 76.5 | 496.9 | 57.7 | 83.1 | 90.4 | 77.1 | 40.3 | 70.1 | 80.8 | 63.7 | 422.4 |
| Digital | Contrast | 53.3 | 70.7 | 75.9 | 66.7 | 39.5 | 62.6 | 70.1 | 57.4 | 372.2 | 41.7 | 64.2 | 72.1 | 59.3 | 29.6 | 54.7 | 64.9 | 49.7 | 327.1 |
| | Elastic | 67.2 | 87.3 | 91.8 | 82.1 | 50.8 | 78.5 | 86.1 | 71.8 | 461.7 | 54.0 | 78.7 | 86.5 | 73.1 | 37.9 | 67.1 | 77.9 | 61.0 | 402.2 |
| | Pixelate | 33.2 | 52.1 | 59.7 | 48.3 | 27.2 | 48.2 | 57.0 | 44.1 | 277.4 | 25.8 | 43.9 | 51.6 | 40.5 | 19.8 | 39.5 | 49.1 | 36.1 | 229.8 |
| | JPEG | 74.1 | 92.3 | 95.8 | 87.4 | 54.6 | 81.8 | 89.0 | 75.1 | 487.6 | 58.4 | 84.2 | 91.1 | 77.9 | 40.7 | 70.4 | 81.1 | 64.0 | 425.8 |
| Stylize | Stylized | 54.2 | 74.0 | 80.4 | 69.5 | 40.1 | 65.1 | 73.4 | 59.5 | 387.1 | 40.6 | 64.2 | 72.8 | 61.6 | 29.0 | 54.7 | 65.4 | 49.7 | 333.9 |

Table 26: CLIP image perturbation performance comparison of Zero-Shot (ZS) image-text retrieval on Flickr30K and COCO datasets (results are averaged on five perturbation levels).

| | | Flickr30K (1K) | | | | | | | | | MSCOCO (5K) | | | | | | | | |
| | Method | Text Retrieval | | | | Image Retrieval | | | | | Text Retrieval | | | | Image Retrieval | | | | |
| | | R@1 | R@5 | R@10 | Mean | R@1 | R@5 | R@10 | Mean | RSUM | R@1 | R@5 | R@10 | Mean | R@1 | R@5 | R@10 | Mean | RSUM |
|---|---|---|---|---|---|---|---|---|---|---|---|---|---|---|---|---|---|---|---|
| Noise | Gaussian | 75.1 | 92.8 | 96.0 | 88.0 | 61.7 | 85.1 | 90.9 | 79.3 | 501.7 | 47.8 | 72.1 | 80.6 | 66.9 | 34.7 | 58.7 | 69.1 | 54.2 | 363.0 |
| | Shot | 75.6 | 93.4 | 96.6 | 88.5 | 61.7 | 85.5 | 91.4 | 79.5 | 504.2 | 47.6 | 71.6 | 80.3 | 66.5 | 34.2 | 58.5 | 69.1 | 53.9 | 361.2 |
| | Impluse | 68.2 | 90.2 | 94.3 | 84.2 | 57.4 | 82.1 | 88.9 | 76.2 | 481.2 | 40.1 | 65.6 | 75.4 | 60.4 | 30.1 | 54.1 | 64.8 | 49.7 | 330.2 |
| | Speckle | 80.2 | 95.8 | 98.0 | 91.3 | 62.9 | 86.4 | 92.2 | 80.5 | 515.5 | 49.5 | 73.9 | 82.0 | 68.5 | 34.6 | 59.1 | 69.6 | 54.4 | 368.7 |
| Blur | Defocus | 74.7 | 93.4 | 96.6 | 88.2 | 61.3 | 85.1 | 91.1 | 79.1 | 502.1 | 46.5 | 71.3 | 80.0 | 65.9 | 33.7 | 58.3 | 68.8 | 53.6 | 358.6 |
| | Glass | 85.5 | 97.8 | 99.0 | 94.1 | 66.1 | 88.4 | 93.4 | 82.6 | 530.1 | 55.6 | 78.9 | 86.4 | 73.6 | 37.3 | 61.7 | 71.7 | 56.9 | 391.6 |
| | Motion | 77.0 | 94.1 | 97.0 | 89.4 | 63.5 | 86.2 | 91.9 | 80.6 | 509.7 | 48.8 | 72.3 | 80.4 | 67.1 | 34.2 | 58.2 | 68.3 | 53.6 | 362.2 |
| | Zoom | 62.3 | 84.6 | 90.6 | 79.1 | 54.8 | 79.2 | 86.3 | 73.5 | 457.8 | 32.4 | 57.0 | 67.2 | 52.2 | 26.9 | 50.1 | 61.0 | 46.0 | 294.6 |
| Weather | Snow | 64.8 | 86.9 | 93.1 | 81.6 | 56.2 | 81.4 | 88.3 | 75.3 | 470.7 | 32.3 | 56.2 | 67.8 | 52.1 | 26.8 | 50.1 | 61.4 | 46.1 | 294.7 |
| | Frost | 72.8 | 92.6 | 96.5 | 87.3 | 59.4 | 84.0 | 90.4 | 77.9 | 495.6 | 41.1 | 65.6 | 75.6 | 60.8 | 29.4 | 53.2 | 64.1 | 48.9 | 329.0 |
| | Fog | 80.8 | 96.1 | 98.2 | 91.7 | 64.6 | 87.3 | 92.7 | 81.5 | 519.7 | 51.3 | 75.5 | 83.6 | 70.2 | 34.0 | 58.5 | 68.8 | 53.8 | 371.8 |
| | Brightness | 85.2 | 97.6 | 98.9 | 93.9 | 66.4 | 88.6 | 93.4 | 82.8 | 530.1 | 56.5 | 79.8 | 87.4 | 74.6 | 36.4 | 60.7 | 71.1 | 56.0 | 391.9 |
| Digital | Contrast | 80.7 | 95.9 | 98.0 | 91.5 | 62.7 | 86.2 | 91.9 | 80.3 | 515.4 | 48.0 | 71.5 | 80.1 | 66.5 | 32.5 | 56.9 | 67.4 | 52.2 | 356.4 |
| | Elastic | 79.5 | 94.9 | 97.3 | 90.6 | 61.6 | 85.8 | 91.4 | 79.6 | 510.4 | 50.6 | 74.7 | 83.1 | 69.5 | 33.8 | 58.5 | 69.1 | 53.8 | 369.7 |
| | Pixelate | 68.4 | 87.6 | 92.0 | 82.7 | 55.5 | 79.6 | 86.4 | 73.8 | 469.5 | 36.3 | 60.4 | 70.3 | 55.7 | 27.9 | 51.3 | 61.9 | 47.0 | 308.2 |
| | JPEG | 83.6 | 96.8 | 98.4 | 92.9 | 65.8 | 87.4 | 92.7 | 82.0 | 524.6 | 55.3 | 78.9 | 86.4 | 73.5 | 35.9 | 60.7 | 70.9 | 55.8 | 388.0 |
| Stylize | Stylized | 65.3 | 83.3 | 88.3 | 79.0 | 51.6 | 75.8 | 83.2 | 70.2 | 447.6 | 39.9 | 62.8 | 72.2 | 58.3 | 28.0 | 50.8 | 61.2 | 46.7 | 314.9 |

Table 27: CLIP image perturbation performance comparison of Fine-tuned (FT) image-text retrieval on Flickr30K and COCO datasets (results are averaged on five perturbation levels).

| | Method | Flickr30K (1K) | | | | | | | | | MSCOCO (5K) | | | | | | | | | |
| | | Text Retrieval | | | | Image Retrieval | | | | | Text Retrieval | | | | Image Retrieval | | | | | |
| | | R@1 | R@5 | R@10 | Mean | R@1 | R@5 | R@10 | Mean | RSUM | R@1 | R@5 | R@10 | Mean | R@1 | R@5 | R@10 | Mean | RSUM |
| Noise | Gaussian | 72.7 | 91.2 | 95.0 | 86.3 | 63.1 | 86.5 | 91.6 | 80.4 | 500.1 | 43.0 | 70.3 | 80.1 | 64.5 | 35.1 | 63.5 | 75.1 | 57.9 | 367.2 |
| | Shot | 73.0 | 91.9 | 95.8 | 86.9 | 63.9 | 87.1 | 92.1 | 81.0 | 503.8 | 42.4 | 69.9 | 79.9 | 64.1 | 34.9 | 63.3 | 74.9 | 57.7 | 365.3 |
| | Impluse | 65.1 | 87.9 | 92.5 | 81.8 | 59.2 | 84.3 | 90.1 | 77.9 | 479.2 | 35.6 | 63.0 | 74.3 | 57.6 | 29.8 | 58.3 | 70.7 | 53.0 | 331.7 |
| | Speckle | 78.1 | 95.0 | 97.8 | 90.3 | 66.9 | 89.9 | 94.4 | 83.7 | 522.1 | 36.5 | 65.7 | 77.1 | 59.8 | 36.5 | 65.7 | 77.1 | 59.8 | 381.5 |
| Blur | Defocus | 70.1 | 90.2 | 94.5 | 84.9 | 61.6 | 85.6 | 91.4 | 79.5 | 493.4 | 43.7 | 71.7 | 81.5 | 65.6 | 35.2 | 63.8 | 75.2 | 58.1 | 371.0 |
| | Glass | 82.3 | 97.1 | 99.1 | 92.9 | 70.6 | 91.9 | 95.8 | 86.1 | 536.9 | 52.3 | 80.1 | 88.5 | 73.7 | 40.8 | 69.9 | 80.6 | 63.8 | 412.2 |
| | Motion | 76.1 | 93.7 | 96.8 | 88.9 | 65.0 | 88.4 | 93.3 | 82.2 | 513.3 | 44.6 | 71.7 | 81.0 | 65.8 | 36.4 | 64.9 | 75.8 | 59.1 | 374.4 |
| | Zoom | 58.7 | 80.9 | 87.8 | 75.8 | 53.0 | 78.5 | 85.5 | 72.3 | 444.3 | 28.4 | 54.1 | 65.1 | 49.2 | 26.6 | 52.3 | 64.4 | 47.8 | 291.0 |
| Weather | Snow | 69.6 | 91.3 | 95.7 | 85.5 | 64.2 | 88.8 | 93.4 | 82.1 | 503.0 | 26.6 | 51.7 | 63.9 | 47.4 | 26.4 | 54.0 | 66.6 | 49.0 | 289.3 |
| | Frost | 81.7 | 97.0 | 98.9 | 92.5 | 69.1 | 90.9 | 95.0 | 85.0 | 532.5 | 37.3 | 65.2 | 75.8 | 59.4 | 30.3 | 58.4 | 70.4 | 53.0 | 337.3 |
| | Fog | 80.5 | 95.9 | 98.3 | 91.6 | 69.0 | 90.8 | 95.2 | 85.0 | 529.7 | 47.0 | 75.3 | 84.6 | 69.0 | 37.7 | 67.0 | 78.2 | 61.0 | 389.9 |
| | Brightness | 85.9 | 97.8 | 99.3 | 94.3 | 72.3 | 92.3 | 96.1 | 86.9 | 543.7 | 52.8 | 80.1 | 88.4 | 73.8 | 41.2 | 70.4 | 80.9 | 64.2 | 413.9 |
| Digital | Contrast | 78.1 | 94.9 | 97.5 | 90.2 | 66.9 | 89.8 | 94.3 | 83.6 | 521.5 | 43.4 | 71.6 | 81.5 | 65.5 | 35.6 | 64.1 | 75.5 | 58.4 | 371.7 |
| | Elastic | 76.9 | 93.8 | 96.9 | 89.2 | 65.4 | 88.0 | 92.9 | 82.1 | 513.9 | 45.8 | 73.6 | 82.8 | 67.4 | 36.2 | 65.0 | 76.3 | 59.1 | 379.7 |
| | Pixelate | 62.5 | 83.9 | 88.8 | 78.4 | 54.4 | 78.6 | 85.5 | 72.8 | 453.8 | 32.4 | 58.3 | 68.9 | 53.2 | 27.3 | 53.8 | 65.7 | 48.9 | 306.4 |
| | JPEG | 81.5 | 96.2 | 98.3 | 92.0 | 68.2 | 90.1 | 94.2 | 84.2 | 528.5 | 50.4 | 78.1 | 86.8 | 71.8 | 39.2 | 68.2 | 79.4 | 62.3 | 402.1 |
| Stylize | Stylized | 59.9 | 80.8 | 86.5 | 75.7 | 51.3 | 76.0 | 82.6 | 70.0 | 437.0 | 33.3 | 59.1 | 69.3 | 53.9 | 28.1 | 54.5 | 65.9 | 49.5 | 310.2 |

Table 28: BLIP image perturbation performance comparison of Fine-tuned (FT) image-text retrieval on Flickr30K and COCO datasets (results are averaged on five perturbation levels).

| | Method | Flickr30K (1K) | | | | | | | | | MSCOCO (5K) | | | | | | | | | |
| | | Text Retrieval | | | | Image Retrieval | | | | | Text Retrieval | | | | Image Retrieval | | | | | |
| | | R@1 | R@5 | R@10 | Mean | R@1 | R@5 | R@10 | Mean | RSUM | R@1 | R@5 | R@10 | Mean | R@1 | R@5 | R@10 | Mean | RSUM |
| Noise | Gaussian | 85.1 | 94.9 | 96.4 | 92.1 | 74.3 | 91.1 | 94.4 | 86.6 | 536.2 | 70.1 | 88.4 | 92.8 | 83.8 | 55.2 | 79.0 | 86.4 | 73.5 | 471.9 |
| | Shot | 85.4 | 95.0 | 96.8 | 92.4 | 75.1 | 91.6 | 95.0 | 87.3 | 538.9 | 70.1 | 88.2 | 92.8 | 83.7 | 55.2 | 79.2 | 86.5 | 73.7 | 472.1 |
| | Impluse | 83.3 | 93.4 | 95.7 | 90.8 | 72.9 | 89.9 | 93.5 | 85.4 | 528.6 | 68.7 | 87.6 | 92.3 | 82.9 | 54.5 | 78.6 | 86.1 | 73.1 | 467.7 |
| | Speckle | 91.3 | 98.2 | 99.1 | 96.2 | 80.2 | 94.8 | 97.2 | 90.7 | 560.8 | 74.4 | 91.5 | 95.0 | 87.0 | 58.4 | 81.6 | 88.5 | 76.2 | 489.5 |
| Blur | Defocus | 83.8 | 93.9 | 96.0 | 91.2 | 73.1 | 89.5 | 93.2 | 85.3 | 529.4 | 68.0 | 87.5 | 92.2 | 82.6 | 54.6 | 78.3 | 85.4 | 72.8 | 466.1 |
| | Glass | 94.6 | 99.6 | 99.8 | 98.0 | 83.4 | 96.1 | 98.0 | 92.5 | 571.6 | 79.1 | 94.3 | 97.2 | 90.2 | 62.0 | 84.3 | 90.3 | 78.9 | 507.2 |
| | Motion | 82.6 | 93.4 | 96.0 | 90.7 | 71.9 | 88.9 | 92.9 | 84.6 | 525.7 | 65.8 | 85.0 | 89.8 | 80.2 | 52.9 | 75.6 | 82.5 | 70.3 | 451.7 |
| | Zoom | 56.2 | 74.9 | 80.4 | 70.5 | 53.3 | 74.7 | 81.6 | 69.9 | 421.1 | 30.7 | 52.2 | 61.0 | 48.0 | 31.8 | 53.4 | 62.5 | 49.2 | 291.6 |
| Weather | Snow | 62.2 | 82.7 | 88.8 | 77.9 | 56.7 | 79.7 | 86.5 | 74.3 | 456.6 | 58.3 | 80.5 | 87.1 | 75.3 | 49.7 | 74.5 | 82.8 | 69.0 | 432.8 |
| | Frost | 79.1 | 93.0 | 96.1 | 89.4 | 66.4 | 86.8 | 91.9 | 81.7 | 513.4 | 69.2 | 88.0 | 92.7 | 83.3 | 55.7 | 79.5 | 86.7 | 74.0 | 471.8 |
| | Fog | 92.9 | 99.2 | 99.6 | 97.2 | 82.8 | 96.0 | 98.0 | 92.3 | 568.5 | 74.7 | 91.7 | 95.4 | 87.2 | 60.1 | 82.9 | 89.4 | 77.5 | 494.2 |
| | Brightness | 95.6 | 99.6 | 99.8 | 98.3 | 84.8 | 96.5 | 98.3 | 93.2 | 574.5 | 79.1 | 94.0 | 96.8 | 90.0 | 61.9 | 84.4 | 90.5 | 78.9 | 506.8 |
| Digital | Contrast | 90.2 | 97.5 | 98.4 | 95.4 | 79.4 | 93.5 | 96.1 | 89.7 | 555.1 | 69.5 | 87.6 | 92.1 | 83.1 | 56.1 | 79.1 | 86.1 | 73.8 | 470.4 |
| | Elastic | 87.3 | 95.4 | 96.8 | 93.2 | 77.5 | 92.8 | 95.7 | 88.7 | 545.6 | 70.4 | 87.9 | 92.4 | 83.6 | 55.9 | 79.3 | 86.4 | 73.9 | 472.3 |
| | Pixelate | 75.6 | 88.2 | 91.5 | 85.1 | 64.7 | 83.0 | 87.8 | 78.5 | 490.8 | 56.1 | 76.3 | 82.6 | 71.6 | 44.9 | 68.3 | 76.5 | 63.3 | 404.7 |
| | JPEG | 92.7 | 98.5 | 99.3 | 96.8 | 81.2 | 94.9 | 97.2 | 91.1 | 563.8 | 77.5 | 93.2 | 96.4 | 89.1 | 60.1 | 83.0 | 89.5 | 77.5 | 499.6 |
| Stylize | Stylized | 73.3 | 86.4 | 89.3 | 83.0 | 64.1 | 82.1 | 87.0 | 77.7 | 482.1 | 55.1 | 75.3 | 81.6 | 70.7 | 45.9 | 68.6 | 76.5 | 63.6 | 402.9 |

Table 29: ALBEF image perturbation performance comparison of Fine-tuned (FT) image-text retrieval on Flickr30K and COCO datasets (results are averaged on five perturbation levels).

| | | Flickr30K (1K) | | | | | | | | | MSCOCO (5K) | | | | | | | | |
| | Method | Text Retrieval | | | | Image Retrieval | | | | | Text Retrieval | | | | Image Retrieval | | | | |
| | | R@1 | R@5 | R@10 | Mean | R@1 | R@5 | R@10 | Mean | RSUM | R@1 | R@5 | R@10 | Mean | R@1 | R@5 | R@10 | Mean | RSUM |
|---|---|---|---|---|---|---|---|---|---|---|---|---|---|---|---|---|---|---|---|
| Noise | Gaussian | 83.9 | 94.6 | 96.5 | 91.7 | 73.4 | 90.9 | 94.5 | 86.3 | 533.8 | 66.1 | 86.5 | 92.0 | 81.5 | 52.1 | 77.6 | 85.7 | 71.8 | 460.0 |
| | Shot | 84.9 | 95.2 | 97.1 | 92.4 | 74.0 | 91.8 | 95.2 | 87.0 | 538.3 | 66.2 | 86.6 | 92.0 | 81.6 | 52.1 | 77.9 | 85.8 | 71.9 | 460.6 |
| | Impulse | 83.7 | 94.4 | 96.3 | 91.5 | 73.0 | 90.5 | 94.1 | 85.9 | 532.0 | 66.0 | 86.8 | 92.1 | 81.6 | 52.1 | 77.6 | 85.7 | 71.8 | 460.3 |
| | Speckle | 90.1 | 98.1 | 99.1 | 95.8 | 78.8 | 94.6 | 97.2 | 90.2 | 557.8 | 69.9 | 89.3 | 94.1 | 84.4 | 54.7 | 80.1 | 87.6 | 74.1 | 475.8 |
| Blur | Defocus | 82.6 | 94.0 | 96.5 | 91.1 | 71.8 | 90.2 | 93.6 | 85.2 | 528.8 | 62.6 | 84.1 | 90.1 | 79.0 | 50.6 | 75.7 | 83.9 | 70.1 | 447.1 |
| | Glass | 93.8 | 99.2 | 99.7 | 97.6 | 82.3 | 96.3 | 97.9 | 92.1 | 569.2 | 75.1 | 92.1 | 96.2 | 87.8 | 58.1 | 82.2 | 89.2 | 76.5 | 493.0 |
| | Motion | 80.0 | 92.0 | 94.2 | 88.7 | 69.3 | 88.2 | 92.3 | 83.3 | 516.0 | 61.6 | 82.4 | 87.9 | 77.3 | 49.3 | 73.8 | 81.5 | 68.2 | 436.5 |
| | Zoom | 56.0 | 73.8 | 79.4 | 69.7 | 52.6 | 73.8 | 80.4 | 69.0 | 416.1 | 29.4 | 51.1 | 60.2 | 46.9 | 29.2 | 51.3 | 60.9 | 47.1 | 282.2 |
| Weather | Snow | 81.7 | 94.4 | 96.8 | 91.0 | 73.2 | 91.2 | 94.7 | 86.4 | 532.0 | 51.3 | 76.8 | 84.8 | 71.0 | 44.9 | 71.0 | 79.9 | 65.3 | 408.8 |
| | Frost | 90.4 | 97.5 | 98.8 | 95.5 | 79.5 | 94.7 | 97.2 | 90.5 | 558.1 | 62.1 | 84.7 | 90.7 | 79.2 | 51.0 | 76.7 | 84.6 | 70.8 | 449.8 |
| | Fog | 90.2 | 98.1 | 99.1 | 95.8 | 80.5 | 95.1 | 97.4 | 91.0 | 560.4 | 68.3 | 89.1 | 94.2 | 83.9 | 54.6 | 79.6 | 86.9 | 73.7 | 472.6 |
| | Brightness | 94.5 | 99.4 | 99.7 | 97.8 | 83.7 | 96.6 | 98.2 | 92.8 | 572.0 | 74.6 | 92.7 | 96.2 | 87.8 | 58.1 | 82.7 | 89.5 | 76.8 | 493.8 |
| Digital | Contrast | 88.2 | 96.7 | 97.9 | 94.3 | 78.3 | 93.4 | 96.0 | 89.2 | 550.6 | 63.8 | 85.0 | 90.8 | 79.9 | 51.7 | 76.5 | 84.3 | 70.8 | 452.1 |
| | Elastic | 85.3 | 94.7 | 96.5 | 92.2 | 75.3 | 91.8 | 95.1 | 87.4 | 538.7 | 65.7 | 85.6 | 91.1 | 80.8 | 51.7 | 76.5 | 84.4 | 70.9 | 455.0 |
| | Pixelate | 63.8 | 78.2 | 82.4 | 74.8 | 55.4 | 75.3 | 80.7 | 70.5 | 435.9 | 45.9 | 65.7 | 72.7 | 61.4 | 36.3 | 58.9 | 67.5 | 54.2 | 347.0 |
| | JPEG | 91.7 | 98.2 | 99.1 | 96.3 | 79.1 | 94.6 | 97.1 | 90.3 | 559.8 | 71.7 | 91.1 | 95.4 | 86.1 | 55.3 | 80.0 | 87.4 | 74.2 | 480.9 |
| Stylize | Stylized | 70.0 | 83.7 | 86.9 | 80.2 | 60.0 | 79.0 | 84.5 | 74.5 | 464.1 | 50.6 | 71.9 | 78.6 | 67.0 | 40.3 | 63.2 | 71.7 | 58.4 | 376.4 |

Table 30: TCL image perturbation performance comparison of Zero-Shot (ZS) image-text retrieval on Flickr30K and COCO datasets (results are averaged on five perturbation levels).

| | | Flickr30K (1K) | | | | | | | | | MSCOCO (5K) | | | | | | | | |
| | Method | Text Retrieval | | | | Image Retrieval | | | | | Text Retrieval | | | | Image Retrieval | | | | |
| | | R@1 | R@5 | R@10 | Mean | R@1 | R@5 | R@10 | Mean | RSUM | R@1 | R@5 | R@10 | Mean | R@1 | R@5 | R@10 | Mean | RSUM |
|---|---|---|---|---|---|---|---|---|---|---|---|---|---|---|---|---|---|---|---|
| Noise | Gaussian | 69.3 | 86.8 | 90.4 | 82.2 | 55.2 | 78.4 | 84.8 | 72.8 | 464.9 | 57.9 | 80.2 | 87.0 | 75.0 | 44.2 | 70.6 | 79.9 | 64.9 | 419.8 |
| | Shot | 70.1 | 87.0 | 91.2 | 82.8 | 55.5 | 78.4 | 84.7 | 72.9 | 467.0 | 57.2 | 79.9 | 86.9 | 74.7 | 44.0 | 70.5 | 79.9 | 64.8 | 418.4 |
| | Impulse | 67.3 | 85.9 | 90.3 | 81.2 | 53.7 | 77.4 | 83.8 | 71.6 | 458.4 | 57.2 | 80.2 | 87.0 | 74.8 | 43.8 | 70.4 | 79.8 | 64.7 | 418.4 |
| | Speckle | 78.1 | 92.9 | 96.4 | 89.1 | 60.3 | 82.3 | 88.2 | 76.9 | 498.0 | 62.0 | 84.2 | 90.5 | 78.9 | 46.7 | 73.3 | 82.4 | 67.5 | 439.0 |
| Blur | Defocus | 60.0 | 82.0 | 87.3 | 76.4 | 50.2 | 71.6 | 78.7 | 66.9 | 429.8 | 54.7 | 79.1 | 86.5 | 73.4 | 39.9 | 65.2 | 74.6 | 59.9 | 400.0 |
| | Glass | 78.2 | 94.0 | 97.2 | 89.8 | 63.8 | 84.1 | 89.4 | 79.1 | 506.6 | 66.7 | 88.7 | 94.7 | 83.4 | 46.5 | 72.6 | 81.6 | 66.9 | 450.8 |
| | Motion | 51.2 | 72.9 | 80.5 | 68.2 | 43.8 | 66.0 | 74.1 | 61.3 | 388.5 | 47.6 | 72.3 | 80.7 | 66.9 | 33.5 | 57.0 | 66.4 | 52.3 | 357.5 |
| | Zoom | 25.0 | 44.5 | 53.5 | 41.0 | 27.5 | 45.9 | 54.9 | 42.8 | 251.3 | 16.7 | 33.5 | 42.7 | 31.0 | 15.3 | 30.5 | 38.7 | 28.1 | 177.3 |
| Weather | Snow | 51.7 | 75.4 | 83.3 | 70.1 | 47.6 | 70.5 | 78.8 | 65.7 | 407.3 | 37.1 | 63.8 | 74.7 | 58.5 | 28.5 | 51.2 | 61.2 | 47.0 | 316.5 |
| | Frost | 62.8 | 85.5 | 91.3 | 79.9 | 52.0 | 75.2 | 82.8 | 70.0 | 449.5 | 48.9 | 75.1 | 83.9 | 69.3 | 34.5 | 59.7 | 69.8 | 54.7 | 372.0 |
| | Fog | 59.0 | 81.7 | 89.2 | 76.6 | 49.5 | 73.2 | 81.6 | 68.1 | 434.2 | 55.7 | 81.3 | 89.1 | 75.4 | 38.1 | 63.3 | 73.1 | 58.2 | 400.6 |
| | Brightness | 82.4 | 96.2 | 98.6 | 92.4 | 61.3 | 82.5 | 88.1 | 77.3 | 509.1 | 66.8 | 88.7 | 94.3 | 83.3 | 47.1 | 73.3 | 82.0 | 67.5 | 452.2 |
| Digital | Contrast | 69.8 | 89.9 | 94.0 | 84.6 | 56.3 | 78.3 | 85.0 | 73.2 | 473.2 | 58.5 | 82.9 | 89.7 | 77.0 | 41.2 | 67.2 | 76.6 | 61.7 | 416.1 |
| | Elastic | 62.4 | 80.6 | 85.9 | 76.3 | 52.0 | 73.3 | 80.3 | 68.5 | 434.4 | 50.6 | 73.3 | 80.7 | 68.2 | 35.6 | 59.6 | 69.2 | 54.8 | 369.0 |
| | Pixelate | 30.4 | 46.4 | 53.3 | 43.4 | 25.8 | 42.2 | 49.1 | 39.0 | 247.2 | 21.2 | 36.4 | 43.3 | 33.7 | 17.4 | 32.4 | 39.5 | 29.8 | 190.3 |
| | JPEG | 78.2 | 93.8 | 96.6 | 89.5 | 61.2 | 83.4 | 89.0 | 77.9 | 502.2 | 63.1 | 86.0 | 92.0 | 80.3 | 46.5 | 73.1 | 82.1 | 67.2 | 442.7 |
| Stylize | Stylized | 44.2 | 64.8 | 71.2 | 60.1 | 38.4 | 58.5 | 66.2 | 54.4 | 343.4 | 33.7 | 55.0 | 63.7 | 50.8 | 26.3 | 46.4 | 55.0 | 42.6 | 280.1 |

Table 31: TCL image perturbation performance comparison of Fine-tuned (FT) image-text retrieval on Flickr30K and COCO datasets (results are averaged on five perturbation levels).

| | | Flickr30K (1K) | | | | | | | | | MSCOCO (5K) | | | | | | | | |
| | | Text Retrieval | | | | Image Retrieval | | | | | Text Retrieval | | | | Image Retrieval | | | | |
| | Method | R@1 | R@5 | R@10 | Mean | R@1 | R@5 | R@10 | Mean | RSUM | R@1 | R@5 | R@10 | Mean | R@1 | R@5 | R@10 | Mean | RSUM |
|---|---|---|---|---|---|---|---|---|---|---|---|---|---|---|---|---|---|---|---|
| Noise | Gaussian | 83.1 | 94.3 | 96.7 | 91.4 | 71.4 | 90.3 | 94.1 | 85.3 | 529.9 | 64.8 | 85.8 | 91.3 | 80.6 | 50.8 | 76.6 | 84.9 | 70.8 | 454.3 |
| | Shot | 83.3 | 95.1 | 97.1 | 91.8 | 71.9 | 90.7 | 94.5 | 85.7 | 532.6 | 64.8 | 85.7 | 91.3 | 80.6 | 50.7 | 76.8 | 85.1 | 70.9 | 454.4 |
| | Impluse | 82.9 | 94.1 | 96.5 | 91.1 | 70.6 | 89.9 | 93.8 | 84.8 | 527.7 | 64.4 | 85.7 | 91.5 | 80.5 | 50.6 | 76.7 | 85.0 | 70.8 | 453.9 |
| | Speckle | 88.8 | 97.8 | 98.7 | 95.1 | 76.3 | 93.5 | 96.5 | 88.8 | 551.6 | 67.9 | 88.1 | 93.4 | 83.2 | 53.0 | 78.8 | 86.8 | 72.9 | 468.1 |
| Blur | Defocus | 77.0 | 90.6 | 93.5 | 87.1 | 66.6 | 86.1 | 90.7 | 81.1 | 504.5 | 62.8 | 84.6 | 90.7 | 79.4 | 50.1 | 75.8 | 83.8 | 69.9 | 447.8 |
| | Glass | 92.7 | 99.1 | 99.7 | 97.2 | 81.2 | 95.6 | 97.7 | 91.5 | 566.0 | 74.1 | 92.4 | 96.3 | 87.6 | 57.7 | 82.3 | 89.2 | 76.4 | 491.9 |
| | Motion | 78.9 | 92.2 | 94.9 | 88.7 | 68.1 | 87.6 | 92.2 | 82.6 | 513.9 | 60.5 | 81.9 | 87.8 | 76.7 | 48.4 | 73.4 | 81.7 | 67.8 | 433.8 |
| | Zoom | 51.8 | 70.5 | 76.4 | 66.2 | 48.4 | 71.3 | 78.9 | 66.2 | 397.3 | 24.5 | 45.2 | 54.6 | 41.5 | 27.2 | 49.3 | 59.1 | 45.2 | 259.9 |
| Weather | Snow | 78.8 | 93.3 | 95.9 | 89.3 | 70.0 | 89.9 | 93.8 | 84.6 | 521.7 | 51.5 | 76.4 | 84.7 | 70.9 | 44.6 | 71.2 | 80.5 | 65.4 | 408.9 |
| | Frost | 88.1 | 97.5 | 98.6 | 94.7 | 76.6 | 93.7 | 96.5 | 88.9 | 551.0 | 61.2 | 83.1 | 89.5 | 77.9 | 49.6 | 75.6 | 84.1 | 69.8 | 443.2 |
| | Fog | 88.1 | 98.0 | 99.1 | 95.1 | 77.9 | 94.2 | 96.7 | 89.6 | 554.1 | 67.7 | 88.3 | 93.5 | 83.2 | 53.9 | 79.5 | 87.3 | 73.5 | 470.1 |
| | Brightness | 93.7 | 99.0 | 99.6 | 97.4 | 81.9 | 95.9 | 97.9 | 91.9 | 568.0 | 73.4 | 91.6 | 95.9 | 87.0 | 57.1 | 82.0 | 89.1 | 76.1 | 489.1 |
| Digital | Contrast | 90.0 | 97.8 | 99.2 | 95.7 | 78.5 | 94.5 | 97.1 | 90.0 | 557.1 | 67.4 | 87.8 | 93.2 | 82.8 | 53.6 | 79.1 | 86.7 | 73.1 | 467.8 |
| | Elastic | 81.3 | 92.4 | 94.7 | 89.5 | 72.1 | 90.1 | 93.8 | 85.3 | 524.4 | 61.3 | 82.4 | 88.4 | 77.4 | 48.9 | 74.4 | 82.8 | 68.7 | 438.2 |
| | Pixelate | 50.1 | 66.2 | 72.0 | 62.8 | 45.7 | 65.4 | 72.5 | 61.2 | 372.0 | 37.7 | 57.1 | 65.0 | 53.3 | 32.0 | 54.1 | 63.1 | 49.8 | 309.1 |
| | JPEG | 90.2 | 98.3 | 99.3 | 95.9 | 77.1 | 93.9 | 96.7 | 89.2 | 555.4 | 69.9 | 89.3 | 94.3 | 84.5 | 54.1 | 79.8 | 87.4 | 73.8 | 474.9 |
| Stylize | Stylized | 65.0 | 80.7 | 85.0 | 76.9 | 57.4 | 77.5 | 83.2 | 72.7 | 448.7 | 45.3 | 67.5 | 75.3 | 62.7 | 38.8 | 62.6 | 71.3 | 57.6 | 360.9 |

Table 32: ViLT text perturbation performance comparison of Fine-tuned (FT) image-text retrieval on Flickr30K and COCO datasets (results are averaged on five perturbation levels).

| | | Flickr30K (1K) | | | | | | | | | MSCOCO (5K) | | | | | | | | |
| | | Text Retrieval | | | | Image Retrieval | | | | | Text Retrieval | | | | Image Retrieval | | | | |
| | Method | R@1 | R@5 | R@10 | Mean | R@1 | R@5 | R@10 | Mean | RSUM | R@1 | R@5 | R@10 | Mean | R@1 | R@5 | R@10 | Mean | RSUM |
|---|---|---|---|---|---|---|---|---|---|---|---|---|---|---|---|---|---|---|---|
| Character | Keyboard | 55.6 | 82.9 | 89.3 | 75.9 | 31.8 | 57.7 | 68.0 | 52.5 | 385.3 | 40.3 | 69.6 | 79.9 | 63.3 | 23.1 | 47.3 | 59.0 | 43.1 | 319.2 |
| | Ocr | 71.1 | 92.0 | 96.1 | 86.4 | 45.8 | 74.1 | 82.8 | 67.6 | 462.0 | 51.9 | 80.1 | 88.5 | 73.5 | 32.5 | 60.8 | 72.5 | 55.2 | 386.2 |
| | CI | 55.3 | 83.2 | 90.1 | 76.2 | 31.9 | 58.5 | 68.9 | 53.1 | 388.0 | 41.1 | 70.8 | 81.4 | 64.4 | 24.0 | 48.9 | 60.8 | 44.6 | 327.0 |
| | CR | 55.7 | 82.5 | 90.1 | 76.1 | 31.8 | 57.7 | 68.3 | 52.6 | 386.2 | 40.8 | 69.8 | 80.5 | 63.7 | 23.5 | 47.7 | 59.4 | 43.5 | 321.7 |
| | CS | 57.6 | 83.8 | 90.7 | 77.4 | 33.7 | 59.8 | 70.0 | 54.5 | 395.6 | 42.3 | 72.2 | 82.0 | 65.5 | 24.9 | 49.9 | 61.7 | 45.5 | 333.1 |
| | CD | 57.3 | 84.0 | 90.8 | 77.4 | 34.6 | 60.9 | 71.0 | 55.5 | 398.6 | 42.3 | 71.9 | 82.3 | 65.5 | 25.1 | 50.3 | 62.3 | 45.9 | 334.1 |
| Word | SR | 71.0 | 92.4 | 96.1 | 86.5 | 48.9 | 77.4 | 86.0 | 70.8 | 471.9 | 52.8 | 80.9 | 88.9 | 74.2 | 35.2 | 64.3 | 75.7 | 58.4 | 397.8 |
| | WI | 75.0 | 94.0 | 97.3 | 88.8 | 53.9 | 82.4 | 89.5 | 75.3 | 492.2 | 56.5 | 83.4 | 90.9 | 76.9 | 38.6 | 68.4 | 79.7 | 62.2 | 417.5 |
| | WS | 71.6 | 93.0 | 96.8 | 87.1 | 50.4 | 80.2 | 88.1 | 72.9 | 480.1 | 53.7 | 81.4 | 89.5 | 74.9 | 35.8 | 66.0 | 78.0 | 60.0 | 404.4 |
| | WD | 74.3 | 93.9 | 97.3 | 88.5 | 53.0 | 82.0 | 89.3 | 74.8 | 489.8 | 55.6 | 82.5 | 90.3 | 76.2 | 37.8 | 68.0 | 79.4 | 61.7 | 413.6 |
| | IP | 79.5 | 95.7 | 98.0 | 91.1 | 58.1 | 85.0 | 91.3 | 78.1 | 507.7 | 59.9 | 85.4 | 92.0 | 79.1 | 41.8 | 71.6 | 82.3 | 65.2 | 433.1 |
| Sentence | Formal | 79.5 | 95.7 | 98.6 | 91.3 | 59.2 | 85.6 | 91.5 | 78.8 | 510.1 | 61.1 | 85.8 | 92.2 | 79.7 | 42.6 | 72.2 | 82.6 | 65.8 | 436.5 |
| | Casual | 78.1 | 95.5 | 97.8 | 90.5 | 57.3 | 84.9 | 90.9 | 77.7 | 504.5 | 60.0 | 85.5 | 91.7 | 79.1 | 42.2 | 71.9 | 82.4 | 65.5 | 433.6 |
| | Passive | 74.0 | 94.6 | 97.4 | 88.7 | 53.2 | 80.8 | 88.1 | 74.0 | 488.1 | 57.9 | 84.4 | 91.4 | 77.9 | 40.0 | 69.3 | 80.2 | 63.2 | 423.2 |
| | Active | 78.5 | 95.1 | 98.3 | 90.6 | 58.6 | 85.7 | 92.1 | 78.8 | 508.3 | 60.9 | 85.9 | 92.2 | 79.7 | 42.9 | 72.3 | 82.9 | 66.0 | 437.1 |
| | Back_trans | 78.0 | 94.8 | 98.0 | 90.3 | 56.1 | 83.0 | 90.2 | 76.4 | 500.1 | 59.1 | 84.4 | 91.3 | 78.3 | 40.5 | 69.9 | 80.7 | 63.7 | 426.0 |

Table 33: CLIP text perturbation performance comparison of Zero-Shot (ZS) image-text retrieval on Flickr30K and COCO datasets (results are averaged on five perturbation levels).

| | | Flickr30K (1K) | | | | | | | | MSCOCO (5K) | | | | | | | | |
| | Method | Text Retrieval | | | | Image Retrieval | | | | Text Retrieval | | | | Image Retrieval | | | | |
| | | R@1 | R@5 | R@10 | Mean | R@1 | R@5 | R@10 | Mean | RSUM | R@1 | R@5 | R@10 | Mean | R@1 | R@5 | R@10 | Mean | RSUM |
|---|---|---|---|---|---|---|---|---|---|---|---|---|---|---|---|---|---|---|---|
| Character | Keyboard | 62.4 | 86.9 | 93.1 | 80.8 | 43.5 | 68.8 | 77.0 | 63.1 | 431.8 | 36.8 | 62.1 | 72.8 | 57.2 | 21.0 | 41.2 | 51.6 | 37.9 | 285.5 |
| | Ocr | 73.4 | 93.2 | 96.7 | 87.8 | 52.9 | 77.3 | 84.6 | 71.6 | 478.2 | 37.2 | 62.2 | 72.6 | 57.4 | 21.1 | 41.5 | 51.8 | 38.1 | 286.4 |
| | CI | 66.4 | 89.6 | 94.7 | 83.6 | 47.3 | 72.3 | 80.2 | 66.6 | 450.5 | 37.0 | 62.1 | 72.8 | 57.3 | 21.2 | 41.4 | 51.6 | 38.1 | 286.1 |
| | CR | 63.0 | 88.4 | 93.8 | 81.7 | 44.1 | 68.7 | 77.2 | 63.3 | 435.2 | 36.6 | 62.1 | 72.7 | 57.1 | 21.0 | 41.4 | 51.7 | 38.0 | 285.4 |
| | CS | 65.5 | 89.3 | 94.9 | 83.2 | 45.7 | 70.4 | 78.7 | 65.0 | 444.6 | 36.5 | 62.2 | 72.6 | 57.1 | 21.1 | 41.4 | 51.8 | 38.1 | 285.6 |
| | CD | 66.3 | 90.4 | 95.4 | 84.0 | 47.2 | 71.9 | 80.1 | 66.4 | 451.3 | 36.6 | 62.2 | 73.0 | 57.3 | 21.1 | 41.4 | 51.6 | 38.0 | 285.8 |
| Word | SR | 76.0 | 95.1 | 98.0 | 89.7 | 58.0 | 81.7 | 88.2 | 76.0 | 497.1 | 47.0 | 72.8 | 81.8 | 67.2 | 29.2 | 53.0 | 63.6 | 48.6 | 347.5 |
| | WI | 78.3 | 95.7 | 98.3 | 90.8 | 61.6 | 84.9 | 90.9 | 79.1 | 509.6 | 49.9 | 74.9 | 83.5 | 69.4 | 32.1 | 56.5 | 66.9 | 51.8 | 363.8 |
| | WS | 77.2 | 95.1 | 98.0 | 90.1 | 59.7 | 83.6 | 89.8 | 77.7 | 503.3 | 48.9 | 73.6 | 82.3 | 68.3 | 30.6 | 54.7 | 65.3 | 50.2 | 355.5 |
| | WD | 80.9 | 96.8 | 98.5 | 92.1 | 61.4 | 85.4 | 91.1 | 79.3 | 514.1 | 51.7 | 76.4 | 84.6 | 70.9 | 32.3 | 56.5 | 67.1 | 51.9 | 368.6 |
| | IP | 81.8 | 97.1 | 98.8 | 92.6 | 63.8 | 86.1 | 91.6 | 80.5 | 519.4 | 52.4 | 76.6 | 84.5 | 71.2 | 34.1 | 58.2 | 68.4 | 53.6 | 374.2 |
| Sentence | Formal | 86.4 | 98.6 | 99.1 | 94.7 | 66.0 | 88.5 | 93.1 | 82.5 | 531.7 | 56.8 | 80.4 | 87.7 | 75.0 | 36.4 | 60.9 | 70.8 | 56.0 | 393.0 |
| | Casual | 84.9 | 97.9 | 99.2 | 94.0 | 66.1 | 88.4 | 92.8 | 82.4 | 529.3 | 57.1 | 79.6 | 87.7 | 74.8 | 35.9 | 60.6 | 70.7 | 55.7 | 391.6 |
| | Passive | 84.3 | 96.9 | 99.2 | 93.5 | 64.8 | 87.3 | 92.2 | 81.5 | 524.8 | 54.3 | 77.8 | 86.1 | 72.7 | 34.1 | 58.4 | 68.9 | 53.8 | 379.6 |
| | Active | 85.6 | 97.9 | 99.2 | 94.2 | 66.9 | 88.8 | 93.1 | 82.9 | 531.4 | 57.5 | 80.3 | 87.9 | 75.2 | 36.1 | 60.8 | 70.9 | 55.9 | 393.5 |
| | Back_trans | 83.9 | 97.0 | 98.5 | 93.1 | 65.5 | 87.2 | 92.2 | 81.6 | 524.2 | 55.1 | 78.2 | 85.7 | 73.0 | 34.3 | 58.9 | 69.1 | 54.1 | 381.2 |

Table 34: CLIP text perturbation performance comparison of Fine-tuned (FT) image-text retrieval on Flickr30K and COCO datasets (results are averaged on five perturbation levels).

| | | Flickr30K (1K) | | | | | | | | MSCOCO (5K) | | | | | | | | |
| | Method | Text Retrieval | | | | Image Retrieval | | | | Text Retrieval | | | | Image Retrieval | | | | |
| | | R@1 | R@5 | R@10 | Mean | R@1 | R@5 | R@10 | Mean | RSUM | R@1 | R@5 | R@10 | Mean | R@1 | R@5 | R@10 | Mean | RSUM |
|---|---|---|---|---|---|---|---|---|---|---|---|---|---|---|---|---|---|---|---|
| Character | Keyboard | 67.0 | 91.2 | 96.2 | 84.8 | 48.3 | 74.0 | 81.6 | 68.0 | 458.4 | 36.8 | 66.1 | 78.1 | 60.3 | 24.3 | 49.4 | 61.3 | 45.0 | 316.1 |
| | Ocr | 76.2 | 95.4 | 98.4 | 90.0 | 58.5 | 83.3 | 89.1 | 77.0 | 500.9 | 36.8 | 66.3 | 77.9 | 60.4 | 24.4 | 49.7 | 61.5 | 45.2 | 316.7 |
| | CI | 71.4 | 93.3 | 96.8 | 87.2 | 53.2 | 78.1 | 84.8 | 72.0 | 477.6 | 36.3 | 66.6 | 78.2 | 60.4 | 24.4 | 49.6 | 61.4 | 45.1 | 316.5 |
| | CR | 68.9 | 91.7 | 96.1 | 85.6 | 48.7 | 74.5 | 81.7 | 68.3 | 461.6 | 36.5 | 66.3 | 78.1 | 60.3 | 24.3 | 49.7 | 61.5 | 45.2 | 316.4 |
| | CS | 70.7 | 92.4 | 96.6 | 86.6 | 51.0 | 76.6 | 83.7 | 70.4 | 471.1 | 36.5 | 66.5 | 78.2 | 60.4 | 24.4 | 49.6 | 61.4 | 45.1 | 316.7 |
| | CD | 70.9 | 93.3 | 97.2 | 87.2 | 52.1 | 77.5 | 84.5 | 71.3 | 475.5 | 36.7 | 66.1 | 77.9 | 60.3 | 24.2 | 49.5 | 61.3 | 45.0 | 315.6 |
| Word | SR | 78.0 | 96.4 | 98.5 | 91.0 | 63.4 | 87.2 | 92.0 | 80.9 | 515.4 | 45.3 | 75.0 | 85.1 | 68.5 | 33.8 | 62.7 | 74.3 | 56.9 | 376.2 |
| | WI | 81.0 | 97.0 | 99.0 | 92.3 | 68.3 | 90.4 | 94.7 | 84.4 | 530.4 | 48.4 | 77.3 | 86.8 | 70.8 | 37.3 | 66.8 | 78.1 | 60.7 | 394.6 |
| | WS | 80.8 | 97.0 | 99.0 | 92.2 | 66.1 | 89.3 | 93.9 | 83.1 | 526.0 | 48.0 | 77.1 | 86.7 | 70.6 | 35.9 | 65.3 | 76.9 | 59.4 | 389.9 |
| | WD | 81.0 | 97.4 | 99.1 | 92.5 | 67.9 | 90.7 | 95.0 | 84.5 | 531.1 | 49.1 | 77.7 | 86.8 | 71.2 | 37.1 | 66.7 | 78.0 | 60.6 | 395.3 |
| | IP | 83.0 | 97.9 | 99.2 | 93.4 | 69.9 | 91.2 | 95.1 | 85.4 | 536.4 | 51.5 | 79.5 | 88.1 | 73.0 | 39.1 | 68.7 | 79.6 | 62.5 | 406.6 |
| Sentence | Formal | 85.2 | 98.4 | 99.5 | 94.4 | 73.3 | 92.9 | 96.4 | 87.6 | 545.8 | 53.5 | 81.0 | 88.9 | 74.5 | 41.7 | 70.8 | 81.3 | 64.6 | 417.3 |
| | Casual | 83.9 | 97.6 | 99.4 | 93.6 | 72.5 | 92.3 | 96.4 | 87.1 | 542.1 | 52.5 | 80.6 | 89.0 | 74.0 | 41.4 | 70.4 | 81.2 | 64.4 | 415.2 |
| | Passive | 82.9 | 97.7 | 99.1 | 93.2 | 71.3 | 91.3 | 95.6 | 86.1 | 537.9 | 51.9 | 80.0 | 88.3 | 73.4 | 39.6 | 68.9 | 80.0 | 62.8 | 408.7 |
| | Active | 85.0 | 97.6 | 99.4 | 94.0 | 73.5 | 92.9 | 96.6 | 87.7 | 545.1 | 54.1 | 81.4 | 89.0 | 74.8 | 42.2 | 71.1 | 81.7 | 65.0 | 419.4 |
| | Back_trans | 83.8 | 97.7 | 99.0 | 93.5 | 70.4 | 91.2 | 95.2 | 85.6 | 537.3 | 51.4 | 79.1 | 88.2 | 72.9 | 39.6 | 68.5 | 79.5 | 62.5 | 406.2 |

Table 35: BLIP text perturbation performance comparison of Fine-tuned (FT) image-text retrieval on Flickr30K and COCO datasets (results are averaged on five perturbation levels).

| | | Flickr30K (1K) | | | | | | | | | MSCOCO (5K) | | | | | | | | |
| | | Text Retrieval | | | | Image Retrieval | | | | | Text Retrieval | | | | Image Retrieval | | | | |
| | Method | R@1 | R@5 | R@10 | Mean | R@1 | R@5 | R@10 | Mean | RSUM | R@1 | R@5 | R@10 | Mean | R@1 | R@5 | R@10 | Mean | RSUM |
|---|---|---|---|---|---|---|---|---|---|---|---|---|---|---|---|---|---|---|---|
| Character | Keyboard | 84.5 | 97.3 | 98.9 | 93.6 | 63.8 | 84.1 | 89.4 | 79.1 | 518.0 | 64.1 | 86.4 | 91.9 | 80.8 | 42.7 | 67.5 | 76.6 | 62.2 | 429.1 |
| | Ocr | 93.6 | 99.5 | 99.8 | 97.6 | 77.5 | 93.1 | 96.0 | 88.9 | 559.5 | 74.3 | 92.2 | 96.0 | 87.5 | 53.6 | 77.7 | 85.3 | 72.2 | 479.1 |
| | CI | 86.6 | 98.0 | 99.3 | 94.7 | 66.3 | 86.1 | 90.9 | 81.1 | 527.3 | 66.7 | 88.1 | 93.4 | 82.7 | 45.0 | 70.2 | 79.0 | 64.7 | 442.4 |
| | CR | 84.6 | 97.5 | 99.0 | 93.7 | 63.9 | 83.8 | 89.2 | 79.0 | 518.0 | 64.5 | 86.7 | 92.1 | 81.1 | 42.9 | 67.7 | 76.9 | 62.5 | 430.8 |
| | CS | 87.4 | 97.9 | 99.3 | 94.9 | 65.9 | 85.4 | 90.5 | 80.6 | 526.4 | 67.0 | 88.1 | 93.2 | 82.8 | 44.6 | 69.7 | 78.6 | 64.3 | 441.3 |
| | CD | 86.8 | 97.7 | 99.2 | 94.6 | 65.9 | 85.7 | 90.4 | 80.7 | 525.7 | 67.0 | 88.1 | 93.3 | 82.8 | 44.8 | 69.7 | 78.6 | 64.4 | 441.4 |
| Word | SR | 93.8 | 99.6 | 99.9 | 97.8 | 80.6 | 94.7 | 97.0 | 90.7 | 565.6 | 74.2 | 92.4 | 96.1 | 87.6 | 55.5 | 79.5 | 86.7 | 73.9 | 484.3 |
| | WI | 96.0 | 99.8 | 99.9 | 98.6 | 85.0 | 96.9 | 98.5 | 93.4 | 576.1 | 78.1 | 94.0 | 97.1 | 89.7 | 60.1 | 83.2 | 89.6 | 77.6 | 502.1 |
| | WS | 94.8 | 99.6 | 100.0 | 98.1 | 83.6 | 96.5 | 98.4 | 92.8 | 572.9 | 75.9 | 93.2 | 96.6 | 88.6 | 58.1 | 82.0 | 88.9 | 76.3 | 494.6 |
| | WD | 95.1 | 99.8 | 100.0 | 98.3 | 83.8 | 96.7 | 98.5 | 93.0 | 573.8 | 77.3 | 93.9 | 97.0 | 89.4 | 59.2 | 82.7 | 89.5 | 77.1 | 499.7 |
| | IP | 97.3 | 99.9 | 100.0 | 99.0 | 87.2 | 97.5 | 98.9 | 94.5 | 580.7 | 81.8 | 95.4 | 97.8 | 91.7 | 63.9 | 85.6 | 91.3 | 80.3 | 515.8 |
| Sentence | Formal | 96.5 | 99.9 | 100.0 | 98.8 | 86.7 | 97.1 | 98.8 | 94.2 | 579.0 | 81.7 | 95.2 | 97.6 | 91.5 | 63.5 | 85.3 | 91.2 | 80.0 | 514.4 |
| | Casual | 96.8 | 100.0 | 100.0 | 98.9 | 86.0 | 97.1 | 98.7 | 93.9 | 578.6 | 81.3 | 95.0 | 97.7 | 91.3 | 63.4 | 85.1 | 91.1 | 79.8 | 513.6 |
| | Passive | 96.8 | 99.8 | 99.9 | 98.8 | 83.3 | 96.5 | 98.2 | 92.7 | 574.5 | 80.5 | 94.7 | 97.3 | 90.8 | 61.7 | 83.8 | 90.2 | 78.6 | 508.1 |
| | Active | 97.1 | 99.9 | 100.0 | 99.0 | 86.6 | 97.2 | 98.7 | 94.2 | 579.6 | 81.6 | 95.2 | 97.7 | 91.5 | 64.0 | 85.5 | 91.3 | 80.3 | 515.4 |
| | Back_trans | 96.0 | 99.9 | 100.0 | 98.6 | 84.5 | 96.1 | 98.2 | 92.9 | 574.7 | 79.9 | 94.2 | 97.0 | 90.4 | 61.0 | 82.9 | 89.3 | 77.8 | 504.3 |

Table 36: ALBEF text perturbation performance comparison of Fine-tuned (FT) image-text retrieval on Flickr30K and COCO datasets (results are averaged on five perturbation levels).

| | | Flickr30K (1K) | | | | | | | | | MSCOCO (5K) | | | | | | | | |
| | | Text Retrieval | | | | Image Retrieval | | | | | Text Retrieval | | | | Image Retrieval | | | | |
| | Method | R@1 | R@5 | R@10 | Mean | R@1 | R@5 | R@10 | Mean | RSUM | R@1 | R@5 | R@10 | Mean | R@1 | R@5 | R@10 | Mean | RSUM |
|---|---|---|---|---|---|---|---|---|---|---|---|---|---|---|---|---|---|---|---|
| Character | Keyboard | 82.1 | 96.0 | 98.5 | 92.2 | 59.7 | 82.1 | 87.7 | 76.5 | 506.2 | 57.9 | 82.6 | 89.6 | 76.7 | 38.0 | 63.4 | 73.0 | 58.1 | 404.5 |
| | Ocr | 91.3 | 99.2 | 99.6 | 96.7 | 74.6 | 92.1 | 95.1 | 87.3 | 552.0 | 69.3 | 89.9 | 94.8 | 84.7 | 49.5 | 74.9 | 83.3 | 69.2 | 461.7 |
| | CI | 84.4 | 97.2 | 98.6 | 93.4 | 62.5 | 84.2 | 89.2 | 78.6 | 516.2 | 60.8 | 84.7 | 91.0 | 78.8 | 40.6 | 66.2 | 75.6 | 60.8 | 418.9 |
| | CR | 82.1 | 95.9 | 98.4 | 92.1 | 59.9 | 81.6 | 87.2 | 76.2 | 505.0 | 58.3 | 82.9 | 89.9 | 77.0 | 38.3 | 63.6 | 73.1 | 58.3 | 406.1 |
| | CS | 82.9 | 96.8 | 98.8 | 92.8 | 61.6 | 83.2 | 88.4 | 77.7 | 511.7 | 59.9 | 84.1 | 90.8 | 78.3 | 39.8 | 65.3 | 74.8 | 60.0 | 414.7 |
| | CD | 83.6 | 96.7 | 98.5 | 92.9 | 61.9 | 83.6 | 88.7 | 78.1 | 513.0 | 60.0 | 84.1 | 90.8 | 78.3 | 39.9 | 65.7 | 75.1 | 60.2 | 415.5 |
| Word | SR | 92.9 | 99.2 | 99.8 | 97.3 | 78.7 | 94.5 | 96.8 | 90.0 | 561.9 | 70.1 | 90.6 | 95.1 | 85.3 | 52.4 | 77.7 | 85.5 | 71.9 | 471.4 |
| | WI | 94.3 | 99.6 | 99.9 | 97.9 | 82.9 | 96.6 | 98.3 | 92.6 | 571.6 | 73.2 | 92.4 | 96.3 | 87.3 | 56.8 | 81.6 | 88.7 | 75.7 | 488.9 |
| | WS | 93.3 | 99.4 | 99.9 | 97.6 | 81.5 | 96.3 | 98.1 | 92.0 | 568.6 | 72.0 | 91.8 | 96.1 | 86.6 | 55.1 | 80.6 | 88.2 | 74.6 | 483.7 |
| | WD | 93.4 | 99.5 | 99.9 | 97.6 | 82.2 | 96.5 | 98.3 | 92.4 | 570.0 | 72.9 | 92.1 | 96.1 | 87.0 | 55.7 | 81.1 | 88.5 | 75.1 | 486.3 |
| | IP | 95.9 | 99.8 | 100.0 | 98.6 | 85.5 | 97.5 | 98.9 | 94.0 | 577.7 | 77.6 | 94.3 | 97.2 | 89.7 | 60.7 | 84.3 | 90.5 | 78.5 | 504.5 |
| Sentence | Formal | 95.4 | 99.7 | 99.9 | 98.3 | 85.2 | 97.3 | 98.7 | 93.7 | 576.2 | 77.6 | 94.1 | 97.0 | 89.6 | 60.2 | 83.9 | 90.3 | 78.1 | 503.1 |
| | Casual | 95.1 | 99.7 | 100.0 | 98.3 | 84.6 | 97.1 | 98.5 | 93.4 | 575.0 | 77.1 | 94.1 | 97.4 | 89.5 | 59.7 | 83.6 | 90.1 | 77.8 | 502.0 |
| | Passive | 94.6 | 99.4 | 100.0 | 98.0 | 81.5 | 96.1 | 98.0 | 91.8 | 569.5 | 76.1 | 93.4 | 96.7 | 88.7 | 58.4 | 82.6 | 89.2 | 76.7 | 496.4 |
| | Active | 95.6 | 99.8 | 100.0 | 98.5 | 85.0 | 97.3 | 98.7 | 93.7 | 576.4 | 77.5 | 94.2 | 97.1 | 89.6 | 60.4 | 84.2 | 90.3 | 78.3 | 503.7 |
| | Back_trans | 95.9 | 99.7 | 99.9 | 98.5 | 83.0 | 96.1 | 98.0 | 92.3 | 572.5 | 75.2 | 93.0 | 96.4 | 88.2 | 57.4 | 81.0 | 88.3 | 75.6 | 491.3 |

Table 37: TCL text perturbation performance comparison of Zero-Shot (ZS) image-text retrieval on Flickr30K and COCO datasets (results are averaged on five perturbation levels).

| | | Flickr30K (1K) | | | | | | | | MSCOCO (5K) | | | | | | | | |
|---|---|---|---|---|---|---|---|---|---|---|---|---|---|---|---|---|---|---|
| | Method | Text Retrieval | | | | Image Retrieval | | | | Text Retrieval | | | | Image Retrieval | | | | |
| | | R@1 | R@5 | R@10 | Mean | R@1 | R@5 | R@10 | Mean | RSUM | R@1 | R@5 | R@10 | Mean | R@1 | R@5 | R@10 | Mean | RSUM |
| Character | Keyboard | 63.8 | 87.2 | 92.7 | 81.2 | 44.1 | 68.8 | 76.7 | 63.2 | 433.3 | 49.6 | 76.1 | 84.9 | 70.2 | 32.3 | 57.2 | 67.8 | 52.4 | 368.0 |
| | Ocr | 78.2 | 94.8 | 97.9 | 90.3 | 58.8 | 82.1 | 88.1 | 76.3 | 499.9 | 61.4 | 85.1 | 91.6 | 79.4 | 42.6 | 69.0 | 78.7 | 63.4 | 428.4 |
| | CI | 67.3 | 88.0 | 93.4 | 82.9 | 45.9 | 70.5 | 78.3 | 64.9 | 443.3 | 51.9 | 78.5 | 86.7 | 72.4 | 34.1 | 59.8 | 70.3 | 54.7 | 381.3 |
| | CR | 63.1 | 85.9 | 91.4 | 80.1 | 43.8 | 68.1 | 76.1 | 62.7 | 428.4 | 49.7 | 76.1 | 85.1 | 70.3 | 32.2 | 57.4 | 67.9 | 52.5 | 368.4 |
| | CS | 66.5 | 88.6 | 93.8 | 83.0 | 46.3 | 70.8 | 78.5 | 65.2 | 444.4 | 52.6 | 78.5 | 87.0 | 72.7 | 34.0 | 59.7 | 70.1 | 54.6 | 382.0 |
| | CD | 66.7 | 89.4 | 94.2 | 83.4 | 47.2 | 71.9 | 79.4 | 66.2 | 448.9 | 52.6 | 78.8 | 86.9 | 72.8 | 34.3 | 60.2 | 70.6 | 55.0 | 383.4 |
| Word | SR | 78.3 | 95.3 | 97.9 | 90.5 | 63.2 | 86.0 | 91.1 | 80.1 | 511.9 | 62.1 | 85.7 | 91.9 | 79.9 | 45.8 | 72.3 | 81.5 | 66.5 | 439.3 |
| | WI | 80.0 | 96.3 | 98.5 | 91.6 | 67.0 | 88.6 | 93.4 | 83.0 | 523.8 | 63.3 | 86.8 | 93.0 | 81.0 | 49.5 | 76.1 | 84.7 | 70.1 | 453.4 |
| | WS | 80.4 | 95.9 | 98.4 | 91.6 | 64.8 | 87.2 | 92.4 | 81.5 | 519.1 | 63.2 | 86.5 | 92.7 | 80.8 | 46.5 | 73.8 | 83.0 | 67.8 | 445.7 |
| | WD | 83.6 | 97.1 | 98.8 | 93.1 | 67.0 | 89.0 | 93.4 | 83.1 | 528.8 | 65.3 | 87.2 | 93.1 | 81.9 | 47.6 | 74.4 | 83.3 | 68.4 | 450.9 |
| | IP | 89.4 | 98.6 | 99.6 | 95.9 | 73.4 | 92.2 | 95.5 | 87.0 | 548.6 | 71.4 | 90.8 | 95.4 | 85.9 | 53.5 | 79.0 | 87.1 | 73.2 | 477.2 |
| Sentence | Formal | 88.0 | 98.0 | 99.8 | 95.3 | 72.0 | 91.6 | 95.1 | 86.2 | 544.4 | 70.8 | 90.6 | 95.2 | 85.5 | 52.9 | 78.4 | 86.5 | 72.6 | 474.4 |
| | Casual | 87.2 | 98.3 | 99.5 | 95.0 | 71.4 | 91.2 | 94.8 | 85.8 | 542.4 | 69.9 | 90.2 | 94.9 | 85.0 | 52.3 | 78.1 | 86.4 | 72.3 | 471.8 |
| | Passive | 84.5 | 97.1 | 99.4 | 93.7 | 67.6 | 88.6 | 92.9 | 83.0 | 530.1 | 68.6 | 89.1 | 94.4 | 84.0 | 50.5 | 76.9 | 85.2 | 70.9 | 464.7 |
| | Active | 89.3 | 98.3 | 99.9 | 95.8 | 72.9 | 91.5 | 95.1 | 86.5 | 547.1 | 70.9 | 90.6 | 95.3 | 85.6 | 53.1 | 78.9 | 86.9 | 73.0 | 475.7 |
| | Back_trans | 86.0 | 97.6 | 99.4 | 94.3 | 69.4 | 89.8 | 93.6 | 84.3 | 535.8 | 68.5 | 89.2 | 94.2 | 83.9 | 50.3 | 75.9 | 84.1 | 70.1 | 462.0 |

Table 38: TCL text perturbation performance comparison of Fine-tuned (FT) image-text retrieval on Flickr30K and COCO datasets (results are averaged on five perturbation levels).

| | | Flickr30K (1K) | | | | | | | | MSCOCO (5K) | | | | | | | | |
|---|---|---|---|---|---|---|---|---|---|---|---|---|---|---|---|---|---|---|
| | Method | Text Retrieval | | | | Image Retrieval | | | | Text Retrieval | | | | Image Retrieval | | | | |
| | | R@1 | R@5 | R@10 | Mean | R@1 | R@5 | R@10 | Mean | RSUM | R@1 | R@5 | R@10 | Mean | R@1 | R@5 | R@10 | Mean | RSUM |
| Character | Keyboard | 79.7 | 95.2 | 97.9 | 90.9 | 57.0 | 79.1 | 85.4 | 73.8 | 494.3 | 55.8 | 81.3 | 88.8 | 75.3 | 36.9 | 62.5 | 72.4 | 57.3 | 397.8 |
| | Ocr | 90.0 | 99.1 | 99.7 | 96.3 | 71.7 | 90.4 | 94.0 | 85.4 | 545.0 | 67.6 | 88.9 | 94.0 | 83.5 | 48.0 | 73.9 | 82.6 | 68.2 | 455.1 |
| | CI | 82.2 | 96.2 | 98.3 | 92.2 | 59.6 | 81.4 | 87.2 | 76.1 | 504.9 | 58.5 | 83.5 | 90.4 | 77.5 | 39.3 | 65.3 | 75.0 | 59.8 | 412.0 |
| | CR | 79.3 | 94.8 | 97.8 | 90.7 | 56.7 | 79.1 | 85.0 | 73.6 | 492.8 | 55.6 | 81.5 | 89.0 | 75.4 | 37.2 | 62.7 | 72.5 | 57.5 | 398.5 |
| | CS | 80.7 | 96.0 | 98.2 | 91.6 | 59.0 | 81.2 | 86.8 | 75.7 | 501.9 | 57.6 | 82.9 | 90.2 | 76.9 | 38.7 | 64.8 | 74.6 | 59.4 | 408.8 |
| | CD | 81.4 | 95.7 | 98.3 | 91.8 | 59.1 | 81.2 | 86.7 | 75.7 | 502.4 | 58.1 | 83.0 | 90.0 | 77.0 | 39.2 | 65.3 | 75.0 | 59.8 | 410.5 |
| Word | SR | 91.0 | 99.1 | 99.7 | 96.6 | 76.1 | 93.0 | 95.8 | 88.3 | 554.7 | 67.8 | 89.1 | 94.2 | 83.7 | 51.0 | 76.8 | 84.8 | 70.8 | 463.7 |
| | WI | 93.4 | 99.4 | 99.8 | 97.5 | 80.5 | 95.5 | 97.7 | 91.2 | 566.4 | 70.8 | 91.0 | 95.6 | 85.8 | 55.3 | 80.6 | 88.0 | 74.6 | 481.3 |
| | WS | 91.0 | 99.1 | 99.6 | 96.6 | 78.2 | 94.7 | 97.4 | 90.1 | 560.0 | 69.2 | 90.3 | 94.9 | 84.8 | 52.3 | 78.5 | 86.6 | 72.5 | 471.8 |
| | WD | 92.6 | 99.4 | 99.8 | 97.3 | 79.5 | 95.3 | 97.6 | 90.8 | 564.2 | 70.8 | 90.7 | 95.5 | 85.7 | 53.7 | 79.7 | 87.3 | 73.6 | 477.7 |
| | IP | 94.9 | 99.5 | 99.8 | 98.1 | 84.0 | 96.7 | 98.5 | 93.1 | 573.4 | 75.6 | 92.8 | 96.7 | 88.3 | 59.0 | 83.2 | 89.9 | 77.3 | 497.1 |
| Sentence | Formal | 94.4 | 99.4 | 99.8 | 97.9 | 83.2 | 96.5 | 98.3 | 92.6 | 571.5 | 75.3 | 92.4 | 96.7 | 88.1 | 58.2 | 82.7 | 89.5 | 76.8 | 494.6 |
| | Casual | 94.0 | 99.5 | 99.9 | 97.8 | 82.1 | 96.0 | 98.0 | 92.1 | 569.6 | 74.6 | 92.1 | 96.5 | 87.8 | 57.9 | 82.5 | 89.4 | 76.6 | 493.0 |
| | Passive | 92.7 | 99.1 | 99.8 | 97.2 | 79.5 | 94.5 | 97.1 | 90.4 | 562.8 | 73.5 | 91.9 | 96.1 | 87.2 | 56.3 | 81.3 | 88.3 | 75.3 | 487.3 |
| | Active | 94.8 | 99.5 | 99.8 | 98.0 | 83.5 | 96.4 | 98.2 | 92.7 | 572.1 | 75.4 | 92.7 | 96.6 | 88.2 | 58.7 | 83.0 | 89.7 | 77.1 | 496.0 |
| | Back_trans | 93.9 | 99.5 | 99.9 | 97.8 | 80.6 | 95.3 | 97.3 | 91.1 | 566.5 | 72.7 | 91.6 | 96.0 | 86.8 | 55.5 | 80.3 | 87.3 | 74.4 | 483.5 |

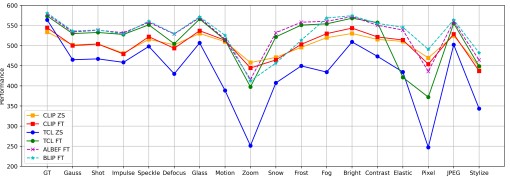

Figure 22: Image-text retrieval results on Flick30K-IP.

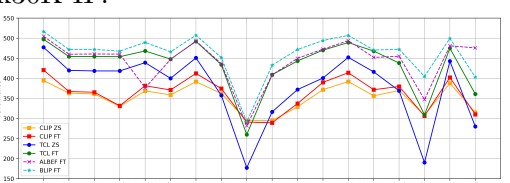

Figure 23: Image-text retrieval results on COCO-IP.

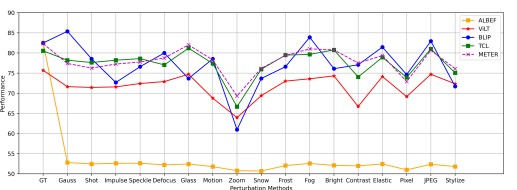

Figure 24: Visual reasoning results on NLVR-IP dev set.

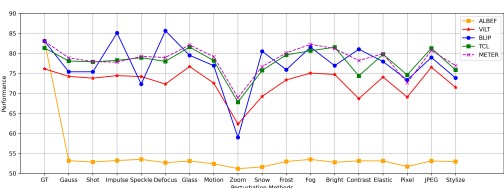

Figure 25: Visual reasoning results on NLVR-IP test set.

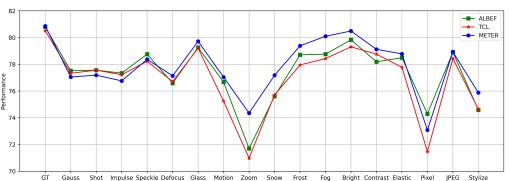

Figure 26: Visual entailment results on SNLI-VE-IP val set.

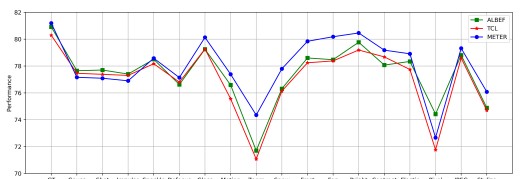

Figure 27: Visual entailment results on SNLI-VE-IP test set.

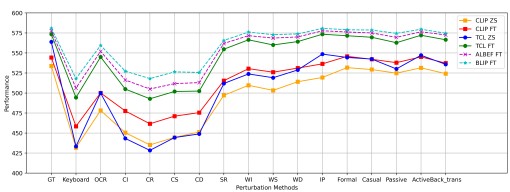

Figure 28: Image-text retrieval results on Flick30K-TP.

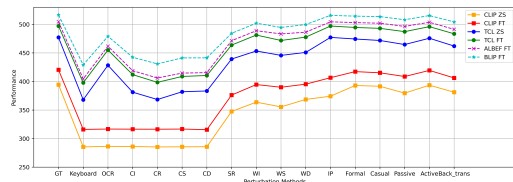

Figure 29: Image-text retrieval results on COCO-TP.

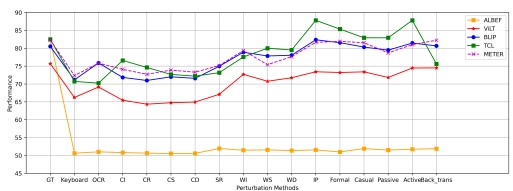

Figure 30: Visual reasoning results on NLVR-TP dev set.

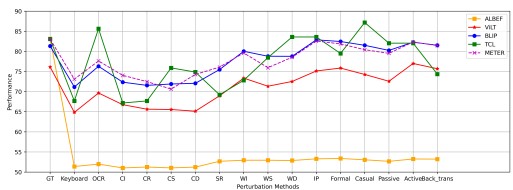

Figure 31: Visual reasoning results on NLVR-TP test set.

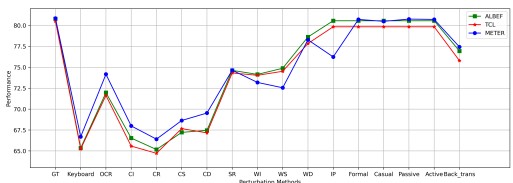

Figure 32: Visual entailment results on SNLI-VE-TP val set.

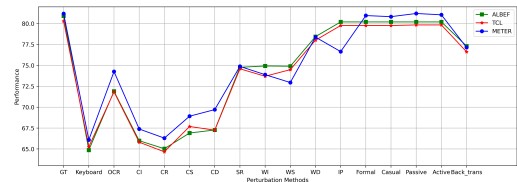

Figure 33: Visual entailment results on SNLI-VE-TP test set.

