# OpenReview forum: "Benchmarking Robustness of Multimodal Image-Text Models under Distribution Shift"
_DMLR — Accepted by DMLR_

### Review · Reviewer_N3DS · 2023-11-18

**Recommendation:** 3
**Confidence:** 2

**Summary Of Contributions:**

This paper presents a comprehensive study on the robustness of multimodal image-text models under distribution shift. On 5 common image-text tasks, the authors analyze the robustness of 9 open-sourced image-text models under various types of image perturbations and text perturbations. Two new metrics (MMI and MOR) are introduced for evaluating the robustness. Through experiments, the authors show that multimodel image-text models are not robust to the distribution shift caused by image or text perturbations, especially the shift in image spaces. They also find that "zoom blur" is more impactful than other image perturbations, and that "character perturbation" is more impactful than other text perturbations.

**Strengths:**

The paper is well-organized and easy to follow. The experimental sections are very comprehensive and detailed. The authors choose 5 representative tasks for image-text models, and make great effort on benchmarking the robustness under various types of image or text perturbations. The message is clear and the results are convicing.

**Claims And Evidence:**

Yes

**Datasets And Benchmarks:**

Yes

**Extended Submissions:**

Yes

**Limitations:**

1. My primary concern lies on the limited analysis of models and datasets in this paper. While the authers did comprehensive experiments on 17 image perturbations and 16 text perturbations with 5 levels of severity, the influence of models and datasets are less considered. Only 5 datasets and 9 models are considered in this paper. Given large volumns of relevant models and datasets in this field, I would be worried that whether the conclusions could be extrapolated to other models and datasets. Moreover, a discussion of the influences of models, datasets, training and inference strategies on robustness would be beneficial.

2. To make a fair comparison between different image perturbations, it'd be good to make sure that they're under the same "perturbation budget" on changing the input's information. For text perturbations, this is ensured by the threshold of fidelity. However, a corresponding metric is lacking for image perturbations.

3. It'd also be good to discuss the relationship between robustness under distribution shift and robustness under adversarial perturbations for multimodal image-text models.

**Requested Changes:**

1. More analysis on the influence "model" and "dataset" part to the robustness to image or text perturbation.

2. Discussion on the relationship between robustness under distribution shift and robustness under adversarial perturbations.

---

### Review · Reviewer_6qDS · 2023-11-20

**Recommendation:** 3
**Confidence:** 3

**Summary Of Contributions:**

This study introduces a benchmark designed to assess vision-language models in the face of artificial distribution shifts, which are instantiated by image-based corruptions and text-based corruptions. Building on previous research, the benchmark incorporates 17 methods of image perturbation and 16 text perturbation techniques, most varying in severity. The evaluation of 9 representative vision-language models and 5 standard tasks reveals the vulnerabilities of the models to various types of perturbation, with certain models exhibiting an increased sensitivity to the disturbances and certain types of perturbation showing more impact. In addition, the paper also proposed two metrics, the Multimodal Index (MMI) score and the Missing Object Rate (MOS) score, to better evaluate the robustness.

**Strengths:**

1.	Benchmarking the robustness of vision-language models is crucial for an in-depth assessment of their capabilities. This work represents the first systematic study in this area, providing a comprehensive analysis that covers 33 types of perturbations, 9 vision-language models, and 7 representative tasks.

2.	Two new metrics are introduced for evaluating model robustness. The first, the MMI score, measures the relative performance drop to isolate the effects of distribution shifts from performance on clean data. The second, the missing object rate, assesses the accuracy of generated images compared to their captions by quantifying missing captioned objects in the images.

3.	Overall, the paper is well-presented. Its logical flow is clear and coherent, while the figures and tables effectively aid in understanding the evaluations.

4.	The provided evaluation codebase is well-documented and seems easy to use.

**Broader Impact Concerns:**

No additional concerns beyond those discussed in the Broader Impact section.

**Claims And Evidence:**

Yes

**Datasets And Benchmarks:**

Yes

**Extended Submissions:**

No

**Limitations:**

**Weaknesses:**

1.	The benchmarking results for recent LLM-based vision-language models such as LLaVa [1], Mini-GPT4 [2], and BLIP 2 [3] are not included. Given the remarkable capabilities of LLMs for visual understanding, their robustness is of crucial interest to a wider audience and vital for practical applications.
2.	The paper could be strengthened by incorporating the Language Rewrite [4] technique as a text-based augmentation. This technique leverages LLMs to produce varied yet semantically coherent captions, which could provide additional insights into model robustness.
3.	Although four questions were posited at the beginning of Section 4, there is no clear answer or summary provided for these queries. A concise summary addressing these points would significantly help readers in capturing the key messages from the benchmark results.

**Limitations:**

The paper focuses on synthetic distribution shifts created by artificial perturbations. This does not adequately emulate the complex nature of real-world scenarios, which could limit the applicability of the findings to practical applications.


**References**

[1] Liu, Haotian, et al. "Visual instruction tuning." arXiv preprint arXiv:2304.08485 (2023).

[2] Zhu, Deyao, et al. "Minigpt-4: Enhancing vision-language understanding with advanced large language models." arXiv preprint arXiv:2304.10592 (2023).

[3] Li, Junnan, et al. "Blip-2: Bootstrapping language-image pre-training with frozen image encoders and large language models." arXiv preprint arXiv:2301.12597 (2023).

[4] Fan, Lijie, et al. "Improving CLIP Training with Language Rewrites." arXiv preprint arXiv:2305.20088 (2023).

**Requested Changes:**

Those listed in the Weaknesses.

Minor: Some of the tables are significantly overlengthy. Consider rotating the names of perturbations by 90 degrees to fit within the linewidth, thus improving readability.

---

### Review · Reviewer_JhTC · 2023-11-20

**Recommendation:** 3
**Confidence:** 2

**Summary Of Contributions:**

The paper addresses the vulnerability of multimodal models under distribution shifts, illustrated through examples of image captioning and text-to-image generation errors induced by simple perturbations. It highlights a lack of comprehensive studies and benchmarks in the robustness evaluation of multimodal image-text models under such conditions. To fill this gap, the authors create benchmarks for multimodal robustness evaluation, introduce new robustness metrics (MMI and MOR), and find that these models are more sensitive to image perturbations, with zoom blur and character-level text perturbations being particularly impactful.

**Strengths:**

1. Creating benchmarks for robustness evaluation, leveraging existing datasets and tasks, which is a crucial step in understanding and improving the resilience of these models.

2. Introducing innovative robustness metrics, such as the MultiModal Impact score (MMI) and Missing Object Rate (MOR), offering new tools to assess model performance under perturbed conditions effectively.

3. Providing detailed insights into how different types of perturbations (image and text) affect multimodal models, with specific findings like the higher impact of image perturbations and the effectiveness of zoom blur and character-level text perturbations. This granular analysis is valuable for future research and development in the field.

**Claims And Evidence:**

Yes

**Datasets And Benchmarks:**

Yes

**Extended Submissions:**

No

**Limitations:**

1. The type of perturbation on images mainly focus on noise, blur, weather, digital, and stylize. While the reviewer understands these types of attacks are important in studying distribution shift, is it possible to include learning-based distribution shift (e.g. adversarial robustness) into the evaluation dataset?

**Requested Changes:**

It would greatly strengthen the work if the authors can include learning-based distribution shift into their benchmarks.